# Expressive Higher-Order Link Prediction through Hypergraph Symmetry Breaking

**Simon Zhang** *zhan4125@purdue.edu*
*Department of Computer Science*
*Purdue University*

**Cheng Xin** *cx122@cs.rutgers.edu*
*Department of Computer Science*
*Rutgers University*

**Tamal K. Dey** *tamaldey@purdue.edu*
*Department of Computer Science*
*Purdue University*

**Reviewed on OpenReview:** *https://openreview.net/forum?id=oG65SjZNIF*

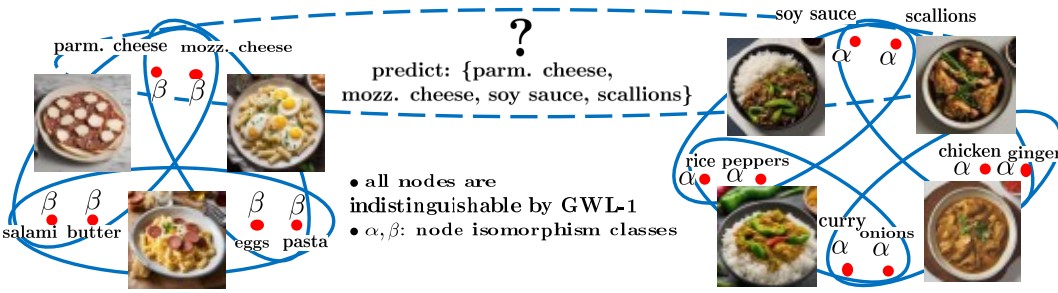

Figure 1: An illustration of a hypergraph of recipes. The nodes are the ingredients and the hyperedges are the recipes. The task of higher order link prediction is to predict hyperedges in the hypergraph. A negative hyperedge sample would be the dotted hyperedge. The Asian ingredient nodes ($\alpha$) and the European ingredient nodes ($\beta$) form two separate isomorphism classes. However, GWL-1 cannot distinguish between these classes and will predict a false positive for the negative sample.

## Abstract

A hypergraph consists of a set of nodes along with a collection of subsets of the nodes called hyperedges. Higher order link prediction is the task of predicting the existence of a missing hyperedge in a hypergraph. A hyperedge representation learned for higher order link prediction is fully expressive when it does not lose distinguishing power up to an isomorphism. Many existing hypergraph representation learners, are bounded in expressive power by the Generalized Weisfeiler Lehman-1 (GWL-1) algorithm, a generalization of the Weisfeiler Lehman-1 (WL-1) algorithm. The WL-1 algorithm can approximately decide whether two graphs are isomorphic. However, GWL-1 has limited expressive power. In fact, GWL-1 can only view the hypergraph as a collection of trees rooted at each of the nodes in the hypergraph. Furthermore, message passing on hypergraphs can already be computationally expensive, particularly with limited GPU device memory. To address these limitations, we devise a preprocessing algorithm that can identify certain regular subhypergraphs exhibiting symmetry with respect to GWL-1. Our preprocessing algorithm runs once with the time complexity linear in the size of the input hypergraph. During training, we randomly

drop the hyperedges of the subhypergraphs identifed by the algorithm and add covering hyperedges to break symmetry. We show that our method improves the expressivity of GWL-1. Our extensive experiments [1] also demonstrate the effectiveness of our approach for higher-order link prediction on both graph and hypergraph datasets with negligible change in computation.

## 1    Introduction

Hypergraphs can model complex relationships in real-world networks, extending beyond the pairwise connections captured by traditional graphs. Figure 1 is an example hypergraph consisting of recipes of two different types of dishes, which are largely determined by the ingredients to be used. In this hypergraph, the hyperedges are the recipes, and the nodes are the ingredients used in each recipe. The Asian recipes are presented in the right part of the figure, which consist of combinations of the ingredients of soy sauce, scallions, rice, peppers, chicken, ginger, curry and onions. The European recipes are presented in the left part of the figure, which consist of combinations of the ingredients of Parmesan cheese, mozzarella cheese, salami, butter, eggs, and pasta.

Hypergraphs have found applications in diverse fields such as recommender systems Lü et al. (2012), visual classification Feng et al. (2019), and social networks Li et al. (2013). Higher-order link prediction is the task of predicting missing hyperedges in a hypergraph. For this task, when the hypergraph is unattributed, it is important to respect the hypergraph's symmetries, or automorphism group. This brings about challenges to learning an expressive view of the hypergraph.

A hypergraph neural network (hyperGNN) is any neural network that learns on a hypergraph. This is in analogy to a graph neural network (GNN), which is a neural network that learns on a graph. Many existing hyperGNN architectures follow a computational message passing model called the Generalized Weisfeiler Lehman-1 (GWL-1) algorithm Huang & Yang (2021), a hypergraph isomorphism testing approximation scheme. GWL-1 is a generalization of the message passing algorithm called Weisfeiler Lehman-1 (WL-1) algorithm Weisfeiler & Leman (1968) used for graph isomorphism testing.

GWL-1, like WL-1 on graphs, is limited in how well it can express its input. Specifically, by viewing a hypergraph as a collection of rooted trees, GWL-1 loses topological information of its input hypergraph and thus cannot fully recognize the symmetries, or automorphism group, of the hypergraph. In fact, the hyperGNN views the hypergraph as having false-positive symmetries. For a task like transductive hyperlink prediction, this can result in predicting false-positive hyperlinks as shown in Figure 1. Furthermore, such an issue can become even worse during test time since the automorphism group of the hypergraph might change. It is thus important to find a way to improve the expressivity of existing hyperGNNs.

Let a hypergraph $\mathcal{H} = (\mathcal{V}, \mathcal{E})$ denote a pair where $\mathcal{V}$ is a set of nodes and $\mathcal{E} \subseteq 2^{\mathcal{V}}$, a collection of subsets of $\mathcal{V}$, indexes a set of hyperedges. GWL-1 views a hypergraph as a collection of trees rooted at the nodes. These rooted trees are formed by viewing each node-hyperedge incidence as an edge and recursively expanding about the nodes and hyperedges alternately. As an example of the GWL-1 algorithm, one step of GWL-1 would output a collection of depth 1 trees rooted at each node where leaves are the incident hyperedges of each node. Two steps of GWL-1 would output a collection of depth 2 trees where the depth 1 trees of one step of GWL-1 have their leaves expanded with the incident nodes of the hyperedges the new leaves. This leaf expansion process can be repeated, alternating nodes and hyperedges. This is the recursive expansion of the GWL-1 algorithm.

We can recover hyperGNNs by expressing the computation of GWL-1 as a matrix equation. Parameterizing the expression with learnable weights, GWL-1 becomes a neural network, called a hypergraph neural network (hyperGNN), similar to graph neural networks (GNN)s. In practice this is implemented through repeated sparse matrix multiplication.

Computing on a hypergraph can also be very expensive. The subsets $e \in \mathcal{E}$ that contain a node $v \in \mathcal{V}$ form the neighborhood of $v \in \mathcal{V}$. This means just the neighborhood size of the nodes in hypergraphs can

---

[1] https://github.com/simonzhang00/HypergraphSymmetryBreaking

grow exponentially with the number of nodes of the hypergraph. Thus, a computationally more expensive message passing scheme over GWL-1 based hyperGNNs may bring difficulties.

In order to address the issue of the expressivity of hyperGNNS for the task of hyperlink prediction while also respecting the computational complexity of computing on a hypergraph, we devise a method that selectively breaks the symmetry of the hypergraph topology itself coming from the limitations of the hyperGNN architecture. Our method is designed as an efficient preprocessing algorithm that can improve the expressive power of GWL-1 for higher order link prediction. Since the preprocessing only runs once with complexity linear in the input, we do not increase the computational complexity of training.

Similar to a substructure counting algorithm Bouritsas et al. (2022), we identify certain symmetries in induced subhypergraphs. However, unlike in existing work where node attributes are modified, such as random noise being appended to the node attributes Sato et al. (2021), we directly target and modify the symmetries in the topology. This limits the space for augmentation, which can prevent extreme perturbations of the data. The algorithm identifies a cover of the hypergraph by disjoint connected components whose nodes are indistinguishable by GWL-1. During training, we randomly replace the hyperedges of the identified symmetric regular induced subhypergraphs with single hyperedges that cover the nodes of each subhypergraph. We show that our method of hyperedge hallucination to break symmetry can increase the expressivity of existing hypergraph neural networks both theoretically and experimentally.

**Contributions.** In the context of hypergraph representation learning and hyperlink prediction, we have a method that can break the symmetries introduced by conventional hypergraph neural networks. Conventional hypergraph neural networks are based on the GWL-1 algorithm on hypergraphs. However, the GWL-1 algorithm on hypergraphs views the hypergraph as a collection of rooted trees. This introduces false positive symmetries. We summarize our contributions in this work as follows:

- **Provide a formal analysis of GWL-1 on hypergraphs** from the perspective of algebraic topology. Our analysis offers a novel characterization of the expressive power and limitations of GWL-1. By leveraging concepts from algebraic topology, we establish a precise connection between GWL-1 and the universal covers of hypergraphs, providing deeper insights into the algorithm's behavior on complex hypergraph structures.

- **Devise an efficient hypergraph preprocessing algorithm** to identify false positive symmetries of GWL-1. We propose a linear time preprocessing algorithm to identify specific regular subhypergraphs that exhibit symmetry with respect to GWL-1, which are potential sources of expressivity limitations.

- **Introduce a data augmentation scheme** that leverages the preprocessing algorithm's output to improve GWL-1's expressivity. During training, we randomly modify the hyperedges of the identified symmetric subhypergraphs, effectively breaking symmetries that GWL-1 cannot distinguish. This approach enhances the model's ability to capture fine-grained structural information without significantly increasing computational complexity.

- **Provide formal analysis and performance guarantees** for our method. We rigorously prove how our approach improves the expressivity of GWL-1 under certain conditions. These theoretical results offer valuable insights into the circumstances under which our method can enhance hypergraph representation learning, providing a solid foundation for its practical application.

- **Perform extensive experiments on real-world datasets** to demonstrate the effectiveness of our approach. Our comprehensive evaluation spans various hypergraph and graph datasets, showcasing consistent improvements across different hypergraph neural network architectures for higher-order link prediction tasks. These empirical results validate the practical utility of our method and its ability to enhance existing models with minimal computational overhead.

## 2 Background

The following notation is used throughout the paper:

Let $\mathbb{N} \triangleq \{0, 1, ...\}, \mathbb{Z} \triangleq \{..., -1, 0, 1, ...\}, \mathbb{Z}^+ \triangleq \{1, ...\}$, and $\mathbb{R}$ denote the naturals, integers, positive integers, and real numbers respectively. Let $[n] \triangleq \{1, ..., n\} \subseteq \mathbb{Z}^+$ denote the integers from 1 to $n \in \mathbb{Z}^+$.

For a set $A$, let $\binom{A}{k}$ denote the set of all subsets of $A$ of size $k$. Given the set $A$, a multiset is defined by $\tilde{A} \triangleq (A, m)$, $m : A \to \mathbb{Z}^+$. A set is also a multiset with $m = \mathbf{1}$. A submultiset $\tilde{B} \subseteq \tilde{A}$ is defined by $\tilde{B} \triangleq (B, m')$ with $B \subseteq A$ and $m'(e) \leq m(e), \forall e \in B$. A multiset with its elements explicitly enumerated with the double curly brace notation: $\{\!\{a, a, a, ...\}\!\}$. The cardinality of a multiset $\tilde{A}$, is defined as $|\tilde{A}| \triangleq \sum_{e \in A} m(e)$. For two multisets $\tilde{A} = (A, m_A), \tilde{B} = (B, m_B)$, we may define their multiset sum by $\tilde{A} \sqcup \tilde{B} \triangleq (A \cup B, m_{A \cup B} = m_A + m_B)$.

For two pairs of multisets $\tilde{A} = (\tilde{A}_1, \tilde{A}_2), \tilde{B} = (\tilde{B}_1, \tilde{B}_2)$, denote their multiset sum by their elementwise multiset sum: $\tilde{A} \sqcup \tilde{B} \triangleq (\tilde{A}_1 \sqcup \tilde{B}_1, \tilde{A}_2 \sqcup \tilde{B}_2)$. Similarly, for two pairs of sets $A = (A_1, A_2), B = (B_1, B_2)$, denote their union by their elementwise union: $A \cup B \triangleq (A_1 \cup B_1, A_2 \cup B_2)$.

Let $P_t \triangleq P(\bullet; t)$ denote a probability distribution parameterized by $t \in \mathbb{R}$. The distribution $P_t$ has some domain $\mathcal{D}$, which we denote by $\mathrm{dom}(P_t)$. The notation $\mathrm{supp}(P_t) \triangleq \{x \in \mathrm{dom}(P_t) : P_t(x) > 0\}$ denotes the **support** of a distribution $P_t$.

A Bernoulli distribution $P_p \triangleq P(X; p)$ is a probability distribution parameterized by a $p : 0 < p < 1$ with the following definition:

$$P_p(X = k) = \begin{cases} p & k = 1 \\ 1 - p & k = 0 \end{cases} \tag{1}$$

The random variable $Bernoulli(p) \sim P_p$ is called a Bernoulli random variable.

## 2.1 Group Theory

To better study hypergraphs, we use some concepts of group theory. Here we give a brief introduction to some of them. For more details, see Dummit & Foote (2004).

**Definition 2.1.** *A* group *$G$ is a set equipped with a binary operation "$*$" that satisfies the following four properties:*

1. ***Closure**: For every $a, b \in G$, the result of the operation $a * b$ is also in $G$.*

2. ***Associativity**: For every $a, b, c \in G$, $(a * b) * c = a * (b * c)$.*

3. ***Identity Element**: There exists an element $e \in G$ such that for every $a \in G$, $e * a = a * e = a$.*

4. ***Inverse Element**: For each $a \in G$, there exists an element $b \in G$ such that $a * b = b * a = e$ (such element $b$ is unique for $a$ and is often denoted as $a^{-1}$.)*

**Permutation Groups**   A *permutation group* is a group where the elements are permutations of a set, and the group operation is the composition of these permutations. Permutations are bijective functions that rearrange the elements of a set.

**Group Isomorphism and Automorphism**   Two groups $G$ and $H$ are called *isomorphic* if there exists a bijective function $\phi : G \to H$ such that for all $a, b \in G, \phi(a * b) = \phi(a) * \phi(b)$. This means that $G$ and $H$ have the same group structure, even if their elements are different. Such bijective funtions are called *isomorphisms*. If $G = H$, an isomorphism to a group itself is called an *automorphism*. The set of all automorphisms of a group $G$ forms a group, with group operations given by compositions. Such group is called the *automorphism group* of $G$, denoted as $Aut(G)$.

**Group Action**   A group action is a formal way in which a group $G$ operates on a set $X$. Formally, a group action is a map $G \times X \to X$ (denoted $(g, x) \mapsto g \cdot x$) that satisfies the following two properties:

1. For all $x \in X$, $e \cdot x = x$ where $e$ is the identity element in $G$.

2. For all $g, h \in G$ and $x \in X$, $(gh) \cdot x = g \cdot (h \cdot x)$.

Group actions are useful for studying the symmetry of objects, as they describe how the elements of a group move or transform the elements of a set.

**Stabilizer** In the context of group actions, the stabilizer of an element $x$ in a set $X$ (where a group $G$ acts on $X$) is the set of elements in $G$ that leave $x$ fixed. Formally, the stabilizer of $x$ in $G$ is given by:

$$Stab_G(x) = \{g \in G \mid g \cdot x = x\}$$

### 2.2 Isomorphisms on Higher Order Structures

In this section, we go over what a hypergraph is and how this structure is represented as tensors. We then define what a hypergraph isomorphism is.

A hypergraph is a generalization of a graph. Hypergraphs allow for all possible subsets over a set of vertices, called hyperedges. We can thus formally define a hypergraph as:

**Definition 2.2.** *An undirected hypergraph is a pair $\mathcal{H} = (\mathcal{V}, \mathcal{E})$ consisting of a set of vertices $\mathcal{V}$ and a set of hyperedges $\mathcal{E} \subseteq 2^{\mathcal{V}}$ where $2^{\mathcal{V}}$ is the power set of the vertex set $\mathcal{V}$.*

For a given hypergraph $\mathcal{H} = (\mathcal{V}, \mathcal{E})$, a hypergraph $\mathcal{G} = (\mathcal{V}', \mathcal{E}')$ is a **subhypergraph** of $\mathcal{H}$ if $\mathcal{V}' \subseteq \mathcal{V}$ and $\mathcal{E}' \subseteq \mathcal{E}$.

A subhypergraph **induced** by $\mathcal{W} \subseteq \mathcal{V}$ is defined as $(\mathcal{W}, \mathcal{F} = 2^{\mathcal{W}} \cap \mathcal{E})$. Similarly, for a subset of hyperedges $\mathcal{F} \subseteq \mathcal{E}$, a subhypergraph **induced** by $\mathcal{F}$ is defined as $(\bigcup_{e \in \mathcal{F}} e, \mathcal{F})$

For a given hypergraph $\mathcal{H}$, we also use $\mathcal{V}_{\mathcal{H}}$ and $\mathcal{E}_{\mathcal{H}}$ to denote the sets of vertices and hyperedges of $\mathcal{H}$ respectively. According to the definition, a hyperedge is a nonempty subset of the vertices. A hypergraph with all hyperedges the same size $d$ is called a $d$-uniform hypergraph. A 2-uniform hypergraph is an undirected graph, or just graph.

When viewed combinatorially, a hypergraph can include some symmetries that are captured by isomorphisms. These isomorphisms are defined by bijective structure preserving maps.

**Definition 2.3.** *For two hypergraphs $\mathcal{H}$ and $\mathcal{D}$, a structure preserving map $\rho : \mathcal{H} \to \mathcal{D}$ is a pair of maps $\rho = (\rho_{\mathcal{V}} : \mathcal{V}_{\mathcal{H}} \to \mathcal{V}_{\mathcal{D}}, \rho_{\mathcal{E}} : \mathcal{E}_{\mathcal{H}} \to \mathcal{E}_{\mathcal{D}})$ such that $\forall e \in \mathcal{E}_{\mathcal{H}}, \rho_{\mathcal{E}}(e) \triangleq \{\rho_{\mathcal{V}}(v_i) \mid v_i \in e\} \in \mathcal{E}_{\mathcal{D}}$. A hypergraph isomorphism is a structure preserving map $\rho = (\rho_{\mathcal{V}}, \rho_{\mathcal{E}})$ such that both $\rho_{\mathcal{V}}$ and $\rho_{\mathcal{E}}$ are bijective. Two hypergraphs are said to be isomorphic, denoted as $\mathcal{H} \cong \mathcal{D}$, if there exists an isomorphism between them. When $\mathcal{H} = \mathcal{D}$, an isomorphism $\rho$ is called an automorphism on $\mathcal{H}$. All the automorphisms form a group, which we denote as $Aut(\mathcal{H})$.*

A graph isomorphism is the special case of a hypergraph isomorphism between 2-uniform hypergraphs according to Definition 2.3.

A *neighborhood* $N(v) \triangleq (\bigcup_{v \in e} e, \{e : v \in e\})$ of a node $v \in \mathcal{V}$ of a hypergraph $\mathcal{H} = (\mathcal{V}, \mathcal{E})$ is the subhypergraph of $\mathcal{H}$ induced by the set of all hyperedges incident to $v$. The *degree* of $v$ is denoted $deg(v) = |\mathcal{E}_{N(v)}|$. A degree vector of a node $v$ of $\mathcal{H}$ is defined as: $\text{degvec}_{\mathcal{H}}(v) = (|\{e : e \ni v, |e| = k\}|)_{k=1}^{n}$

A simple but very common symmetric hypergraph is of importance to our task, namely the neighborhood-regular hypergraph, or just regular hypergraph.

**Definition 2.4.** *A neighborhood-regular hypergraph is a hypergraph where all neighborhoods of each node are isomorphic to each other.*

A $d$-uniform neighborhood of $v$ is the set of all hyperedges of size $d$ in the neighborhood of $v$. Thus, in a neighborhood-regular hypergraph, all nodes have their $d$-uniform neighborhoods of the same cardinality for all $d \in \mathbb{N}$.

**Representing Higher Order Structures as Tensors** : There are many data stuctures one can define on a higher order structure like a hypergraph. An $n$-order tensor Maron et al. (2018), as a generalization of

an adjacency matrix on graphs can be used to characterize the higher order connectivities. For simplicial complexes, which are hypergraphs where all subsets of a hyperedge are also hyperedges, a Hasse diagram, which is a multipartite graph induced by the poset relation of subset amongst hyperedges, or simplices, differing in exactly one node, is a common data structure Birkhoff (1940). Similarly, the star expansion matrix Agarwal et al. (2006) can be used to characterize hypergraphs up to isomorphism.

In order to define the star expansion matrix, we define the star expansion bipartite graph.

**Definition 2.5** (star expansion bipartite graph)**.** *Given a hypergraph* $\mathcal{H} = (\mathcal{V}, \mathcal{E})$*, the* star expansion bipartite graph $\mathcal{B}_{\mathcal{V},\mathcal{E}}$ *is the bipartite graph with vertices* $\mathcal{V} \bigsqcup \mathcal{E}$ *and edges* $\{(v, e) \in \mathcal{V} \times \mathcal{E} \mid v \in e\}$.

**Definition 2.6.** *The star expansion incidence matrix* $H$ *of a hypergraph* $\mathcal{H} = (\mathcal{V}, \mathcal{E})$ *is the* $|\mathcal{V}| \times 2^{|\mathcal{V}|}$ *0-1 incidence matrix* $H$ *where* $H_{v,e} = 1$ *iff* $v \in e$ *for* $(v, e) \in \mathcal{V} \times \mathcal{E}$ *for some fixed orderings on both* $\mathcal{V}$ *and* $2^{\mathcal{V}}$.

In practice, as data to machine learning algorithms, the matrix $H$ is sparsely represented by its nonzero entries.

To study the symmetries of a given hypergraph $\mathcal{H} = (\mathcal{V}, \mathcal{E})$, we consider the permutation group on the vertices $\mathcal{V}$, denoted as $Sym(\mathcal{V})$, which acts jointly on the rows and columns of star expansion adjacency matrices. We assume the rows and columns of a star expansion adjacency matrix have some canonical ordering, say lexicographic ordering, given by some prefixed ordering of the vertices. Therefore, each hypergraph $\mathcal{H}$ has a unique canonical matrix representation $H$.

We define the action of a permutation $\pi \in Sym(\mathcal{V})$ on a star expansion adjacency matrix $H$:

$$(\pi \cdot H)_{v,e=(u_1,\ldots,v,\ldots,u_k)} \triangleq H_{\pi^{-1}(v),\pi^{-1}(e)=(\pi^{-1}(u_1),\ldots,\pi^{-1}(v),\ldots,\pi^{-1}(u_k))} \tag{2}$$

Based on the group action, consider the stabilizer subgroup of $Sym(\mathcal{V})$ on an incidence matrix $H$:

$$Stab_{Sym(\mathcal{V})}(H) = \{\pi \in Sym(\mathcal{V}) \mid \pi \cdot H = H\} \tag{3}$$

For simplicity we omit the lower index of $Sym(\mathcal{V})$ when the permutation group is clear from context. It can be checked that $Stab(H) \subseteq Sym(\mathcal{V})$ is a subgroup. Intuitively, $Stab(H)$ consists of all permutations that fix $H$. These are equivalent to hypergraph automorphisms on the original hypergraph $\mathcal{H}$.

**Proposition 2.1.** $Aut(\mathcal{H}) \cong Stab(H)$ *are equivalent as isomorphic groups.*

We can also define a notion of isomorphism between $k$-node sets using the stabilizers on $H$.

**Definition 2.7.** *For a given hypergraph* $\mathcal{H}$ *with star expansion matrix* $H$*, two* $k$*-node sets* $S, T \subseteq \mathcal{V}$ *are called* isomorphic*, denoted as* $S \simeq T$*, if* $\exists \pi \in Stab(H), \pi(S) = T$.

Such isomorphism is an equivalence relation on $k$-node sets. When $k = 1$, we have isomorphic nodes, denoted $u \cong_{\mathcal{H}} v$ for $u, v \in \mathcal{V}$. Node isomorphism is also studied as the so-called structural equivalence in Lorrain & White (1971). Furthermore, when $S \simeq T$ we can then say that there is a matching due to the graph of the $\pi$ map of the form $\{(s, \pi(s)) : s \in S\}$. This matching is between the node sets $S$ and $T$ so that matched nodes are isomorphic.

## 2.3 Invariance and Expressivity

For a given hypergraph $\mathcal{H} = (\mathcal{V}, \mathcal{E})$, we want to do hyperedge prediction on $\mathcal{H}$, which is to predict missing hyperedges from $k$-node sets for $k \geq 2$. Let $|\mathcal{V}| = n$, $|\mathcal{E}| = m$, and $H \in \mathbb{Z}_2^{n \times 2^n}$ be the star expansion adjacency matrix of $\mathcal{H}$. To do hyperedge prediction, we study $k$-node representations $g : \binom{\mathcal{V}}{k} \times \mathbb{Z}_2^{n \times 2^n} \to \mathbb{R}^d$ that map $k$-node sets of hypergraphs to $d$-dimensional Euclidean space. Ideally, we want a most-expressive $k$-node representation for hyperedge prediction, which is intuitively a $k$-node representation that is injective on $k$-node set isomorphism classes from $\mathcal{H}$. We break up the definition of most-expressive $k$-node representation into possessing two properties, as follows:

**Definition 2.8.** *Let* $g : \binom{\mathcal{V}}{k} \times \mathbb{Z}_2^{n \times 2^n} \to \mathbb{R}^d$ *be a* $k$*-node representation on a hypergraph* $\mathcal{H}$*. Let* $H \in \mathbb{Z}_2^{n \times 2^n}$ *be the star expansion adjacency matrix of* $\mathcal{H}$ *for* $n$ *nodes. The representation* $g$ *is* $k$*-node most expressive if* $\forall S, S' \subseteq \mathcal{V}, |S| = |S'| = k$*, the following two conditions are satisfied:*

1. *g is **k-node invariant**:* $\exists \pi \in Stab(H), \pi(S) = S' \implies g(S, H) = g(S', H)$

2. *g is **k-node expressive*** $\nexists \pi \in Stab(H), \pi(S) = S' \implies g(S, H) \neq g(S', H)$

The first condition of a most expressive $k$-node representation states that the representation must be well defined on the $k$ nodes up to isomorphism. The second condition requires the injectivity of our representation. These two conditions mean that the representation does not lose any information when doing prediction for missing $k$-sized hyperedges on a set of $k$ nodes.

We can also define the symmetry group of a $k$-node representation map $g$ on $H$ as the set of all permutations on $\mathcal{V}$ that make the representation map $g$ $k$-node invariant. This is formally defined below:

**Definition 2.9.** *For* $g : \binom{\mathcal{V}}{k} \times \mathbb{Z}_2^{n \times 2^n} \to \mathbb{R}^d$ *a* $k$-node representation on a hypergraph $\mathcal{H}$,

$$Sym(g(H)) \triangleq \{\pi \in Sym(\mathcal{V}) : \forall S, S' \in \binom{\mathcal{V}}{k}, \pi(S) = S' \Rightarrow g(S, H) = g(S', H)\} \tag{4}$$

### 2.4 Generalized Weisfeiler-Lehman-1

We describe a generalized Weisfeiler-Lehman-1 (GWL-1) hypergraph isomorphism test similar to Huang & Yang (2021); Feng et al. (2023) based on the WL-1 algorithm for graph isomorphism testing. There have been many parameterized variants of the GWL-1 algorithm implemented as neural networks, see Section 3.

Let $H$ be the star expansion matrix for a hypergraph $\mathcal{H}$. We define the GWL-1 algorithm as the following two step procedure on $H$ at iteration number $i \geq 0$.

$$f_e^0 \leftarrow \{\}, h_v^0 \leftarrow \{\}$$
$$f_e^{i+1} \leftarrow \{\!\!\{(f_e^i, h_v^i)\}\!\!\}_{v \in e}, \forall e \in \mathcal{E}_{\mathcal{H}}(H) \tag{5}$$
$$h_v^{i+1} \leftarrow \{\!\!\{(h_v^i, f_e^{i+1})\}\!\!\}_{v \in e}, \forall v \in \mathcal{V}_{\mathcal{H}}(H)$$

This is slightly different from the algorithm presented in Huang & Yang (2021) at the $f_e^{i+1}$ update step. Our update step involves an edge representation $f_e^i$, which is not present in their version. Thus our version of GWL-1 is more expressive than that in Huang & Yang (2021). However, they both possess some of the same issues that we identify. We denote $f_e^i(H)$ and $h_v^i(H)$ as the hyperedge and node $i$th iteration GWL-1, called $i$-GWL-1, values on an unattributed hypergraph $\mathcal{H}$ with star expansion $H$. If GWL-1 is run to convergence then we omit the iteration number $i$. We also mean this when we say $i = \infty$.

For a hypergraph $\mathcal{H}$ with star expansion matrix $H$, GWL-1 is strictly more expressive than WL-1 on $A = H \cdot D_e^{-1} \cdot H^T$ with $D_e = diag(H^T \cdot \mathbf{1}_n)$, the node to node adjacency matrix, also called the clique expansion of $\mathcal{H}$. This follows since a triangle with its 3-cycle boundary: $T$ and a 3-cycle $C_3$ have exactly the same clique expansions. Thus WL-1 will give the same node values for both $T$ and $C_3$. GWL-1 on the star expansions $H_T$ and $H_{C_3}$, on the other hand, will identify the triangle as different from its bounding edges.

Let $f^i(H) \triangleq [f_{e_1}^i(H), \cdots, f_{e_m}^i(H)]$ and $h^i(H) \triangleq [h_{v_1}^i(H), \cdots, h_{v_n}^i(H)]$ be two vectors whose entries are ordered by the column and row order of $H$, respectively.

**Proposition 2.2.** *The update steps* $f^i(H)$ *and* $h^i(H)$ *of GWL-1 are permutation equivariant; For any* $\pi \in Sym(\mathcal{V})$, *let:* $\pi \cdot f^i(H) \triangleq [f_{\pi^{-1}(e_1)}^i(H), \cdots, f_{\pi^{-1}(e_m)}^i(H)]$ *and* $\pi \cdot h^i(H) \triangleq [h_{\pi^{-1}(v_1)}^i(H), \cdots, h_{\pi^{-1}(v_n)}^i(H)]$:

$$\forall i \in \mathbb{N}, \pi \cdot f^i(H) = f^i(\pi \cdot H), \ \pi \cdot h^i(H) = h^i(\pi \cdot H) \tag{6}$$

Define the operator $AGG$ as a $k$-set map to representation space $\mathbb{R}^d$. Define the following representation of a $k$-node subset $S \subseteq \mathcal{V}$ of hypergraph $\mathcal{H}$ with star expansion matrix $H$:

$$h^i(S, H) \triangleq AGG[\{h_v^i(H)\}_{v \in S}] \tag{7}$$

where $h_v^i(H)$ is the node value of $i$-GWL-1 on $H$ for node $v$. The representation $h(S, H)$ preserves hyperedge isomorphism classes as shown below:

**Proposition 2.3.** *Let $h^i(S, H) = AGG[\{h_v^i(H)\}_{v \in S}]$ with an injective AGG map. The representation $h^i(S, H)$ is k-node invariant but not necessarily k-node expressive for S a set of k nodes.*

It follows that we can guarantee a *k*-node invariant representation by using GWL-1. For deep learning, we parameterize *AGG* as a universal set learner.

The node representations $h_v^i(H)$ are also parameterized and rewritten into a message passing hypergraph neural network with matrix equations Huang & Yang (2021). For example, the HNHN Dong et al. (2020) hyperGNN architecture can be viewed as a parameterization of GWL-1:

$$X_E^{(l)} = \sigma(H^T X_V^{(l)} W_E^{(l)} + b_E^{(l)})$$
$$X_V^{(l+1)} = \sigma(H X_E^{(l)} W_V^{(l)} + b_V^{(l)}) \tag{8}$$

where $\sigma$ is a nonlinearity, $X_V, X_E$ are vector representations of $h^i(H)$ and $f^i(H)$ respectively, and $W_E, W_V, b_E, b_V$ are learnable weight matrices. Setting $b_E^{(l)} = 0, b_V^{(l)} = 0$, we maintain permutation equivariance as in Proposition 2.2. Furthermore, the HNHN equations of Equation 8 become in direct analogy to the steps of GWL-1 from Equation 5.

## 3  Related Work and Existing Issues

There are many hyperlink prediction methods. Most message passing based methods for hypergraphs are based on the GWL-1 algorithm. These include Huang & Yang (2021); Yadati et al. (2019); Feng et al. (2019); Gao et al. (2022); Dong et al. (2020); Srinivasan et al. (2021); Chien et al. (2022); Zhang et al. (2018). Examples of message passing based approaches that incorporate positional encodings on hypergraphs include SNALS Wan et al. (2021). The paper Zhang et al. (2019) uses a pair-wise node attention mechanism to do higher order link prediction. For a survey on hyperlink prediction, see Chen & Liu (2022).

Various methods have been proposed to improve the expressive power of GNNs due to symmetries in graphs. In Papp & Wattenhofer (2022), substructure labeling is formally analyzed. One of the methods analyzed includes labeling fixed radius ego-graphs as in You et al. (2021); Zhang & Li (2021). Other methods include appending random node features Sato et al. (2021), labeling breadth-first and depth-first search trees Li et al. (2023b) and encoding substructures Zeng et al. (2023); Wijesinghe & Wang (2021). All of the previously mentioned methods depend on a fixed subgraph radius size. This prevents capturing symmetries that span long ranges across the graph. Zhang et al. (2023) proposes to add metric information of each node relative to all other nodes to improve WL-1. This would be very computationally expensive on hypergraphs.

Cycles are a common symmetric substructure. There are many methods that identify this symmetry. Cy2C Choi et al. is a method that encodes cycles to cliques. It has the issue that if the the cycle-basis algorithm is not permutation invariant, isomorphic graphs could get different cycle bases and thus get encoded by Cy2C differently, violating the invariance of WL-1. Similarly, the CW Network Bodnar et al. (2021) is a method that attaches cells to cycles to improve upon the distinguishing power of WL-1 for graph classification. However, inflating the input topology with cells as in Bodnar et al. (2021) would not work for link predicting since it will shift the hyperedge distribution to become much denser. Other works include cell attention networks Giusti et al. (2022) and cycle basis based methods Zhang et al. (2022). For more related work, see the Appendix.

Data augmentation is a commonly used approach to improve robustness to distribution shifts Yao et al. (2022b), recognize symmetries Chen et al. (2020b), and handle data imbalance Chawla et al. (2002). In the graph domain, a priori knowledge of the data distribution can inform rule-based data augmentations. In molecular classification, prior knowledge of the physical meaning of the data can be used to augment graphs Sun et al. (2021). Data augmentations can also be learned through data generation methods. For example a link prediction neural network GAE Kipf & Welling (2016b) can be used to propose edges to to improve node classification Zhao et al. (2021). For a survey on graph data augmentation, see Zhao et al. (2022).

## 4    A Characterization of GWL-1

A hypergraph can be represented by a bipartite graph $\mathcal{B}_{\mathcal{V},\mathcal{E}}$ from $\mathcal{V}$ to $\mathcal{E}$ where there is an edge $(v, e)$ in the bipartite graph iff node $v$ is incident to hyperedge $e$. This bipartite graph is called the star expansion bipartite graph.

We introduce a more structured version of graph isomorphism called a 2-color isomorphism to characterize hypergraphs. It is a map on 2-colored graphs, which are graphs that can be colored with two colors so that no two nodes in any graph with the same color are connected by an edge. We define a 2-colored isomorphism formally here:

**Definition 4.1.** *A 2-colored isomorphism is a graph isomorphism on two 2-colored graphs that preserves node colors. It is denoted by $\cong_c$.*

A 2-colored isomorphism from a graph $G$ to itself is called a 2-colored automorphism. The set of all 2-colored automorphisms on $G$ is denoted $Aut_c(G)$.

A bipartite graph always has a 2-coloring. In this paper, we canonically fix a 2-coloring on all star expansion bipartite graphs by assigning red to all the nodes in the node partition and and blue to all the nodes in the hyperedge partition. See Figure 2(a) as an example. We let $\mathcal{B}_{\mathcal{V}}, \mathcal{B}_{\mathcal{E}}$ be the red and blue colored nodes in $\mathcal{B}_{\mathcal{V},\mathcal{E}}$ respectively.

**Proposition 4.1.** *We have two hypergraphs $(\mathcal{V}_1, \mathcal{E}_1) \cong (\mathcal{V}_2, \mathcal{E}_2)$ iff $\mathcal{B}_{\mathcal{V}_1,\mathcal{E}_1} \cong_c \mathcal{B}_{\mathcal{V}_2,\mathcal{E}_2}$ where $\mathcal{B}_{\mathcal{V}_i,\mathcal{E}_i}$ is the star expansion bipartite graph of $(\mathcal{V}_i, \mathcal{E}_i)$*

We define a topological object for a graph originally from algebraic topology called a universal cover:

**Definition 4.2** (Hatcher (2005))**.** *The* universal covering *of a connected graph $G$ is a (potentially infinite) graph $\tilde{G}$ together with a map $p_G : \tilde{G} \to G$ such that:*

1. *$\forall x \in \mathcal{V}(\tilde{G})$, $p_G|_{N(x)}$ is an isomorphism onto $N(p_G(x))$.*

2. *$\tilde{G}$ is simply connected (a tree)*

We call such $p_G$ the *universal covering map* and $\tilde{G}$ the *universal cover* of $G$. A covering graph is a graph that satisfies property 1 but not necessarily 2 in Definition 4.2. The universal covering $\tilde{G}$ is essentially unique Hatcher (2005) in the sense that it can cover all connected covering graphs of $G$. Furthermore, define a rooted isomorphism $G_x \cong H_y$ as an isomorphism between graphs $G$ and $H$ that maps $x$ to $y$ and vice versa. Let $\tilde{G}_{\tilde{x}}^i$ denote the rooted universal cover $\tilde{G}_{\tilde{x}}$ with every leaf of depth (number of edges) exactly $i \in \mathbb{Z}^+$. It is a known result that:

**Theorem 4.2.** *[Krebs & Verbitsky (2015)] Let $G$ and $H$ be two connected graphs. Let $p_G : \tilde{G} \to G, p_H : \tilde{H} \to H$ be the universal covering maps of $G$ and $H$ respectively. For any $i \in \mathbb{N}$, for any two nodes $x \in G$ and $y \in H$: $\tilde{G}_{\tilde{x}}^i \cong \tilde{G}_{\tilde{y}}^i$ iff the WL-1 algorithm assigns the same value to nodes $x = p_G(\tilde{x})$ and $y = p_H(\tilde{y})$.*

We generalize the second result stated above about a topological characterization of WL-1 for GWL-1 for hypergraphs. In order to do this, we need to generalize the definition of a universal covering to suite the requirements of a bipartite star expansion graph. To do this, we lift $\mathcal{B}_{\mathcal{V},\mathcal{E}}$ to a 2-colored tree universal cover $\tilde{\mathcal{B}}_{\mathcal{V},\mathcal{E}}$ where the red/blue nodes of $\mathcal{B}_{\mathcal{V},\mathcal{E}}$ are lifted to red/blue nodes in $\tilde{\mathcal{B}}_{\mathcal{V},\mathcal{E}}$. Furthermore, the labels {} are placed on the blue nodes corresponding to the hyperedges in the lift and the labels {} are placed on all its corresponding red nodes in the lift. Let $(\tilde{\mathcal{B}}_{\mathcal{V},\mathcal{E}}^k)_{\tilde{x}}$ denote the $k$-hop rooted 2-colored subtree $\tilde{\mathcal{B}}_{\mathcal{V},\mathcal{E}}$ with root $\tilde{x}$ and $p_{\mathcal{B}_{\mathcal{V},\mathcal{E}}}(\tilde{x}) = x$ for any $x \in \mathcal{V}(\mathcal{B}_{\mathcal{V},\mathcal{E}})$.

**Theorem 4.3.** *Let $\mathcal{H}_1 = (\mathcal{V}_1, \mathcal{E}_1)$ and $\mathcal{H}_2 = (\mathcal{V}_2, \mathcal{E}_2)$ be two connected hypergraphs. Let $\mathcal{B}_{\mathcal{V}_1,\mathcal{E}_1}$ and $\mathcal{B}_{\mathcal{V}_2,\mathcal{E}_2}$ be two canonically colored bipartite graphs for $\mathcal{H}_1$ and $\mathcal{H}_2$ (vertices colored red and hyperedges colored blue). Let $p_{\mathcal{B}_{\mathcal{V}_1,\mathcal{E}_1}} : \tilde{\mathcal{B}}_{\mathcal{V}_1,\mathcal{E}_1} \to \mathcal{B}_{\mathcal{V}_1,\mathcal{E}_1}, p_{\mathcal{B}_{\mathcal{V}_2,\mathcal{E}_2}} : \tilde{\mathcal{B}}_{\mathcal{V}_2,\mathcal{E}_2} \to \mathcal{B}_{\mathcal{V}_2,\mathcal{E}_2}$ be the universal coverings of $\mathcal{B}_{\mathcal{V}_1,\mathcal{E}_1}$ and $\mathcal{B}_{\mathcal{V}_2,\mathcal{E}_2}$ respectively. For any $i \in \mathbb{Z}^+$, for any of the nodes $x_1 \in \mathcal{B}_{\mathcal{V}_1}, e_1 \in \mathcal{B}_{\mathcal{E}_1}$ and $x_2 \in \mathcal{B}_{\mathcal{V}_2}, e_2 \in \mathcal{B}_{\mathcal{E}_2}$:*

- *$(\tilde{\mathcal{B}}_{\mathcal{V}_1,\mathcal{E}_1}^{2i-1})_{\tilde{e}_1} \cong_c (\tilde{\mathcal{B}}_{\mathcal{V}_2,\mathcal{E}_2}^{2i-1})_{\tilde{e}_2}$ iff $f_{e_1}^i = f_{e_2}^i$*

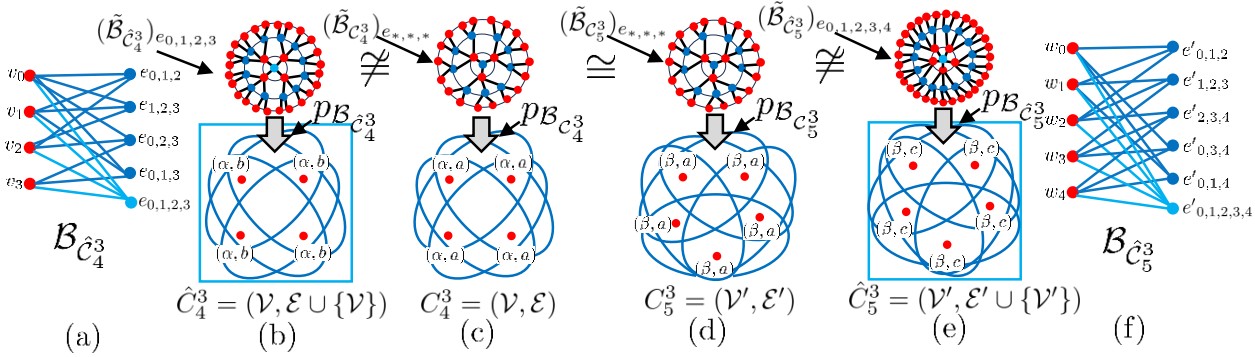

Figure 2: An illustration of hypergraph symmetry breaking. (c,d) 3-regular hypergraphs $C_4^3, C_5^3$ with 4 and 5 nodes respectively and their corresponding universal covers centered at any hyperedge $(\tilde{\mathcal{B}}_{C_4^3})_{e_*,*,*}, (\tilde{\mathcal{B}}_{C_5^3})_{e_*,*,*}$ with universal covering maps $p_{\mathcal{B}_{C_4^3}}, p_{\mathcal{B}_{C_5^3}}$. (b,e) the hypergraphs $\hat{C}_4^3, \hat{C}_5^3$, which are $C_4^3, C_5^3$ with $4, 5$-sized hyperedges attached to them and their corresponding universal covers and universal covering maps. (a,f) are the corresponding bipartite graphs of $\hat{C}_4^3, \hat{C}_5^3$. (c,d) are indistinguishable by GWL-1 and thus will give identical node values by Theorem 4.3. On the other hand, (b,e) gives node values which are now sensitive to the the order of the hypergraphs $4, 5$, also by Theorem 4.3.

- $(\tilde{\mathcal{B}}_{\mathcal{V}_1,\mathcal{E}_1}^{2i})_{\tilde{x}_1} \cong_c (\tilde{\mathcal{B}}_{\mathcal{V}_2,\mathcal{E}_2}^{2i})_{\tilde{x}_2}$ iff $h_{x_1}^i = h_{x_2}^i$,

with $f_\bullet^i, h_\bullet^i$ the ith GWL-1 values for the hyperedges and nodes respectively where $e_1 = p_{\mathcal{B}_{\mathcal{V}_1,\mathcal{E}_1}}(\tilde{e}_1)$, $x_1 = p_{\mathcal{B}_{\mathcal{V}_1,\mathcal{E}_1}}(\tilde{x}_1)$, $e_2 = p_{\mathcal{B}_{\mathcal{V}_2,\mathcal{E}_2}}(\tilde{e}_2)$, $x_2 = p_{\mathcal{B}_{\mathcal{V}_2,\mathcal{E}_2}}(\tilde{x}_2)$.

Theorem 4.3 states that a 2-colored isomorphism is maintained during each step of the GWL-1 algorithm. Thus we can view the GWL-1 algorithm on a hypergraph as equivalent to computing a universal cover of the star expansion bipartite graph up to a 2-colored isomorphism. We can thus deduce from Theorems 4.3, 4.2 that GWL-1 reduces to computing WL-1 on the bipartite graph up to the 2-colored isomorphism.

**Corollary 1.** *Let $\mathcal{H}_1 = (\mathcal{V}_1,\mathcal{E}_1)$ and $\mathcal{H}_2 = (\mathcal{V}_2,\mathcal{E}_2)$ be two connected hypergraphs. Let $\mathcal{B}_{\mathcal{V}_1,\mathcal{E}_1}$ and $\mathcal{B}_{\mathcal{V}_2,\mathcal{E}_2}$ be two canonically colored bipartite graphs for $\mathcal{H}_1$ and $\mathcal{H}_2$ (vertices colored red and hyperedges colored blue). Let $p_{\mathcal{B}_{\mathcal{V}_1,\mathcal{E}_1}} : \tilde{\mathcal{B}}_{\mathcal{V}_1,\mathcal{E}_1} \to \mathcal{B}_{\mathcal{V}_1,\mathcal{E}_1}, p_{\mathcal{B}_{\mathcal{V}_2,\mathcal{E}_2}} : \tilde{\mathcal{B}}_{\mathcal{V}_2,\mathcal{E}_2} \to \mathcal{B}_{\mathcal{V}_2,\mathcal{E}_2}$ be the universal coverings of $\mathcal{B}_{\mathcal{V}_1,\mathcal{E}_1}$ and $\mathcal{B}_{\mathcal{V}_2,\mathcal{E}_2}$ respectively. For any $i \in \mathbb{Z}^+$,*

- $(\tilde{\mathcal{B}}_{\mathcal{V}_1,\mathcal{E}_1}^{2i-1})_{\tilde{v}_1} \cong_c (\tilde{\mathcal{B}}_{\mathcal{V}_2,\mathcal{E}_2}^{2i-1})_{\tilde{v}_2}$ iff $g_{v_1}^i = g_{v_2}^i$

with $g_{v_1}^i, g_{v_2}^i$ the i-th WL-1 values for $v_1, v_2 \in \mathcal{V}(\mathcal{B}_{\mathcal{V}_1,\mathcal{E}_1}), \mathcal{V}(\mathcal{B}_{\mathcal{V}_2,\mathcal{E}_2})$ respectively where $v_1 = p_{\mathcal{B}_{\mathcal{V}_1,\mathcal{E}_1}}(\tilde{v}_1)$, $v_2 = p_{\mathcal{B}_{\mathcal{V}_2,\mathcal{E}_2}}(\tilde{v}_2)$.

See Figure 2 for an illustration of the universal covering of the corresponding bipartite graphs for two 3-uniform neighborhood regular hypergraphs.

## 4.1  A Limitation of GWL-1

Due to the equivalence of GWL-1 to viewing the hypergraph $\mathcal{H}$ as a collection of rooted trees, we can show that the automorphism group of $\mathcal{H}$ is a subgroup of the automorphism of this collection of rooted trees. This is stated in the following proposition:

**Theorem 4.4.** *Let $h^L : [\mathcal{V}]^1 \times \mathbb{Z}_2^{n \times 2^n} \to \mathbb{R}^d$ be the L-GWL-1 representation of nodes for hypergraph $\mathcal{H}$ in Equation 7, then*

$$Aut(\mathcal{H}) \cong Stab(H) \subseteq Sym(h^L(H)) \cong Aut_c(\tilde{\mathcal{B}}_{\mathcal{V},\mathcal{E}}^{2L}), \forall L \geq 1 \qquad (9)$$

By Proposition 2.1, $Aut(\mathcal{H}) \cong Stab(H)$. The subgroup relationship follows by definition of the symmetry group of a representation map given in Definition 2.9 and the equivariance of $L$-GWL-1 due to Proposition

2.2. The last group isomorphism follows by the equivalence between $L$-GWL-1 and the universal cover of the star expansion bipartite graph $\mathcal{B}_{\mathcal{V},\mathcal{E}}$ up to $2L$-hops.

Since there are more automorphisms over the GWL-1 view of the hypergraph, many false positive symmetries might exist. Consider the following example. For two neighborhood-regular hypergraphs $C_1$ and $C_2$, the red/blue colored universal covers $\tilde{B}_{C_1}, \tilde{B}_{C_2}$ of the star expansions of $C_1$ and $C_2$ are isomorphic, with the same GWL-1 values on all nodes. However, two neighborhood-regular hypergraphs of different order become distinguishable if a single hyperedge covering all the nodes of each neighborhood-regular hypergraph is added. Furthermore, deleting the original hyperedges, does not change the node isomorphism classes of each hypergraph. Referring to Figure 2, consider the hypergraph $\mathcal{C} = C_4^3 \sqcup C_5^3$, the hypergraph with two 3-regular hypergraphs $C_4^3$ and $C_5^3$ acting as two connected components of $\mathcal{C}$. As shown in Figure 2, the node representations of the two hypergraphs are identical due to Theorem 4.3.

Given a hypergraph $\mathcal{H}$, we define a special induced subhypergraph $\mathcal{R} \subseteq \mathcal{H}$ whose node set GWL-1 cannot distinguish from other such special induced subhypergraphs.

**Definition 4.3.** *A $L$-GWL-1 symmetric induced subhypergraph $\mathcal{R} \subset \mathcal{H}$ of $\mathcal{H}$ is a connected induced subhypergraph determined by $\mathcal{V}_{\mathcal{R}} \subseteq \mathcal{V}_{\mathcal{H}}$, some subset of nodes that are all indistinguishable amongst each other by $L$-GWL-1:*

$$h_u^L(H) = h_v^L(H), \forall u, v \in \mathcal{V}_{\mathcal{R}} \tag{10}$$

*When $L = \infty$, we call such $\mathcal{R}$ a GWL-1 symmetric induced subhypergraph. Furthermore, if $\mathcal{R} = \mathcal{H}$, then we say $\mathcal{H}$ is GWL-1 symmetric.*

This definition is similar to that of a symmetric graph from graph theory Godsil & Royle (2001), except that isomorphic nodes are determined by the GWL-1 approximator instead of an automorphism. The following observation follows from the definitions.

**Observation 1.** *A hypergraph $\mathcal{H}$ is GWL-1 symmetric if and only if it is $L$-GWL-1 symmetric for all $L \geq 1$ if and only if $\mathcal{H}$ is neighborhood regular.*

Our goal is to find GWL-1 symmetric induced subhypergraphs in a given hypergraph and break their symmetry without affecting any other nodes.

## 5 Method

Our goal is to learn from a training hypergraph and then predict higher order links in a temporally later hypergraph transductively. This can be formulated as follows:

**Problem 1.** *Hyperlink Transductive Learning:* *Let $\mathcal{V}$ be $n$ nodes.*

1. *Given a training hypergraph sampled at time $t_{tr} \in \mathbb{R}$:*

   *For $(\mathcal{V}, (\mathcal{E}_{tr})_{gt}) \sim P(\mathcal{H}; t_{tr})$, learn a boolean predictor $\hat{h}$ on hypergraph input so that:*

2. *For a testing hypergraph sampled at time $t_{te} \in \mathbb{R}, t_{te} > t_{tr}$ :*

   *For $(\mathcal{V}, \mathcal{E}_{te}) \sim P(\mathcal{H}; t_{te})$ and $\mathcal{E}_{te} \subseteq (\mathcal{E}_{te})_{gt}$,*

   *$\hat{h}((\mathcal{V}, \mathcal{E}_{te}), e)$ predicts whether $e \in (\mathcal{E}_{te})_{gt} \setminus \mathcal{E}_{te}, \forall e \in 2^{\mathcal{V}}$*

We will assume that the unobservable hyperedges are of the same size $k$ so that we only need to predict on $k$-node sets. In order to preserve the most information while still respecting topological structure, we aim to start with an invariant multi-node representation to predict hyperedges and increase its expressiveness, as defined in Definition 2.8. For input hypergraph $\mathcal{H}$ and its matrix representation $H$, to do the prediction of a missing hyperedge on node subsets, we use a multi-node representation $h(S, H)$ for $S \subseteq \mathcal{V}(H)$ as in Equation 7 due to its simplicity, guaranteed invariance, and improve its expressivity. We aim to not affect the computational complexity since message passing on hypergraphs is already quite expensive, especially on GPU memory.

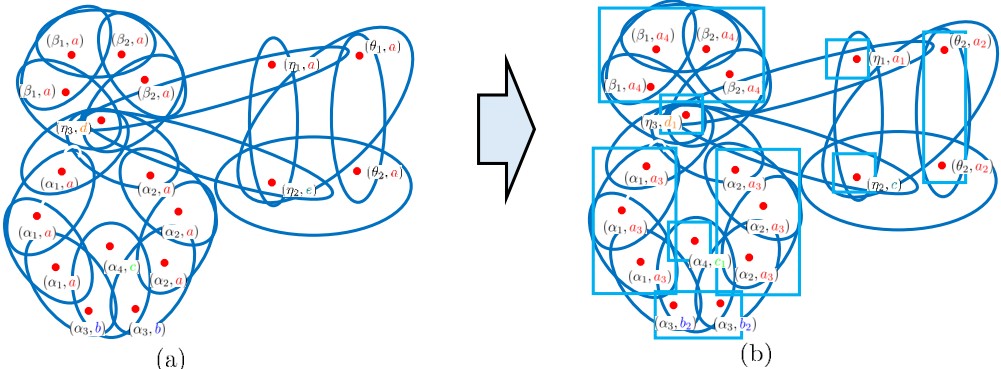

(a)                                        (b)

Figure 3: An illustration of Algorithm 1 for 1-GWL-1.
In (a) a hypergraph is shown. Each node is labeled with a pair. The left part of the pair in Greek alphabet is its isomorphism class. The right part of the pair in Latin alphabet is its 1-GWL-1 class, which is determined by its neighborhood of hyperedges. In (b) the multi-hypergraph formed by covering the original hypergraph by hyperedges (light blue boxes) which are determined by the connected components of 1-GWL-1 indistinguishable node sets. The nodes can now be relabeled by 1-GWL-1. All hyperedges can be assigned learnable weights. For downstream training, the new hyperedges are randomly added and the existing hyperedges within each new hyperedge are randomly dropped.

Our method is a preprocessing algorithm that operates on an input training hypergraph $\mathcal{H} = (\mathcal{V}, \mathcal{E}_{tr})$ with $\mathcal{E}_{tr} \subset (\mathcal{E}_{tr})_{gt}$. In order to increase expressivity, we search for potentially indistinguishable regular induced subhypergraphs so that they can be replaced with hyperedges that span the subhypergraph to break the symmetries that prevent GWL-1 from being more expressive. We devise an algorithm, which is shown in Algorithm 1. It takes as input a hypergraph $\mathcal{H}$ with star expansion matrix $H$. The idea of the algorithm is to identify nodes of the same GWL-1 value that are maximally connected and use this collection of node subsets to break the symmetry of $\mathcal{H}$.

First we introduce some combinatorial definitions for hypergraph data that we will use in our algorithm:

**Definition 5.1.** *A hypergraph $\mathcal{H} = (\mathcal{V}, \mathcal{E})$ is **connected** if $\mathcal{B}_{\mathcal{V},\mathcal{E}}$ is a connected graph.*

*A **connected component** of $\mathcal{H}$ is a connected induced subhypergraph which is not properly contained in any connected subhypergraph of $\mathcal{H}$.*

Our preprocessing algorithm is explicitly given in Algorithm 1. We describe it in words here:

**Algorithm:**

1. For a given $L \in \mathbb{Z}^+$ and any $L$-GWL-1 node value $c_L$, the first step is to construct the induced subhypergraph $\mathcal{H}_{c_L}$ from the $L$-GWL-1 class of nodes:

$$\mathcal{V}_{c_L} \triangleq \{v \in \mathcal{V} : c_L = h_v^L(H)\}, \tag{11}$$

   where $h_v^L$ denotes the $L$-GWL-1 class of node $v$.

2. We then compute the connected components of $\mathcal{H}_{c_L}$. Denote $\mathcal{C}_{c_L}$ as the set of all connected components of $\mathcal{H}_{c_L}$. If $L = \infty$, then drop $L$. Each of these connected components is a subhypergraph of $\mathcal{H}$, denoted $\mathcal{R}_{c_L,i}$ where $\mathcal{R}_{c_L,i} \subseteq \mathcal{H}_{c_L} \subseteq \mathcal{H}$ for $i = 1, ..., |\mathcal{C}_{c_L}|$.

3. Gather the collection of node sets and hyperedge sets formed by $\mathcal{R}_{c_L,i}, i = 1, ..., |\mathcal{C}_{c_L}|$.

We call the subhypergraphs in $\mathcal{C}_{c_L}$ found by the symmetry finding Algorithm 1 as **maximally connected $L$-GWL-1 equal valued subhypergraphs**.

---

**Algorithm 1:** A Symmetry Finding Algorithm

---

**Data:** Hypergraph $\mathcal{H} = (\mathcal{V}, \mathcal{E})$, represented by its star expansion matrix $H$. $L \in \mathbb{Z}^+$ is the number of iterations to run GWL-1.

**Result:** A pair of collections: $(\mathcal{R}_V = \{\mathcal{V}_{R_j}\}, \mathcal{R}_E = \cup_j \{\mathcal{E}_{R_j}\})$ where $R_j$ are disconnected subhypergraphs exhibiting symmetry in $\mathcal{H}$ that are indistinguishable by $L$-GWL-1.

**1** $U_L \leftarrow h_v^L(H); \mathcal{G}_L \leftarrow \{U_L[v] : \forall v \in \mathcal{V}\}$ ;     /* $U_L[v]$ is the $L$-GWL-1 value of node $v \in \mathcal{V}$. */

**2** $\mathcal{B}_{\mathcal{V}_{\mathcal{H}}, \mathcal{E}_{\mathcal{H}}} \leftarrow Bipartite(\mathcal{H})$ /* Construct the bipartite graph from $\mathcal{H}$. */

**3** $\mathcal{R}_V \leftarrow \{\}; \mathcal{R}_E \leftarrow \{\}$

**4 for** $c_L \in \mathcal{G}_L$ **do**

**5**  $\mathcal{V}_{c_L} \leftarrow \{v \in \mathcal{V} : U_L[v] = c_L\}, \mathcal{E}_{c_L} \leftarrow \{e \in \mathcal{E} : u \in \mathcal{V}_{c_L}, \forall u \in e\}$

**6**  $\mathcal{C}_{c_L} \leftarrow \text{ConnectedComponents}(\mathcal{H}_{c_L} = (\mathcal{V}_{c_L}, \mathcal{E}_{c_L}))$

**7**  **for** $\mathcal{R}_{c_L, i} \in \mathcal{C}_{c_L}$ **do**

**8**    $\mathcal{R}_V \leftarrow \mathcal{R}_V \cup \{\mathcal{V}_{\mathcal{R}_{c_L, i}}\}; \mathcal{R}_E \leftarrow \mathcal{R}_E \cup \mathcal{E}_{\mathcal{R}_{c_L, i}}$

**9**  **end**

**10 end**

**11 return** $(\mathcal{R}_V, \mathcal{R}_E)$

---

We will use the output hyperedges of the preprocessing algorithm to "softly" cover the input hypergraph. This will introduce multiplicities to the hyperedges. We formally define such a generalization of a hypergraph, called a multi-hypergraph here:

**Definition 5.2.** *A **multi**-hypergraph $\mathcal{H} = (\mathcal{V}, \tilde{\mathcal{E}})$ is a pair consisting of a set of nodes $\mathcal{V}$ and a multiset of hyperedges $\tilde{\mathcal{E}} \triangleq (\mathcal{E}, m)$ where $\mathcal{E} \subseteq 2^{\mathcal{V}}$ and $m : \mathcal{E} \to \mathbb{Z}^+$ is a multiplicity function.*

The star expansion incidence matrix of a multi-hypergraph $\mathcal{H}$ can now be defined as:

**Definition 5.3.** *The star expansion incidence matrix $H$ of a multi-hypergraph $\mathcal{H} = (\mathcal{V}, \tilde{\mathcal{E}})$ is the $|\mathcal{V}| \times (2^{|\mathcal{V}|} \times \mathbb{Z}^+)$ (infinite) 0-1 incidence matrix $H$ where $H_{v,e} = 1$ iff $v \in e$ for $(v, e) \in \mathcal{V} \times \tilde{\mathcal{E}}$ for some fixed orderings on both $\mathcal{V}$ and $2^{\mathcal{V}} \times \mathbb{Z}^+$.*

In practice, of course, the multiplicity function of $\mathcal{H}$ is bounded and only the nonzeros of this matrix are kept track of. Thus the star expansion incidence matrix is actually a sparse finite matrix.

On a multi-hypergraph $\mathcal{H} = (\mathcal{V}, \mathcal{E})$, we can define the degree of a vertex $v \in \mathcal{V}$ in terms of the cardinality of a multiset: $deg(v) \triangleq |\{\!\{e : e \ni v\}\!\}| = \sum_{e \ni v} m(e)$ where $m(e)$ is the multiplicity, or number of repeated occurrences, of $e$ in $\{\!\{e : e \ni v\}\!\}$.

Letting $\mathcal{E}_{\mathcal{H}}(H)$ be the multiset of hyperedges of $\mathcal{H}$ and $\mathcal{V}_{\mathcal{H}}(H)$ the set of nodes of $\mathcal{H}$, message passing through incidences is still well defined. We summarize this in the following proposition:

**Proposition 5.1.** *The GWL-1 algorithm of Equation 5 can be computed on a multi-hypergraph $\mathcal{H}$*

*Thus, $h_v^i(H)$ and $h^i(S, H)$ are well defined on its star incidence matrix $H$.*

This implication of Proposition 5.1 is that GWL-1 based hyperGNNs can learn on multi-hypergraphs using the star expansion incidence matrix representations.

**Downstream Training:** After executing Algorithm 1 on $\mathcal{H}$, we collect its output $(\mathcal{R}_V, \mathcal{R}_E)$. During training, for each $i = 1, ..., |\mathcal{C}_{c_L}|$ we randomly perturb $\mathcal{R}_{c_L, i}$ to form a random multi-hypergraph $\hat{\mathcal{H}}_L$ by:

- Attaching (with multiplicity) a single hyperedge that covers $\mathcal{V}_{\mathcal{R}_{c_L, i}}$ with probability $q_i$ and not attaching with probability $1 - q_i$.

- All the hyperedges in $\mathcal{R}_{c_L, i}$ are dropped or kept with probability $p$ and $1 - p$ respectively.

Our method is similar to the concept of adding virtual nodes Hwang et al. (2022) in graph representation learning. This is due to the equivalence between virtual nodes and hyperedges by Proposition 4.1. For a

guarantee of improving expressivity, see Lemma 5.4 and Theorems 5.5, 5.6. For an illustration of the data augmentation, see Figure 2.

Alternatively, downstream training using the output of Algorithm 1 can be done. Similar to subgraph NNs, this is done by applying an ensemble of models Alsentzer et al. (2020); Papp et al. (2021); Tan et al. (2023), with each model trained on transformations of $\mathcal{H}$ with its symmetric subhypergraphs randomly replaced. This, however, is computationally expensive.

**Illustration:** In Figure 3 an illustration of Algorithm 1 is shown. For the hypergraph shown in Figure 3 (a), two graph cycles are glued at a single node while a separate hypergraph is glued at that same node. With 1-GWL-1, there is a large class of nodes labeled "a" that are indistinguishable. Each of the connected components of these 1-GWL-1 node classes is covered by a single hyperedge (light blue box) to form a multi-hypergraph. We show in Section 5.1 that in this multihypergraph, the nodes express more of the original hypergraph. The data augmentation procedure during downstream training can then be applied to the blue boxes and the original hyperedges within each blue box separately.

## 5.1 Algorithm Guarantees

We show some guarantees for the output of Algorithm 1. We will assume the following notation to denote the relevant substructures on a single hypergraph found by Algorithm 1.

**Notation:**

Let $\mathcal{H} = (\mathcal{V}, \mathcal{E})$ be a hypergraph with star expansion matrix $H$ as before.

Let $(\mathcal{R}_{\mathcal{V}}, \mathcal{R}_{\mathcal{E}})$ be the output of Algorithm 1 on $H$ for $L \in \mathbb{Z}^+$. We call this collection of nodes and hyperedges as GWL-1 symmetric induced components. Let:

$$\hat{\mathcal{H}}_L \triangleq (\mathcal{V}, \mathcal{E} \sqcup \mathcal{R}_V) \tag{12}$$

be the multi-hypergraph formed from $\mathcal{H}$ after adding all the hyperedges from $\mathcal{R}_{\mathcal{V}}$ and let $\hat{H}_L$ be the star expansion matrix of the resulting multi-hypergraph $\hat{\mathcal{H}}_L$. Let:

$$V_{c_L, s} \triangleq \{v \in \mathcal{V}_{c_L} : v \in R, R \in \mathcal{C}_{c_L}, |\mathcal{V}_R| = s\} \tag{13}$$

be the set of all nodes of $L$-GWL-1 class $c_L$ belonging to a connected component in $\mathcal{C}_{c_L}$ of $s \geq 1$ nodes in $\mathcal{H}_{c_L}$, the induced subhypergraph of $L$-GWL-1. Let:

$$\mathcal{G}_L \triangleq \{h_v^L(H) : v \in \mathcal{V}\} \tag{14}$$

be the set of all $L$-GWL-1 values on $H$. Let:

$$\mathcal{S}_{c_L} \triangleq \{|\mathcal{V}_{\mathcal{R}_{c_L, i}}| : \mathcal{R}_{c_L, i} \in \mathcal{C}_{c_L}\} \tag{15}$$

be the set of node set sizes of the connected components in $\mathcal{H}_{c_L}$.

**Properties of $(\mathcal{R}_{\mathcal{V}}, \mathcal{R}_{\mathcal{E}})$ from Algorithm 1:**

In the following proposition, we show that the subhypergraphs found by Algorithm 1 are GWL-1 symmetric. This should be evident by line 5 of Algorithm 1 where an induced subgraph of the bipartite graph is formed from nodes of the same $L$-GWL-1 values.

**Proposition 5.2.** *If $L = \infty$, for any GWL-1 node value $c$ computed on $\mathcal{H}$, all connected component subhypergraphs $\mathcal{R}_{c,i} \in \mathcal{C}_c$ are GWL-1 symmetric as hypergraphs.*

Algorithm 1 outputs a partition of the entire hypergraph. This follows from the fact that GWL-1 already covers the entire hypergraph after a single hop.

**Proposition 5.3.** *If $L \geq 1$, the output $(\mathcal{R}_{\mathcal{V}}, \mathcal{R}_{\mathcal{E}})$ of Algorithm 1 partitions a subgraph of $\mathcal{H}$, meaning:*

$$\mathcal{V} = \sqcup_{V \in \mathcal{R}_{\mathcal{V}}} V \text{ and } \mathcal{E} \supset \sqcup_{E \in \mathcal{R}_{\mathcal{E}}} E \tag{16}$$

The output of Algorithm 1 can additionally be used to guarantee improvement in expressing the hypergraph:

**Prediction Guarantees:**

In order to guarantee that the GWL-1 symmetric components $\mathcal{R}_{c,i}$ found by Algorithm 1 carry additional information, there needs to be a separation between them to prevent an intersection between the rooted trees computed by GWL-1. We define what it means for two node subsets to be sufficiently separated via the shortest hyperedge path distance between nodes in $\mathcal{V}$ as follows:

**Definition 5.4.** *Two subsets of nodes* $\mathcal{U}_1, \mathcal{U}_2 \subseteq \mathcal{V}$ *are* ***sufficiently*** $L$***-separated*** *if:*

$$\min_{v_1 \in \mathcal{U}_1, v_2 \in \mathcal{U}_2} d(v_1, v_2) > L \tag{17}$$

*where* $d(v_1, v_2) \triangleq \min_{e_1, \dots, e_k \in \mathcal{E}, v_1 \in e_1, v_2 \in e_k} k$ *is the shortest hyperedge path distance from* $v_1 \in \mathcal{V}$ *to* $v_2 \in \mathcal{V}$.

*A collection of node subsets* $\mathcal{C} \subseteq 2^{\mathcal{V}}$ *is* ***sufficiently*** $L$***-separated*** *if all pairs of node subsets are* ***sufficiently*** $L$***-separated***.

Our definition of sufficiently $L$-separated is similar in nature to that of well separation between point sets Callahan & Kosaraju (1995) in Euclidean space. Assuming that the $\mathcal{C}_{c_L}$ are sufficiently $L$-separated from each other, intuitively meaning that no two nodes from two separate $\mathcal{V}_{\mathcal{R}_{c_L},i} \in \mathcal{R}_V$ are within $L$ hyperedges away, then the cardinality of each component $|\mathcal{V}_{\mathcal{R}_{c_L},i}|$ is recognizable. This is stated in the following lemma:

**Lemma 5.4.** *If* $L \in \mathbb{Z}^+$ *is small enough so that after running Algorithm 1 on* $L$, *for any* $L$*-GWL-1 node class* $c_L$ *on* $\mathcal{V}$ *the collection of* $\mathcal{C}_{c_L}$ *is* ***sufficiently*** $L$***-separated***,

*then after forming* $\hat{\mathcal{H}}_L$, *the new* $L$*-GWL-1 node classes of* $\mathcal{V}_{\mathcal{R}_{c_L},i}$ *for* $i = 1, \dots, \mathcal{C}_{c_L}$ *in* $\hat{\mathcal{H}}_L$ *are all the same class* $c_L'$ *but are distinguishable from* $c_L$ *depending on* $|\mathcal{V}_{\mathcal{R}_{c_L},i}|$.

We can then use this lemma to show that under certain conditions for large hypergraphs, augmenting the hypergraph $\mathcal{H}$ to the multi-hypergraph $\hat{\mathcal{H}}_L$ will give a guarantee on the number of pairs of $k$-node sets that become distinguished which were indistinguishable as sets of GWL-1 values.

**Theorem 5.5.** *Let* $|\mathcal{V}| = n, L \in \mathbb{Z}^+$ *and* $vol(v) \triangleq \sum_{e \in \mathcal{E}: e \ni v} |e|$ *and assuming that the collection of node subsets* $\mathcal{C}_{c_L}$ *is sufficiently* $L$*-separated.*

*If* $vol(v) = O(\log^{\frac{1-\epsilon}{4L}} n), \forall v \in \mathcal{V}$ *for any constant* $\epsilon > 0$; $|\mathcal{S}_{c_L}| \le S, \forall c_L \in \mathcal{C}_L$, $S$ *constant, and* $|V_{c_L,s}| = O(\frac{n^\epsilon}{\log^{\frac{1}{2k}}(n)}), \forall s \in \mathcal{C}_{c_L}$ , *then for* $k \in \mathbb{Z}^+$ *and* $k$*-tuple* $C = (c_{L,1}, \dots, c_{L,k}), c_{L,i} \in \mathcal{G}_L, i = 1..k$ *there exists* $\omega(n^{2k\epsilon})$ *many pairs of* $k$*-node sets* $S_1 \not\simeq S_2$ *such that* $(h_u^L(H))_{u \in S_1} = (h_{v \in S_2}^L(H)) = C$, *as ordered* $k$*-tuples, while* $h(S_1, \hat{H}_L) \ne h(S_2, \hat{H}_L)$ *also by* $L$ *steps of GWL-1.*

The conditions of Theorem 5.5 assume an arbitrarily large hypergraph that is sparse and for every $c_L \in \mathcal{G}_L$ has $\mathcal{C}_{c_L}$ sufficiently $L$-separated, has a bounded set of node set sizes from $\mathcal{S}_{c_L}$ and a controlled growth for each node set size from $\mathcal{S}_{c_L}$. The idea behind the proof of Theorem 5.5 is that under these conditions, there are enough isomorphic rooted trees computed by $L$-GWL-1. It can then be shown that over all pairs of $k$-sets of nodes with elementwise isomorphic rooted trees, that they can be distinguished by the component size they belong in. We give a simple example hypergraph that illustrates the condition of Theorem 5.5.

**Example:** A simple example of a hypergraph that statisfies the conditions of Theorem 5.5 is a union of many disconnected hypergraphs $\mathcal{H} = \cup_i \mathcal{H}_i = (\mathcal{V}, \mathcal{E})$ with $|\mathcal{V}_{\mathcal{H}_i}| \le S$ where $S < \infty$ is a small constant independent of $n = |\mathcal{V}| \ge S$. Such a hypergraph could be a social network where the nodes are user instances and the hyperedges are private groups. The disconnected hypergraphs represent disconnected communities where a user can only belong to a single community.

We show that our algorithm increases expressivity (Definition 2.8) for $h(S, H)$ of Equation 7.

**Theorem 5.6** (Invariance and Expressivity). *If* $L = \infty$, *GWL-1 enhanced by Algorithm 1 is still invariant to node isomorphism classes of* $\mathcal{H}$ *and can be strictly more expressive than GWL-1 to determine node isomorphism classes.*

Proving expressivity from Theorem 5.4 follows from the added information of component sizes viewable by each node in its vicinity. Proving the invariance from Theorem 5.4 follows by a proof by contradiction, which uses the maximality of the connected components found in Algorithm 1.

When training on a hypergraph with the data augmentations, every possible symmetry observed from the random augmentations will be learned. Thus the symmetry group of $h^L$ on the random matrix $\hat{H}_L$ is isomorphic to the intersection of all symmetries over each augmentation sample. This is expressed as follows:

$$Sym(h^L(\hat{H}_L)) \triangleq \bigcap_{\hat{H}'_L \sim P(\hat{H}_L)} Sym(h^L(\hat{H}'_L)) \tag{18}$$

We show that the intersection over all symmetries of the estimated multi-hypergraph $\hat{\mathcal{H}}_L$ has fewer symmetries than that of the $L$-GWL-1 view of the hypergraph as a collection of rooted trees. We call this **symmetry breaking**.

**Proposition 5.7.** *The multi-hypergraph $\hat{\mathcal{H}}_L$ breaks the symmetry of the L-GWL-1 view of the hypergraph $\mathcal{H}$:*

$$Sym(h^L(\hat{H}_L)) \subseteq Aut_c(\tilde{\mathcal{B}}^{2L}_{\mathcal{V},\mathcal{E}}), \forall L \geq 1 \tag{19}$$

This follows by the fact that the identity augmentation is in the support of the distribution of random augmentations and that $Sym(h^L(\hat{H}_L)) \cong Aut_c(\tilde{\mathcal{B}}^{2L}_{\mathcal{V},\mathcal{E}(\hat{\mathcal{H}})})$ by Theorem 4.4.

Since hyperGNNs represent each node $v \in \mathcal{V}$ by message passing through the neighbors in the rooted tree $(\tilde{\mathcal{B}}_{\mathcal{V},\mathcal{E}})_v$ at $v$. If probabilities are assigned between nodes, then $T$ layers of a hyperGNN can be viewed as computing the random walk probability of ending on any node starting from some uniformly chosen node. We define these terms in the following:

**Definition 5.5** (Chitra & Raphael (2019)). *A **random walk** on a (multi) hypergraph $\mathcal{H} = (\mathcal{V}, \mathcal{E})$ is a Markov chain with state space $\mathcal{V}$ with transition probabilities $P_{u,v} \triangleq \sum_{e \supset \{u,v\}:e \in \mathcal{E}} \frac{\omega(e)}{deg(u)|e|}$, where $\omega(e) : \mathcal{E} \to [0,1]$ is some discrete probability distribution on the hyperedges. When not specified, this is the constant 1 function.*

Assuming $\mathcal{H}$ is connected, let $X_t \in \mathcal{V}$ denote the state of the Markov chain at step $t$ with $P(X_0 = v) = \frac{1}{|\mathcal{V}|}, \forall v \in \mathcal{V}$. Letting $t \to \infty$, this probability converges to the stationary distribution on the nodes $\mathcal{V}$, which is independent of the time. This is expressed in the following definition:

**Definition 5.6.** *A **stationary distribution** $\pi : \mathcal{V} \to [0,1]$ for a Markov chain with transition probabilities $P_{u,v}$ is defined by the relationship $\sum_{u \in \mathcal{V}} P_{u,v}\pi(u) = \pi(v)$.*

*For a (multi) hypergraph random walk we have the closed form: $\pi(v) = \frac{deg(v)}{\sum_{u \in \mathcal{V}} deg(u)}$ for $v \in \mathcal{V}$ assuming $\mathcal{H}$ is a connected (multi) hypergraph.*

For the downstream training, we show that there are Bernoulli hyperedge drop/attachment probabilities $p, q_i$ respectively for each $\mathcal{R}_{c_L,i}$ so that the stationary distribution doesn't change. This shows that our data augmentation can still preserve the low frequency random walk signal.

**Proposition 5.8.** *For a connected hypergraph $\mathcal{H} = (\mathcal{V}, \mathcal{E})$, let $(\mathcal{R}_V, \mathcal{R}_E)$ be the output of Algorithm 1 on $\mathcal{H}$. Then there are Bernoulli probabilities $p, q_i$ for $i = 1, ..., |\mathcal{R}_V|$ for attaching a covering hyperedge so that $\hat{\pi}$ is an unbiased estimator of $\pi$.*

The intuition for Proposition 5.8 is that if a hyperedge is added to cover a connected subhypergraph $\mathcal{R}_{c_L,i}$ containing at least one hyperedge, then allowing any of the hyperedges in $\mathcal{R}_{c_L,i}$ to drop is enough to keep the estimated stationary distribution $\hat{\pi}$ unbiased.

**Time Complexity:**

Proposition 5.9 provides the time complexity of our algorithm.

**Proposition 5.9** (Complexity). *Algorithm 1 runs in time $O(nnz(H)L + (n + m))$, which is order linear in the size of the input star expansion matrix $H$ for hypergraph $\mathcal{H} = (\mathcal{V}, \mathcal{E})$, if $L$ is independent of $nnz(H)$, where $n = |\mathcal{V}|$, $nnz(H) = vol(\mathcal{V}) \triangleq \sum_{v \in \mathcal{V}} deg(v)$ and $m = |\mathcal{E}|$.*

Since Algorithm 1 runs in time linear in the size of the input when $L$ is constant, in practice it only takes a small fraction of the training time for hypergraph neural networks.

# 6 Evaluation

Table 1: Transductive hyperedge prediction PR-AUC scores on six different hypergraph datasets. The highest scores per hyperGNN architecture (row) is colored. Red text denotes the highest average scoring method. Orange text denotes a two-way tie and brown text denotes a three-way tie. All datasets involve predicting hyperedges of size 3.

| PR-AUC ↑ | Baseline | Ours | Baseln.+edrop |
|---|---|---|---|
| HGNN | 0.98 ± 0.03 | 0.99 ± 0.08 | 0.96 ± 0.02 |
| HGNNP | 0.98 ± 0.02 | 0.98 ± 0.09 | 0.96 ± 0.10 |
| HNHN | 0.98 ± 0.01 | 0.96 ± 0.07 | 0.97 ± 0.04 |
| HyperGCN | 0.98 ± 0.07 | 0.98 ± 0.11 | 0.98 ± 0.03 |
| UniGAT | 0.99 ± 0.06 | 0.99 ± 0.03 | 0.99 ± 0.07 |
| UniGCN | 0.99 ± 0.00 | 0.99 ± 0.03 | 0.99 ± 0.08 |
| UniGIN | 0.87 ± 0.12 | 0.86 ± 0.10 | 0.85 ± 0.08 |
| UniSAGE | 0.86 ± 0.04 | 0.86 ± 0.05 | 0.84 ± 0.09 |

(a) CAT-EDGE-DAWN

| PR-AUC ↑ | Baseline | Ours | Baseln.+edrop |
|---|---|---|---|
| HGNN | 0.90 ± 0.13 | 1.00 ± 0.00 | 0.90 ± 0.13 |
| HGNNP | 0.90 ± 0.09 | 1.00 ± 0.07 | 1.00 ± 0.03 |
| HNHN | 0.90 ± 0.09 | 0.91 ± 0.02 | 0.90 ± 0.08 |
| HyperGCN | 1.00 ± 0.00 | 1.00 ± 0.03 | 1.00 ± 0.02 |
| UniGAT | 0.90 ± 0.06 | 1.00 ± 0.03 | 1.00 ± 0.06 |
| UniGCN | 1.00 ± 0.01 | 0.91 ± 0.01 | 0.82 ± 0.09 |
| UniGIN | 0.90 ± 0.12 | 0.95 ± 0.06 | 0.90 ± 0.11 |
| UniSAGE | 0.90 ± 0.16 | 1.00 ± 0.08 | 0.90 ± 0.17 |

(b) CAT-EDGE-MUSIC-BLUES-REVIEWS

| PR-AUC ↑ | Baseline | Ours | Baseln.+ edrop |
|---|---|---|---|
| HGNN | 0.96 ± 0.10 | 0.98 ± 0.05 | 0.96 ± 0.04 |
| HGNNP | 0.96 ± 0.05 | 0.98 ± 0.09 | 0.97 ± 0.07 |
| HNHN | 0.96 ± 0.02 | 0.97 ± 0.08 | 0.97 ± 0.06 |
| HyperGCN | 0.93 ± 0.05 | 0.98 ± 0.07 | 0.96 ± 0.09 |
| UniGAT | 0.96 ± 0.01 | 0.98 ± 0.14 | 0.97 ± 0.04 |
| UniGCN | 0.96 ± 0.04 | 0.96 ± 0.11 | 0.96 ± 0.09 |
| UniGIN | 0.97 ± 0.03 | 0.97 ± 0.11 | 0.96 ± 0.05 |
| UniSAGE | 0.96 ± 0.10 | 0.96 ± 0.10 | 0.96 ± 0.02 |

(c) CONTACT-HIGH-SCHOOL

| PR-AUC ↑ | Baseline | Ours | Baseln.+edrop |
|---|---|---|---|
| HGNN | 0.95 ± 0.03 | 0.96 ± 0.01 | 0.95 ± 0.03 |
| HGNNP | 0.95 ± 0.02 | 0.96 ± 0.09 | 0.96 ± 0.07 |
| HNHN | 0.94 ± 0.07 | 0.97 ± 0.10 | 0.95 ± 0.05 |
| HyperGCN | 0.97 ± 0.01 | 0.97 ± 0.05 | 0.96 ± 0.08 |
| UniGAT | 0.95 ± 0.02 | 0.98 ± 0.07 | 0.98 ± 0.02 |
| UniGCN | 0.96 ± 0.00 | 0.97 ± 0.14 | 0.97 ± 0.10 |
| UniGIN | 0.95 ± 0.09 | 0.97 ± 0.02 | 0.95 ± 0.05 |
| UniSAGE | 0.96 ± 0.08 | 0.95 ± 0.05 | 0.96 ± 0.02 |

(d) CONTACT-PRIMARY-SCHOOL

| PR-AUC ↑ | Baseline | Ours | Baseln.+edrop |
|---|---|---|---|
| HGNN | 0.95 ± 0.07 | 0.97 ± 0.08 | 0.96 ± 0.07 |
| HGNNP | 0.95 ± 0.07 | 0.96 ± 0.02 | 0.96 ± 0.01 |
| HNHN | 0.94 ± 0.01 | 0.97 ± 0.02 | 0.95 ± 0.06 |
| HyperGCN | 0.92 ± 0.01 | 0.94 ± 0.06 | 0.94 ± 0.05 |
| UniGAT | 0.94 ± 0.08 | 0.98 ± 0.14 | 0.97 ± 0.08 |
| UniGCN | 0.97 ± 0.08 | 0.97 ± 0.14 | 0.97 ± 0.06 |
| UniGIN | 0.93 ± 0.07 | 0.94 ± 0.11 | 0.93 ± 0.09 |
| UniSAGE | 0.93 ± 0.07 | 0.93 ± 0.08 | 0.92 ± 0.04 |

(e) EMAIL-EU

| PR-AUC ↑ | Baseline | Ours | Baseln.+edrop |
|---|---|---|---|
| HGNN | 0.75 ± 0.09 | 0.85 ± 0.09 | 0.71 ± 0.14 |
| HGNNP | 0.83 ± 0.09 | 0.85 ± 0.08 | 0.85 ± 0.04 |
| HNHN | 0.72 ± 0.09 | 0.82 ± 0.03 | 0.74 ± 0.09 |
| HyperGCN | 0.87 ± 0.08 | 0.83 ± 0.05 | 1.00 ± 0.07 |
| UniGAT | 0.80 ± 0.09 | 0.83 ± 0.03 | 0.78 ± 0.05 |
| UniGCN | 0.84 ± 0.08 | 0.89 ± 0.10 | 0.71 ± 0.07 |
| UniGIN | 0.69 ± 0.14 | 0.76 ± 0.05 | 0.61 ± 0.11 |
| UniSAGE | 0.72 ± 0.11 | 0.71 ± 0.10 | 0.64 ± 0.10 |

(f) CAT-EDGE-MADISON-RESTAURANTS

We evaluate our method on higher order link prediction with many of the standard hypergraph neural network methods. Due to potential class imbalance, we measure the PR-AUC of higher order link prediction on the hypergraph datasets. These datasets are: CAT-EDGE-DAWN, CAT-EDGE-MUSIC-BLUES-REVIEWS, CONTACT-HIGH-SCHOOL, CONTACT-PRIMARY-SCHOOL, EMAIL-Eu, CAT-EDGE-MADISON-RESTAURANTS. These datasets range from representing social interactions as they develop over time to collections of reviews to drug combinations before overdose. We also evaluate on the AMHERST41 dataset, which is a graph dataset. All of our datasets are unattributed hypergraphs/graphs.

**Data Splitting:** For the hypergraph datasets, each hyperedge in it is paired with a timestamp (a real number). These timestamps are a physical time for which a higher order interaction, represented by a hyperedge, occurs. We form a train-val-test split by letting the train be the hyperedges associated with the 80th percentile of timestamps, the validation be the hyperedges associated with the timestamps in between the 80th and 85th percentiles. The test hyperedges are the remaining hyperedges. The train validation and test datasets thus form a partition of the nodes. We do the task of hyperedge prediction for sets of nodes of size 3, also known as triangle prediction. Half of the size 3 hyperedges in each of train, validation and test are used as positive examples. For each split, we select random subsets of nodes of size 3 that do not form hyperedges for negative sampling. We maintain positive/negative class balance by sampling the same number of negative samples as positive samples. Since the test distribution comes from later time stamps than those in training, there is a possibility that certain datasets are out-of-distribution if the hyperedge distribution changes.

For the graph dataset, the single graph is deterministically split into 80/5/15 for train/val/test. We remove 10% of the edges in training and let them be positive examples $P_{tr}$ to predict. For validation and test, we remove 50% of the edges from both validation and test to set as the positive examples $P_{val}, P_{te}$ to predict. For train, validation, and test, we sample $|P_{tr}|, |P_{val}|, |P_{te}|$ negative link samples from the links of train, validation and test.

Table 2: PR-AUC on graph dataset AMHERST41. Each column is a comparison of the baseline PR-AUC scores against the PR-AUC score for our method (first row) applied to a standard hyperGNN architecture. The coloring scheme is the same as in Table 1.

| PR-AUC ↑ | HGNN | HGNNP | HNHN | HyperGCN | UniGAT | UniGCN | UniGIN | UniSAGE |
|---|---|---|---|---|---|---|---|---|
| Ours | 0.73 ± 0.10 | 0.61 ± 0.05 | 0.64 ± 0.06 | 0.71 ± 0.09 | 0.72 ± 0.08 | 0.70 ± 0.08 | 0.73 ± 0.03 | 0.73 ± 0.06 |
| hyperGNN Baseline | 0.62 ± 0.09 | 0.62 ± 0.10 | 0.63 ± 0.04 | 0.71 ± 0.07 | 0.70 ± 0.06 | 0.69 ± 0.07 | 0.73 ± 0.06 | 0.73 ± 0.09 |
| hyperGNN Baseln.+edrop | 0.61 ± 0.03 | 0.61 ± 0.03 | 0.61 ± 0.09 | 0.71 ± 0.06 | 0.71 ± 0.02 | 0.69 ± 0.05 | 0.73 ± 0.09 | 0.73 ± 0.04 |
| APPNP | 0.42 ± 0.07 | 0.42 ± 0.07 | 0.42 ± 0.07 | 0.42 ± 0.07 | 0.42 ± 0.07 | 0.42 ± 0.07 | 0.42 ± 0.07 | 0.42 ± 0.07 |
| APPNP+edrop | 0.42 ± 0.03 | 0.42 ± 0.03 | 0.42 ± 0.03 | 0.42 ± 0.03 | 0.42 ± 0.03 | 0.42 ± 0.03 | 0.42 ± 0.03 | 0.42 ± 0.03 |
| GAT | 0.49 ± 0.06 | 0.49 ± 0.06 | 0.49 ± 0.06 | 0.49 ± 0.06 | 0.49 ± 0.06 | 0.49 ± 0.06 | 0.49 ± 0.06 | 0.49 ± 0.06 |
| GAT+edrop | 0.49 ± 0.06 | 0.49 ± 0.06 | 0.49 ± 0.06 | 0.49 ± 0.06 | 0.49 ± 0.06 | 0.49 ± 0.06 | 0.49 ± 0.06 | 0.49 ± 0.06 |
| GCN2 | 0.56 ± 0.12 | 0.56 ± 0.12 | 0.56 ± 0.12 | 0.56 ± 0.12 | 0.56 ± 0.12 | 0.56 ± 0.12 | 0.56 ± 0.12 | 0.56 ± 0.12 |
| GCN2+edrop | 0.54 ± 0.02 | 0.54 ± 0.02 | 0.54 ± 0.02 | 0.54 ± 0.02 | 0.54 ± 0.02 | 0.54 ± 0.02 | 0.54 ± 0.02 | 0.54 ± 0.02 |
| GCN | 0.40 ± 0.03 | 0.40 ± 0.03 | 0.40 ± 0.03 | 0.40 ± 0.03 | 0.40 ± 0.03 | 0.40 ± 0.03 | 0.40 ± 0.03 | 0.40 ± 0.03 |
| GCN+edrop | 0.65 ± 0.04 | 0.65 ± 0.04 | 0.65 ± 0.04 | 0.65 ± 0.04 | 0.65 ± 0.04 | 0.65 ± 0.04 | 0.65 ± 0.04 | 0.65 ± 0.04 |
| GIN | 0.73 ± 0.10 | 0.73 ± 0.10 | 0.73 ± 0.10 | 0.73 ± 0.10 | 0.73 ± 0.10 | 0.73 ± 0.10 | 0.73 ± 0.10 | 0.73 ± 0.10 |
| GIN+edrop | 0.73 ± 0.10 | 0.73 ± 0.10 | 0.73 ± 0.10 | 0.73 ± 0.10 | 0.73 ± 0.10 | 0.73 ± 0.10 | 0.73 ± 0.10 | 0.73 ± 0.10 |
| GraphSAGE | 0.44 ± 0.01 | 0.44 ± 0.01 | 0.44 ± 0.01 | 0.44 ± 0.01 | 0.44 ± 0.01 | 0.44 ± 0.01 | 0.44 ± 0.01 | 0.44 ± 0.01 |
| GraphSAGE+edrop | 0.44 ± 0.10 | 0.44 ± 0.10 | 0.44 ± 0.10 | 0.44 ± 0.10 | 0.44 ± 0.10 | 0.44 ± 0.10 | 0.44 ± 0.10 | 0.44 ± 0.10 |

## 6.1 Architecture and Training

Our algorithm serves as a preprocessing step for selective data augmentation. Given a single training hypergraph $\mathcal{H}$, the Algorithm 1 is applied and during training, the identified hyperedges of the symmetric induced subhypergraphs of $\mathcal{H}$ are randomly replaced with single hyperedges that cover all the nodes of each induced subhypergraph. Each symmetric subhypergraph has a $p = 0.5$ probability of being selected. To get a large set of symmetric subhypergraphs, we run 2 iterations of GWL-1.

We implement $h(S, H)$ from Equation 7 as follows. Upon extracting the node representations from the hypergraph neural network, we use a multi-layer-perceptron (MLP) on each node representation, sum across such compositions, then apply a final MLP layer after the aggregation. We use the binary cross entropy loss on this multi-node representation for training. We always use 5 layers of hyperGNN convolutions, a hidden dimension of 1024, and a learning rate of 0.01.

## 6.2 Higher Order Link Prediction Results

We show in Table 1 the comparison of PR-AUC scores amongst the baseline methods of HGNN, HGNNP, HNHN, HyperGCN, UniGIN, UniGAT, UniSAGE, their hyperedge dropped versions, and "Our" method, which preprocesses the hypergraph to break symmetry during training. For the hyperedge drop baselines, there is a uniform 50% chance of dropping any hyperedge. We use the Laplacian eigenmap Belkin & Niyogi (2003) positional encoding on the clique expansion of the input hypergraph. This is common practice in (hyper)link prediction and required for using a hypergraph neural network on an unattributed hypergraph.

We show in Table 2 the PR-AUC scores on the AMHREST41. Along with hyperGNN architectures we use for the hypergraph experiments, we also compare with standard GNN architectures: APPNP Gasteiger et al. (2018), GAT Veličković et al. (2017), GCN2 Chen et al. (2020a), GCN Kipf & Welling (2016a), GIN Xu et al. (2018), and GraphSAGE Hamilton et al. (2017). For every hyperGNN/GNN architecture, we also apply drop-edge Rong et al. (2019) to the input graph and use this also as baseline. The number of layers of each GNN is set to 5 and the hidden dimension at 1024. For APPNP and GCN2, one MLP is used on the initial node positional encodings.

Overall, our method performs well across a diverse range of higher order network datasets. We observe that our method can often outperform the baseline of not performing any data perturbations as well as the same baseline with uniformly random hyperedge dropping. Our method has an added advantage of being explainable since our algorithm works at the data level. There was also not much of a concern for computational time since our algorithm runs in time $O(nnz(H) + n + m)$, which is optimal since it is the size of the input.

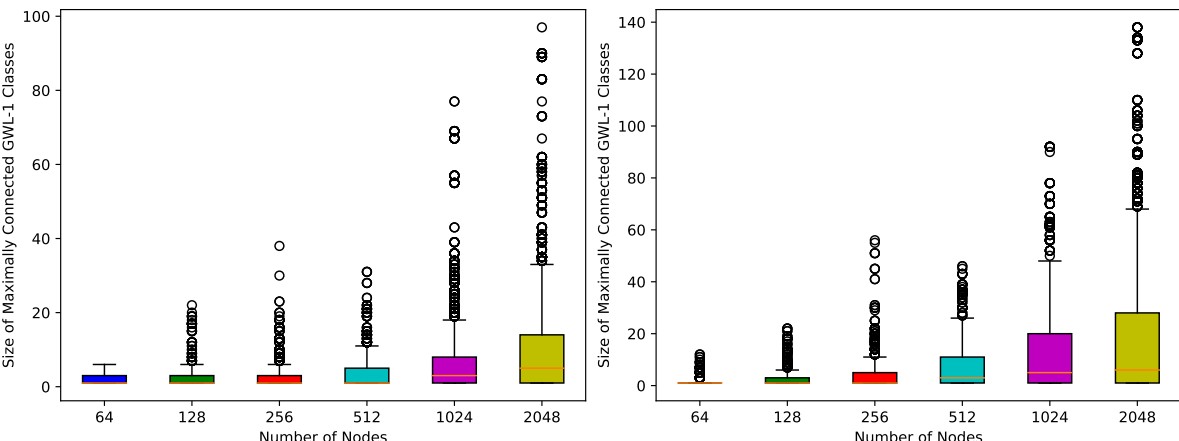

(a) Boxplot of the sizes of the connected components with equal GWL-1 node values from the hy-MMSBM sampling algorithm where there are three independent communities.

(b) Boxplot of the sizes of the connected components with equal GWL-1 node values from the hy-MMSBM sampling algorithm where any two of the three communities can communicate.

Figure 4: Experiment on the relationship between the sizes of connected components of equal GWL-1 node values and the communication between communities.

## 6.3 Empirical Observations on the Components Discovered by the Algorithm

According to Proposition B.14, we know that the symmetry finding algorithm always covers the hypergraph and thus that we can generate on the order of $O(2^{|\mathcal{V}|})$ counterfactual multi-hypergraphs. It is known that a large set of data augmentations during learning improves learner generalization. The cover by GWL-1 symmetric components follows a distribution depending on the data.

We show in Figure 4 the distributions for the component sizes over all GWL-1 symmetric connected components for samples from the Hy-MMSBM model Ruggeri et al. (2023). This is a model to sample hypergraphs with community structure. In Figure 4a we sample hypergraphs with 3 isolated communities, meaning that there is 0 chance of any interconnections between any two communities. In Figure 4b we sample hypergraphs with 3 communities where every node in a community has a weight of 1 to stay in its community and a weight of 0.2 to move out to any other community. We plot the boxplots as a function of increasing number of nodes. We notice that the more communication there is between communities for more nodes there is more spread in possible connected component sizes. Isolated communities should make for predictable clusters/connected components.

## 7 Discussion

Our proposed data augmentation method uses symmetry breaking to handle the symmetries induced by GWL-1 based hyperGNNs. Symmetry breaking provides some guarantees that the symmetries induced by the hyperGNN are brought closer to the symmetries of the training hypergraph. These include hyperlink prediction guarantees. In addition, symmetry breaking prepares the hyperGNN for an automorphism group different from the symmetries it views during training. This brings about an important question regarding the invariant automorphism group across training and testing, namely:

*What is the relationship between the symmetries of the training and testing hypergraphs?*

We answer this question in the context of a temporal shift from training to testing. The term "temporal shift" is usually used in the context of distribution shifts Yao et al. (2022a). We define it in terms of transductive hyperlink prediction.

A temporal shift from training to testing means that the testing hyperedges have a temporal future relationship with the training hyperedges.

We formalize a hypergraph in terms of a physical system that can change over time with the following assumptions.

**Assumption 7.1.** *If we view a hypergraph as a physical system of particles where only the nodes have mass, then the task of transductive higher order link prediction is on a closed system Landau & Lifshitz (1960). This means no mass can enter or leave this system.*

*Given $n$ nodes $\mathcal{V}$, we can view each node $v \in \mathcal{V}$ along with the hypergraph $\mathcal{H} = (\mathcal{V}, \mathcal{E})$ it belongs in as a microstate $(v, \mathcal{H})$ of a statistical ensemble.*

*In Rashevsky (1955) the entropy of a graph is defined through the orbits of the automorphism group. We generalize this to a node based definition for hypergraphs. Our definition uses a similar probability on microstates. We define the **hypergraph topological entropy** of a hypergraph $\mathcal{H} = (\mathcal{V}, \mathcal{E})$ by:*

$$S \triangleq -\sum_{v \in \mathcal{V}} p_v(\mathcal{H}) \log(p_v(\mathcal{H}));$$
$$\text{with } p_v(\mathcal{H}) \triangleq \frac{n_v(\mathcal{H})}{\sum_{v \in \mathcal{V}} n_v(\mathcal{H})} \tag{20}$$

*where $n_v(\mathcal{H}) \triangleq |\{u \in \mathcal{V} : u \cong_{\mathcal{H}} v\}|$ is the number of nodes $u \in \mathcal{V}$ isomorphic to node $v$, including $v$ itself, (See Definition 2.7).*

**Assumption 7.2.** *(**Second Law of Thermodynamics** ): The second law of thermodynamics Carnot (1978) states that the entropy of a closed system must increase over time. This is denoted by the following equation:*

$$\Delta S > 0 \tag{21}$$

*This law can be used in terms of the hypergraph topological entropy as defined in Assumption 7.1.*

We also assume the following random model on the hypergraph we predict on. Let $\bullet = \mathrm{tr}, \mathrm{te}$ represent temporally ordered training and testing distributions where $(\mathcal{E}_\bullet)_{gt}$ are the hyperedges of $(\mathcal{H}_\bullet)_{gt}$.

**Assumption 7.3.** *For each node $v \in \mathcal{V}$, let $X_v$ be independent Bernoulli random variables of some probability $q_v \in [0, 1]$. Let $|(\mathcal{E}_{te})_{gt}|$ be the number of testing hyperedges of $(\mathcal{H}_{te})_{gt}$. It is a random variable defined by:*

$$f((X_v)_{v \in \mathcal{V}}) = \sum_{e \subseteq \mathcal{V}} \Pi_{u \in e} X_u \tag{22}$$

*Assume further that $|(\mathcal{E}_{te})_{gt}|, (\mathcal{V}, (\mathcal{E}_{te})_{gt}) \sim P(\mathcal{H}; t_{te})$ only depends on $n$.*

*Let $\tilde{X}_v$ be a Bernoulli random variable with probability $q_{max}$. Assume that $f$ satisfies*

$$P(f(\tilde{X}_{v \neq u}, ..., do(\tilde{X}_u = 1), ..., \tilde{X}_{v \neq u}) - f(\tilde{X}_{v \neq u}, ..., do(\tilde{X}_u = 0), ..., \tilde{X}_{v \neq u}) \leq 2) \leq n^{-\omega(1)}, \forall u \in \mathcal{V} \tag{23}$$

These assumptions show that with high probability there are node isomorphism classes which decrease in size:

**Theorem 7.1.** *Under Assumptions 7.1, 7.3, and the Second Law of Thermodynamics on the hypergraph viewed as a closed system:*

$$\exists \mathcal{U} \subseteq \mathcal{V}, \mathcal{U} \neq \emptyset, \text{ so that: } n_v((\mathcal{H}_{tr})_{gt}) > n_v((\mathcal{H}_{te})_{gt}), \forall v \in \mathcal{U}, \text{ with probability } 1 - O(\frac{1}{\sqrt{n}}) \tag{24}$$

*Proof.* By the second law of thermodynamics, we must have that between the training hypergraph $(\mathcal{H}_{tr})_{gt} = (\mathcal{V}, (\mathcal{E}_{tr})_{gt})$ and testing hypergraph $(\mathcal{H}_{te})_{gt} = (\mathcal{V}, (\mathcal{E}_{te})_{gt})$ which is temporally later than $(\mathcal{H}_{tr})_{gt}$,

$$\Delta S = -\sum_{v \in \mathcal{V}} p_v((\mathcal{H}_{te})_{gt}) \log(p_v((\mathcal{H}_{te})_{gt})) + \sum_{v \in \mathcal{V}} p_v((\mathcal{H}_{tr})_{gt}) \log(p_v((\mathcal{H}_{tr})_{gt})) > 0 \tag{25a}$$

If we take an upper bound on $\Delta S$, we get the following consequence:

$$0 < \Delta S \leq \sum_{v \in \mathcal{V}} \log(\frac{p_v((\mathcal{H}_{tr})_{gt})}{p_v((\mathcal{H}_{te})_{gt})}) \Rightarrow p_v((\mathcal{H}_{tr})_{gt}) > p_v((\mathcal{H}_{te})_{gt}), \forall v \in \mathcal{U}, \text{ for some } \mathcal{U} \subseteq \mathcal{V}, \mathcal{U} \neq \emptyset \quad (26)$$

If $n_v((\mathcal{H}_{tr})_{gt}) > n_v((\mathcal{H}_{te})_{gt}), \forall v \in \mathcal{U} \subseteq \mathcal{V}$, we can conclude that some nodes will shrink the size of their ground truth isomorphism class.

We show that this occurs with high probability by bounding the complementary case.

**Nodes rarely increase their isomorphism class size:**

For the complementary case, $n_v((\mathcal{H}_{tr})_{gt}) \leq n_v((\mathcal{H}_{te})_{gt}), \forall v \in \mathcal{U} \subseteq \mathcal{V}$, we show that under Assumption 7.3, node isomorphism class cardinality growth occurs with low probability due to an anti-concentration bound on multivariate polynomials Fox et al. (2021).

In this complementary case, we must have that the nodes in $\mathcal{U} \subseteq \mathcal{V}$ increased the number of nodes isomorphic to them. This implies that each of these nodes $v \in \mathcal{U}$ must have changed their degree vector upon changing $(\mathcal{H}_{tr})_{gt}$ to $(\mathcal{H}_{te})_{gt}$. This change requires that some other node $u \in \mathcal{V}, u \neq v$ obtains a degree vector equal to the degree vector of $v$.

Let us define this space of all possible testing hyperedge sets with this necessary condition:

$$\text{DegVec}_{eq}(v, (\mathcal{H}_{te})_{gt}) \triangleq \{E \in supp(P((\mathcal{E}_{te})_{gt}; t_{te})) : \exists u \in \mathcal{V}, \text{degvec}_{(\mathcal{V},E)}(v) = \text{degvec}_{(\mathcal{V},E)}(u), u \neq v \} \quad (27)$$

We can express the relationship between the change in $n_v$ with membership in $\text{DegVec}_{eq}(v, (\mathcal{H}_{te})_{gt})$:

$$n_v((\mathcal{H}_{tr})_{gt}) \leq n_v((\mathcal{H}_{te})_{gt}) \Rightarrow (\mathcal{E}_{te})_{gt} \in \text{DegVec}_{eq}(v, (\mathcal{H}_{te})_{gt}) \quad (28)$$

Letting

$$x_{min}(v) = \min_{E \in \text{DegVec}_{eq}(v, (\mathcal{H}_{te})_{gt})} |E|, \forall v \in \mathcal{V} \quad (29a)$$

and

$$x_{max}(v) = \max_{E \in \text{DegVec}_{eq}(v, (\mathcal{H}_{te})_{gt})} |E|, \forall v \in \mathcal{V} \quad (29b)$$

We can then say that the number of testing hyperedges $|(\mathcal{E}_{te})_{gt}|$ is in the interval $[x_{min}(v), x_{max}(v)]$:

$$(\mathcal{E}_{te})_{gt} \in DegVec_{eq}(v, (\mathcal{H}_{te})_{gt}) \Rightarrow x_{min}(v) \leq f((X_u)_{u \in \mathcal{V}}) \leq x_{max}(v) \quad (30)$$

Let the diameter of the interval $[x_{min}(v), x_{max}(v)]$ be given by $D$:

$$D \triangleq x_{max}(v) - x_{min}(v) \quad (31)$$

This only depends on $n$ since the minimizers and maximizers $E_{x_{min}}, E_{x_{max}}$ of Equations 29a and 29b satisfy: $E_{x_{min}}, E_{x_{max}} \in \text{DegVec}_{eq}(v, (\mathcal{H}_{te})_{gt})$ and thus belong to the $supp(P(E; t_{te}))$. We know that any $E \in supp(P(E; t_{te}))$ has that $|E|$ depends only on $n$ by Assumption 7.3.

Define the median $M$ on $[x_{min}(v), x_{max}(v)]$ by:

$$M \triangleq \frac{x_{max}(v) + x_{min}(v)}{2} \quad (32)$$

We thus have that:

$$x_{min}(v) \leq f((X_u)_{u \in \mathcal{V}}) \leq x_{max}(v) \Rightarrow |f((X_u)_{u \in \mathcal{V}}) - M| \leq D \quad (33)$$

Piecing together Equations 28, 30, and 33, we have by monotonicity:

$$P(n_v((\mathcal{H}_{tr})_{gt}) \leq n_v((\mathcal{H}_{te})_{gt})) \leq P(x_{min}(v) \leq f((X_u)_{u \in \mathcal{V}}) \leq x_{max}(v)) \quad (34a)$$

$$\leq P(|f((X_u)_{u \in \mathcal{V}}) - M| < D) \tag{34b}$$

$$\leq P(|M - f((\tilde{X}_u)_{u \in \mathcal{V}})| < D) \leq O(\frac{1}{\sqrt{n}}), \forall v \in \mathcal{U} \subseteq \mathcal{V} \tag{34c}$$

Where the last inequality comes from the anti-concentration bound of Theorem 1.2 of Fox et al. (2021), which states:

$$P(|f((\tilde{X}_v)_{v \in \mathcal{V}}) - x| < s) \leq O(\frac{1}{\sqrt{n}}), \forall x \in \mathbb{R} \tag{35}$$

for any $s > 0$ which may depend on $n$ and any $x \in \mathbb{R}$. Setting $x := M, s := D$, gives the last inequality.

$\square$

Theorem 7.1 states that the ground truth node isomorphism classes for the nodes must shrink with probability on order $1 - O(\frac{1}{\sqrt{n}})$. Thus, for large $n \gg 0$, we have that with high probability that the nodes $v \in \mathcal{U} \subseteq \mathcal{V}, \mathcal{U} \neq \emptyset$ have $n_v((\mathcal{H}_{tr})_{gt}) > n_v((\mathcal{H}_{te})_{gt})$.

We can recognize this property of $(\mathcal{H}_{te})_{gt}$ by shrinking the automorphism group that the hypergraph encoder recognizes from the training hypergraph. According to Proposition 5.7, symmetry breaking as given in Equation 18, does this.

Our method breaks the symmetry of a GWL-1 based hyperGNN. This, of course, is not of importance if the symmetry group of the GWL-1 based hyperGNN on some training hypergraph is already the trivial group. Nonetheless, our symmetry breaking method is theoretically beneficial. Our method can also be used within other downstream learning methods such as feature averaging Lyle et al. (2020), and ensemble methods, as mentioned in Section 5.

## 8 Conclusion

Many existing hyperGNN architectures are based on the GWL-1 algorithm, which is a hypergraph isomorphism testing algorithm. We have characterized and identified the limitations of GWL-1. GWL-1 views the hypergraph as a collection of rooted trees. This means that hyperGNNs recognize more symmetries than the natural automorphisms of the training hypergraph. In fact, maximally connected subsets of nodes that share the same value of GWL-1, which act like regular hypergraphs, are indistinguishable. To address this issue while respecting the structure of a hypergraph, we have devised a preprocessing algorithm that identifies all such connected components. These components cover the hypergraph and allow for downstream data augmentation of symmetry breaking by training on a random multi-hypergraph. We show that this approach improves the expressivity of a hyperGNN learner, including in the case of hyperlink prediction. We perform extensive experiments to evaluate the effectiveness of our approach and make empirical observations about the output of the algorithm on hypergraph data.

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

# Appendix

## A   More Background

We discuss in this section about the basics of graph representation learning and link prediction. Graphs are hypergraphs with all hyperedges of size 2. Simplicial complexes and hypergraphs are generalizations of graphs. We also discuss more related work.

### A.1   Graph Neural Networks and Weisfeiler-Lehman 1

The Weisfeiler-Lehman (WL-1) algorithm is an isomorphism testing approximation algorithm. It involves repeatedly message passing all nodes with their neighbors, a step called node label refinement. The WL-1 algorithm never gives false negatives when predicting whether two graphs are isomorphic. In other words, two isomorphic graphs are always indistinguishable by WL-1.

The WL-1 algorithm is the following successive vertex relabeling applied until convergence on a graph $G = (X, A)$ (a pair of the set of node attributes and the graph's adjacency structure):

$$
\begin{aligned}
h_v^0 &\leftarrow X_v, \forall v \in \mathcal{V}_G \\
h_v^{i+1} &\leftarrow \{\!\!\{ (h_v^i, h_u^i) \}\!\!\}_{u \in Nbr_A(v)}, \forall v \in \mathcal{V}_G
\end{aligned}
\tag{36}
$$

The algorithm terminates after the vertex labels converge. For graph isomorphism testing, the concatenation of the histograms of vertex labels for each iteration is output as the graph representation. Since we are only concerned with node isomorphism classes, we ignore this step and just consider the node labels $h_v^i$ for every $v \in \mathcal{V_C}$.

The WL-1 isomorphism test can be characterized in terms of rooted tree isomorphisms between the universal covers for connected graphs Krebs & Verbitsky (2015). There have also been characterizations of WL-1 in terms of counting homomorphisms Knill (2013) as well as the Wasserstein Distance Chen et al. (2022) and Markov chains Chen et al. (2023).

A graph neural network (GNN) is a message passing based node representation learner modeled after the WL-1 algorithm. It has the important inductive bias of being equivariant to node indices. As a neural model of the WL-1 algorithm, it learns neural weights common across all nodes in order to obtain a vector representation for each node. A GNN must use some initial node attributes in order to update its neural weights. There are many variations on GNNs, including those that improve the distinguishing power beyond WL-1. For two surveys on the GNNs and their applications, see Zhou et al. (2020); Wu et al. (2020).

### A.2   Link Prediction

The task of link prediction on graphs involves the prediction of the existence of links. There are two kinds of link prediction. There is transductive link prediction where the same nodes are used for all of train validation and testing. There is also inductive link prediction where the test validation and training nodes can all be disjoint. Some existing works on link prediction include Zhang & Chen (2017). Higher order link prediction is a generalization of link prediction to hypergraph data.

A common way to do link prediction is to compute a node-based GNN and for a pair of nodes, aggregate, similar to in graph auto encoders Kipf & Welling (2016b), the node representations in any target pair in order to obtain a 2-node representation. Such aggregations are of the form:

$$
h(S = \{u, v\}) = \sigma(h_u \cdot h_v)
\tag{37}
$$

where $S$ is a pair of nodes. As shown in Proposition B.4, this guaranteems an equivariant 2-node representation but can often give false predictions even with a fully expressive node-based GNN Wang et al. (2023). A common remedy for this problem is to introduce positional encodings such as SEAL Wang et al. (2022) and DistanceEncoding Li et al. (2020). Positional encodings encode the relative distances amongst nodes via a low distortion embedding for example. In the related work section we have gone over many of these embeddings. We have also used these in our evaluation since they are common practice and must exist to compute a hypergraph neural network if there are no ground truth node attributes. According to Srinivasan & Ribeiro (2019), fully expressive pairwise node representations, as defined by 2-node invariance and expressivity, can be represented by some fully expressive positional embedding, which is a positional embedding that is injective on the node pair isomorphism classes. It is not clear how one would achieve this in practice, however. Another remedy is to increase the expressive power of WL-1 to WL-2 for link prediction Hu et al. (2022).

### A.3 More Related Work

The work of Wei et al. (2022) also does a data augmentation scheme. It considers randomly dropping edges and generating data through a generative model on hypergraphs. The work of Lee & Shin (2022) also performs data augmentation on a hypergraph so that homophilic relationships are maintained. It does this through contrastive losses at the node to node, hyperedge to hyperedge and intra hyperedge level. Neither of these methods provide guarantees for their data augmentations.

As mentioned in the main text, an ensemble of neural networks can be used with a drop-out Baldi & Sadowski (2014) like method on the output of the Algorithm. Subgraph neural networks Alsentzer et al. (2020); Tan et al. (2023) are ensembles of models on subgraphs of the input graph.

Some more of the many existing hypergraph neural network architectures include: Kim et al. (2022); Cai et al. (2022); Chien et al. (2021); Bai et al. (2021); Li et al. (2023a); Arya et al. (2020).

# B Proofs

In this section we provide the proofs for all of the results in the main paper along with some additional theory.

## B.1 Hypergraph Isomorphism

We first repeat the definition of a hypergraph and its corresponding matrix representation called the star expansion matrix::

**Definition B.1.** *An undirected hypergraph is a pair $\mathcal{H} = (\mathcal{V}, \mathcal{E})$ consisting of a set of vertices $\mathcal{V}$ and a set of hyperedges $\mathcal{E} \subseteq 2^{\mathcal{V}}$ where $2^{\mathcal{V}}$ is the power set of the vertex set $\mathcal{V}$.*

**Definition B.2.** *The star expansion incidence matrix $H$ of a hypergraph $\mathcal{H} = (\mathcal{V}, \mathcal{E})$ is the $|\mathcal{V}| \times 2^{|\mathcal{V}|}$ 0-1 incidence matrix $H$ where $H_{v,e} = 1$ iff $v \in e$ for $(v, e) \in \mathcal{V} \times \mathcal{E}$ for some fixed orderings on both $\mathcal{V}$ and $2^{\mathcal{V}}$.*

We recall the definition of an isomorphism between hypergraphs:

**Definition B.3.** *For two hypergraphs $\mathcal{H}$ and $\mathcal{D}$, a structure preserving map $\rho : \mathcal{H} \to \mathcal{D}$ is a pair of maps $\rho = (\rho_{\mathcal{V}} : \mathcal{V}_{\mathcal{H}} \to \mathcal{V}_{\mathcal{D}}, \rho_{\mathcal{E}} : \mathcal{E}_{\mathcal{H}} \to \mathcal{E}_{\mathcal{D}})$ such that $\forall e \in \mathcal{E}_{\mathcal{H}}, \rho_{\mathcal{E}}(e) \triangleq \{\rho_{\mathcal{V}}(v_i) \mid v_i \in e\} \in \mathcal{E}_{\mathcal{D}}$. A hypergraph isomorphism is a structure preserving map $\rho = (\rho_{\mathcal{V}}, \rho_{\mathcal{E}})$ such that both $\rho_{\mathcal{V}}$ and $\rho_{\mathcal{E}}$ are bijective. Two hypergraphs are said to be isomorphic, denoted as $\mathcal{H} \cong \mathcal{D}$, if there exists an isomorphism between them. When $\mathcal{H} = \mathcal{D}$, an isomorphism $\rho$ is called an automorphism on $\mathcal{H}$. All the automorphisms form a group, which we denote as $Aut(\mathcal{H})$.*

The action of $\pi \in Sym(\mathcal{V})$ on the star expansion adjacency matrix $H$ is repeated here for convenience:

$$(\pi \cdot H)_{v,e=(u_1,\ldots,v,\ldots,u_k)} \triangleq H_{\pi^{-1}(v),\pi^{-1}(e)=(\pi^{-1}(u_1),\ldots,\pi^{-1}(v),\ldots,\pi^{-1}(u_k))} \tag{38}$$

Based on the group action, consider the stabilizer subgroup of $Sym(\mathcal{V})$ on the star expansion adjacency matrix $H$ defined as follows:

$$Stab_{Sym(\mathcal{V})}(H) = \{\pi \in Sym(\mathcal{V}) \mid \pi \cdot H = H\} \tag{39}$$

For simplicity we omit the lower index when the permutation group is clear from the context. It can be checked that $Stab(H) \leq Sym(\mathcal{V})$ is a subgroup. Intuitively, $Stab(H)$ consists of all permuations that leave $H$ fixed.

For a given hypergraph $\mathcal{H} = (\mathcal{V}, \mathcal{E})$, there is a relationship between the group of hypergraph automorphisms $Aut(\mathcal{H})$ and the stabilizer group $Stab(H)$ on the star expansion adjacency matrix.

**Proposition B.1.** *$Aut(\mathcal{H}) \cong Stab(H)$ are equivalent as isomorphic groups.*

*Proof.* Consider $\rho \in Aut(\mathcal{H})$, define the map $\Phi : \rho \mapsto \pi := \rho|_{\mathcal{V}(\mathcal{H})}$. The group element $\pi \in Sym(\mathcal{V})$ acts as a stabilizer of $H$ since for any entry $(v, e)$ in $H$, $H_{\pi^{-1}(v),\pi^{-1}(e)} = (\pi \cdot H)_{v,e} = 1$ iff $\pi^{-1}(e) \in \mathcal{E}_{\mathcal{H}}$ iff $e \in \mathcal{E}_{\mathcal{H}}$ iff $H_{v,e} = 1 = H_{\pi \circ \pi^{-1}(v), \pi \circ \pi^{-1}(e)}$. Since $(v, e)$ was arbitrary, $\pi$ preserves the positions of the nonzeros.

We can check that $\Phi$ is a well defined injective homorphism as a restriction map. Furthermore it is surjective since for any $\pi \in Stab(H)$, we must have $H_{v,e} = 1$ iff $(\pi \cdot H)_{v,e} = H_{\pi^{-1}(v),\pi^{-1}(e)} = 1$ which is equivalent to $v \in e \in \mathcal{E}$ iff $\pi(v) \in \pi(e) \in \mathcal{E}$ which implies $e \in \mathcal{E}$ iff $\pi(e) \in \mathcal{E}$. Thus $\Phi$ is a group isomorphism from $Aut(\mathcal{H})$ to $Stab(H)$ □

In other words, to study the symmetries of a given hypergraph $\mathcal{H}$, we can equivalently study the automorphisms $Aut(\mathcal{H})$ and the stabilizer permutations $Stab(H)$ on its star expansion adjacency matrix $H$. Intuitively, the stabilizer group $0 \leq Stab(H) \leq Sym(\mathcal{V})$ characterizes the symmetries in a graph. When the graph has rich symmetries, say a complete graph, $Stab(H) = Sym(\mathcal{V})$ can be as large as the whole permutaion group.

Nontrivial symmetries can be represented by isomorphic node sets which we define as follow:

**Definition B.4.** *For a given hypergraph $\mathcal{H}$ with star expansion matrix $H$, two $k$-node sets $S, T \subseteq \mathcal{V}$ are called* isomorphic*, denoted as $S \simeq T$, if $\exists \pi \in Stab(H), \pi(S) = T$ and $\pi(T) = S$.*

When $k = 1$, we have isomorphic nodes, denoted $u \cong_{\mathcal{H}} v$ for $u, v \in \mathcal{V}$. Node isomorphism is also studied as the so-called structural equivalence in Lorrain & White (1971). Furthermore, when $S \simeq T$ we can then say that there is a matching due to the graph of the $\pi$ map of the form $\{(s, \pi(s)) : s \in S\}$. This matching is between the node sets $S$ and $T$ so that matched nodes are isomorphic.

**Definition B.5.** *A $k$-node representation $h$ is **$k$-permutation equivariant** if:*

*for all $\pi \in Sym(\mathcal{V})$, $S \in 2^{\mathcal{V}}$ with $|S| = k$: $h(\pi \cdot S, H) = h(S, \pi \cdot H)$*

**Proposition B.2.** *If $k$-node representation $h$ is $k$-permutation equivariant, then $h$ is $k$-node invariant.*

*Proof.* given $S, S' \in \mathcal{C}$ with $|S| = |S'| = k$,

if there exists a $\pi \in Stab(H)$ (meaning $\pi \cdot H = H$) and $\pi(S) = S'$ then

$$
\begin{aligned}
h(S', H) &= h(S', \pi \cdot H) \text{ (by } \pi \cdot H = H) \\
&= h(S, H) \text{ (by } k\text{-permutation equivariance of } h \text{ and } \pi(S) = S')
\end{aligned}
\tag{40}
$$

$\square$

We revisit the definition of the symmetry group of a $k$-node representation map on hypergraph $\mathcal{H}$.

**Definition B.6.** *For $h : [\mathcal{V}]^k \times \mathbb{Z}_2^{n \times 2^n} \to \mathbb{R}^d$ a $k$-node representation on a hypergraph $\mathcal{H}$,*

$$
Sym(h) \triangleq \{\pi \in Sym(\mathcal{V}) : \pi(S) = S' \Rightarrow h(S, H) = h(S', H)\}
\tag{41}
$$

## B.2 Properties of GWL-1

Here are the steps of the GWL-1 algorithm on the star expansion matrix $H$ is repeated here for convenience:

$$f_e^0 \leftarrow \{\}, h_v^0 \leftarrow \{\}$$
$$f_e^{i+1} \leftarrow \{\!\!\{(f_e^i, h_v^i)\}\!\!\}_{v \in e}, \forall e \in \mathcal{E}(H) \tag{42}$$
$$h_v^{i+1} \leftarrow \{\!\!\{(h_v^i, f_e^{i+1})\}\!\!\}_{v \in e}, \forall v \in \mathcal{V}(H)$$

Where $\mathcal{E}(H)$ denotes the nonzero columns of $H$ and $\mathcal{V}(H)$ denotes the rows of $H$.

We make the following observations about each of the two steps of the GWL-1 algorithm:

**Observation 2.**

$$\{\!\!\{(f_e^i, h_v^i)\}\!\!\}_{v \in e} = \{\!\!\{(f'^i_e, h'^i_v)\}\!\!\}_{v \in e} \; \textit{iff} \; (f_e^i, \{\!\!\{h_v^i\}\!\!\}_{v \in e}) = (f'^i_e, \{\!\!\{h'^i_v\}\!\!\}_{v \in e}) \forall e \in \mathcal{E}(H) \; \textit{and} \tag{43a}$$

$$\{\!\!\{(h_v^i, f_e^{i+1}\}\!\!\}_{v \in e} = \{\!\!\{(h'^i_v, f'^{i+1}_e\}\!\!\}_{v \in e} \; \textit{iff} \; (h_v^i, \{\!\!\{f_e^{i+1}\}\!\!\}_{v \in e}) = (h'^i_v, \{\!\!\{f'^{i+1}_e\}\!\!\}_{v \in e}) \forall v \in \mathcal{V}(H) \tag{43b}$$

*Proof.* Equation 43a follows since

$$\{\!\!\{(f_e^i, h_v^i)\}\!\!\}_{v \in e} = \{\!\!\{(f'^i_e, h'^i_v)\}\!\!\}_{v \in e} \forall e \in \mathcal{E}(H) \tag{44a}$$

$$\textit{iff} \; f_e^i = f'^i_e \; \textit{and} \; \{\!\!\{h_v^i\}\!\!\}_{v \in e} = \{\!\!\{h'^i_v\}\!\!\}_{v \in e} \forall e \in \mathcal{E}(H) \tag{44b}$$

$$\textit{iff} \; (f_e^i, \{\!\!\{h_v^i\}\!\!\}_{v \in e}) = (f'^i_e, \{\!\!\{h'^i_v\}\!\!\}_{v \in e}) \forall e \in \mathcal{E}(H) \tag{44c}$$

For Equation 43b, we have:

$$\{\!\!\{(h_v^i, f_e^{i+1}\}\!\!\}_{v \in e} = \{\!\!\{(h'^i_v, f'^{i+1}_e\}\!\!\}_{v \in e} \forall v \in \mathcal{V}(H) \tag{45a}$$

$$\textit{iff} \; \{\!\!\{(h_v^i, \{\!\!\{(f_e^i, h_u^i)\}\!\!\}_{u \in e})\}\!\!\}_{v \in e} = \{\!\!\{(h'^i_v, \{\!\!\{(f'^i_e, h'^i_u)\}\!\!\}_{u \in e})\}\!\!\}_{v \in e} \forall v \in \mathcal{V}(H) \tag{45b}$$

$$\textit{iff} \; h_v^i = h'^i_v \; \textit{and} \; \{\!\!\{(f_e^i, h_u^i)\}\!\!\}_{u \in e, v \in e} = \{\!\!\{(f'^i_e, h'^i_u)\}\!\!\}_{u \in e, v \in e} \forall v \in \mathcal{V}(H) \tag{45c}$$

$$\textit{iff} \; h_v^i = h'^i_v \; \textit{and} \; \{\!\!\{f_e^{i+1}\}\!\!\} = \{\!\!\{f'^{i+1}_e\}\!\!\} \forall v \in \mathcal{V}(H) \tag{45d}$$

These follow by the definition of multiset equality and since there is no loss of information upon factoring out a constant tuple entry of each pair in the multisets. $\square$

**Proposition B.3.** *The update steps of GWL-1:* $f^i(H) \triangleq [f_{e_1}^i(H), \cdots, f_{e_m}^i(H)]$ *and* $h^i(H) \triangleq [h_{v_1}^i(H), \cdots, h_{v_n}^i(H)]$, *are permutation equivariant; in other words, For any* $\pi \in Sym(\mathcal{V})$, *let* $\pi \cdot f^i(H) \triangleq [f_{\pi^{-1}(e_1)}^i(H), \cdots, f_{\pi^{-1}(e_m)}^i(H)]$ *and* $\pi \cdot h^i(H) \triangleq [h_{\pi^{-1}(v_1)}^i(H), \cdots, h_{\pi^{-1}(v_n)}^i(H)]$, *we have* $\forall i \in \mathbb{N}, \pi \cdot f^i(H) = f^i(\pi \cdot H)$ *and* $\pi \cdot h^i(H) = h^i(\pi \cdot H)$

*Proof.* We prove by induction on $i$:

Base case, $i = 0$:

$[\pi \cdot f^0(H)]_{e=\{v_1,\dots,v_k\}} = \{\} = f_{\pi^{-1}(e)=\{\pi^{-1}(v_1),\dots,\pi^{-1}(v_k)\}}^0(H) = f_e^0(\pi \cdot H)$ since the $\pi$ cannot affect a list of empty sets and the definition of the action of $\pi$ on $H$ as defined in Equation 38.

$[\pi \cdot h^0(H)]_v = [\pi \cdot X]_v = X_{\pi^{-1}(v)} = h_{\pi^{-1}(v)}^0(H) = h_v^0(\pi \cdot H)$ by definition of the group action $Sym(\mathcal{V})$ acting on the node indices of a node attribute tensor as defined in Equation 38.

Induction Hypothesis:

$$[\pi \cdot f^i(H)]_e = f_{\pi^{-1}(e)}^i(H) = f_e^i(\pi \cdot H) \text{ and } [\pi \cdot h^i(H)]_v = h_{\pi^{-1}(v)}^i(H) = h_v^i(\pi \cdot H) \tag{46}$$

Induction Step:

$$[\pi \cdot h^{i+1}(H)]_v = \{\!\{([\pi \cdot h^i(H)]_v, [\pi \cdot f^{i+1}(H)]_e)\}\!\}_{v \in e}$$
$$= \{\!\{([\pi \cdot h^i(H)]_v, \{\!\{([\pi \cdot f^i(H)]_e, [\pi \cdot h^i(H)]_u)\}\!\}_{u \in e})\}\!\}_{v \in e}$$
$$= \{\!\{h_v^i(\pi \cdot H), \{\!\{(f_e^i(\pi \cdot H), h_u^i(\pi \cdot H))\}\!\}_{u \in e}\}\!\}_{v \in e} \tag{47}$$
$$= h_v^{i+1}(\pi \cdot H)$$

$$[\pi \cdot f^{i+1}(H)]_e = \{\!\{([\pi \cdot f^i(H)]_e, [\pi \cdot h^i(H)]_v)\}\!\}_{v \in e}$$
$$= \{\!\{(f_e^i(\pi \cdot H), h_v^i(\pi \cdot H)\}\!\}_{v \in e} \tag{48}$$
$$= f_e^{i+1}(\pi \cdot H)$$

$\square$

**Definition B.7.** *Let $h : [\mathcal{V}]^k \times \mathbb{Z}_2^{n \times 2^n} \to \mathbb{R}^d$ be a k-node representation on a hypergraph $\mathcal{H}$. Let $H \in \mathbb{Z}_2^{n \times 2^n}$ be the star expansion adjacency matrix of $\mathcal{H}$ for n nodes. The representation h is k-node most expressive if $\forall S, S' \subseteq \mathcal{V}, |S| = |S'| = k$, the following two conditions are satisfied:*

  1. *h is **k-node invariant**: $\exists \pi \in Stab(H), \pi(S) = S' \implies h(S, H) = h(S', H)$*

  2. *h is **k-node expressive** $\nexists \pi \in Stab(H), \pi(S) = S' \implies h(S, H) \neq h(S', H)$*

Let $AGG$ be a permutation invariant map from a set of node representations to $\mathbb{R}^d$.

**Proposition B.4.** *Let $h(S, H) = AGG_{v \in S}[h_v^i(H)]$ with injective AGG and $h_v^i$ permutation equivariant. The representation $h(S, H)$ is k-node invariant but not necessarily k-node expressive for S a set of k nodes.*

*Proof.* $\exists \pi \in Stab(H)$ s.t. $\pi(S) = S', \pi \cdot H = H$

$\Rightarrow \pi(v_i) = v_i'$ for $i = 1, ..., |S|, \pi \cdot H = H$

$\Rightarrow h_{\pi(v)}^i(H) = h_v^i(\pi \cdot H) = h_v^i(H)$ (By permutation equivariance of $h_v^i$ and $\pi \cdot H = H$)

$\Rightarrow AGG_{v \in S}[h_v^i(H)] = AGG_{v' \in S'}[h_{v'}^i(H)]$ (By Proposition B.2 and AGG being permutation invariant)

The converse, that $h(S, H)$ is k-node expressive, is not necessarily true since we cannot guarantee $h(S, H) = h(S', H)$ implies the existence of a permutation that maps $S$ to $S'$ (see Zhang et al. (2021)). $\square$

A hypergraph can be represented by a bipartite graph $\mathcal{B}_{\mathcal{V}, \mathcal{E}}$ from $\mathcal{V}$ to $\mathcal{E}$ where there is an edge $(v, e)$ in the bipartite graph iff node $v$ is incident to hyperedge $e$. This bipartite graph $\mathcal{B}_{\mathcal{V}, \mathcal{E}}$ is called the star expansion bipartite graph.

We introduce a more structured version of graph isomorphism called a 2-color isomorphism to characterize hypergraphs. It is a map on 2-colored graphs, which are graphs that can be colored with two colors so that no two nodes in any graph with the same color are connected by an edge. We define a 2-colored isomorphism formally here:

**Definition B.8.** *A 2-colored isomorphism is a graph isomorphism on two 2-colored graphs that preserves node colors. In particular, between two graphs $G_1$ and $G_2$ the vertices of one color in $G_1$ must map to vertices of the same color in $G_2$. It is denoted by $\cong_c$.*

A bipartite graph must always have a 2-coloring. In fact, the 2-coloring with all the nodes in the node bipartition colored red and all the nodes in the hyperedge bipartition colored blue forms a canonical 2-coloring of $\mathcal{B}_{\mathcal{V}, \mathcal{E}}$. Assume that all star expansion bipartite graphs are canonically 2-colored.

**Proposition B.5.** *We have two hypergraphs $(\mathcal{V}_1, \mathcal{E}_1) \cong (\mathcal{V}_2, \mathcal{E}_2)$ iff $\mathcal{B}_{\mathcal{V}_1, \mathcal{E}_1} \cong_c \mathcal{B}_{\mathcal{V}_2, \mathcal{E}_2}$ where $\mathcal{B}_{\mathcal{V}, \mathcal{E}}$ is the star expansion bipartite graph of $(\mathcal{V}, \mathcal{E})$*

*Proof.* Denote $L(\mathcal{B}_{\mathcal{V}_i,\mathcal{E}_i})$ as the left hand (red) bipartition of $\mathcal{B}_{\mathcal{V}_i,\mathcal{E}_i}$ to represent the nodes $\mathcal{V}_i$ of $(\mathcal{V}_i,\mathcal{E}_i)$ and $R(\mathcal{B}_{\mathcal{V}_i,\mathcal{E}_i})$ as the right hand (blue) bipartition of $\mathcal{B}_{\mathcal{V}_i,\mathcal{E}_i}$ to represent the hyperedges $\mathcal{E}_i$ of $(\mathcal{V}_i,\mathcal{E}_i)$. We use the left/right bipartition and $\mathcal{V}_i/\mathcal{E}_i$ interchangeably since they are in bijection.

$\Rightarrow$ If there is an isomorphism $\pi : \mathcal{V}_1 \to \mathcal{V}_2$, this means

- $\pi$ is a bijection and

- has the structure preserving property that $(u_1, ..., u_k) \in \mathcal{E}_1$ iff $(\pi(u_1), ..., \pi(u_k)) \in \mathcal{E}_2$.

We may induce a 2-colored isomorphism $\pi^* : \mathcal{V}(\mathcal{B}_{\mathcal{V}_1,\mathcal{E}_1}) \to \mathcal{V}(\mathcal{B}_{\mathcal{V}_1,\mathcal{E}_1})$ so that $\pi^*|_{L(\mathcal{B}_{\mathcal{V}_1,\mathcal{E}_1})} = \pi$ where equality here means that $\pi^*|_{L(\mathcal{B}_{\mathcal{V}_1,\mathcal{E}_1})}$ acts on $L(\mathcal{B}_{\mathcal{V}_1,\mathcal{E}_1})$ the same way that $\pi$ does on $\mathcal{V}_1$. Furthermore $\pi^*$ has the property that $\pi^*|_{R(\mathcal{B}_{\mathcal{V}_1,\mathcal{E}_1})}(u_1, ..., u_k) = (\pi(u_1), ..., \pi(u_k)), \forall (u_1, ..., u_k) \in \mathcal{E}_1$, following the structure preserving property of isomorphism $\pi$.

The map $\pi^*$ is a bijection by definition of being an extension of a bijection.

The map $\pi^*$ is also a 2-colored map since it maps $L(\mathcal{B}_{\mathcal{V}_1,\mathcal{E}_1})$ to $L(\mathcal{B}_{\mathcal{V}_2,\mathcal{E}_2})$ and $R(\mathcal{B}_{\mathcal{V}_1,\mathcal{E}_1})$ to $R(\mathcal{B}_{\mathcal{V}_2,\mathcal{E}_2})$.

We can also check that the map is structure preserving and thus a 2-colored isomorphism since $(u_i, (u_1, ..., u_i, ..., u_k)) \in \mathcal{E}(\mathcal{B}_{\mathcal{V}_1,\mathcal{E}_1}), \forall i = 1, ..., k$ iff $(u_i \in \mathcal{V}_1$ and $(u_1, ..., u_i, ..., u_k) \in \mathcal{E}_1)$ iff $\pi(u_i) \in \mathcal{V}_2$ and $(\pi(u_1), ..., \pi(u_i), ..., \pi(u_k)) \in \mathcal{E}_2$ iff $(\pi^*(u_i), (\pi^*(u_1, ..., u_i, ..., u_k)) \in \mathcal{E}(\mathcal{B}_{\mathcal{V}_2,\mathcal{E}_2}), \forall i = 1, ..., k$. This follows from $\pi$ being structure preserving and the definition of $\pi^*$.

$\Leftarrow$ If there is a 2-colored isomorphism $\pi^* : \mathcal{B}_{\mathcal{V}_1,\mathcal{E}_1} \to \mathcal{B}_{\mathcal{V}_2,\mathcal{E}_2}$ then it has the properties that

- $\pi^*$ is a bijection,

- (is 2-colored): $\pi^*|_{L(\mathcal{B}_{\mathcal{V}_1,\mathcal{E}_1})} : L(\mathcal{B}_{\mathcal{V}_1,\mathcal{E}_1}) \to L(\mathcal{B}_{\mathcal{V}_2,\mathcal{E}_2})$ and $\pi^*|_{R(\mathcal{B}_{\mathcal{V}_1,\mathcal{E}_1})} : R(\mathcal{B}_{\mathcal{V}_1,\mathcal{E}_1}) \to R(\mathcal{B}_{\mathcal{V}_2,\mathcal{E}_2})$

- (it is structure preserving): $(u_i, (u_1, ..., u_i, ..., u_k)) \in \mathcal{E}(\mathcal{B}_{\mathcal{V}_1,\mathcal{E}_1}), \forall i = 1, ..., k$ iff $(\pi^*(u_i), \pi^*(u_1, ..., u_i, ..., u_k)) \in \mathcal{E}(\mathcal{B}_{\mathcal{V}_2,\mathcal{E}_2}), \forall i = 1, ..., k$ .

This then means that we may induce a $\pi : \mathcal{V}_1 \to \mathcal{V}_2$ so that $\pi = \pi^*|_{L(\mathcal{B}_{\mathcal{V}_1,\mathcal{E}_1})}$.

We can check that $\pi$ is a bijection since $\pi$ is the 2-colored bijection $\pi^*$ restricted to $L(\mathcal{B}_{\mathcal{V}_1,\mathcal{E}_1})$, thus remaining a bijection.

We can also check that $\pi$ is structure preserving. This means that $(u_1, ..., u_k) \in \mathcal{E}_1$ iff $(u_i, (u_1, ..., u_i, ..., u_k)) \in \mathcal{E}(\mathcal{B}_{\mathcal{V}_1,\mathcal{E}_1}) \forall i = 1, ..., k$ iff $(\pi^*(u_i), (\pi^*(u_1, ..., u_i, ..., u_k))) \in \mathcal{E}(\mathcal{B}_{\mathcal{V}_2,\mathcal{E}_2}) \forall i = 1, ..., k$ iff $(\pi^*(u_1, ..., u_k)) \in R(\mathcal{B}_{\mathcal{V}_2,\mathcal{E}_2})$ iff $(\pi(u_1), ..., \pi(u_k)) \in \mathcal{E}_2$ □

We define a topological object for a graph originally from algebraic topology called a universal cover:

**Definition B.9.** *(Hatcher (2005)) A universal covering of a connected graph $G$ is a (potentially infinite) graph $\tilde{G}$, s.t. there is a map $p_G : \tilde{G} \to G$ called the universal covering map where:*

*1. $\forall x \in \mathcal{V}(\tilde{G})$, $p_G|_{N(x)}$ is an isomorphism onto $N(p_G(x))$.*

*2. $\tilde{G}$ is simply connected (a tree)*

A covering graph is a graph that satisfies property 1 but not necessarily property 2 in Definition B.9. It is known that a universal covering $\tilde{G}$ covers all the graph covers of the graph $G$. Let $T_x^r$ denote a tree with root $x$ where every node has depth $r$. Furthermore, define a rooted isomorphism $G_x \cong H_y$ as an isomorphism between graphs $G$ and $H$ that maps $x$ to $y$ and vice versa. We will use the following result to prove a characterization of GWL-1:

**Lemma B.6** (Krebs & Verbitsky (2015))**.** *Let $T$ and $S$ be trees and $x \in V(T)$ and $y \in V(S)$ be their vertices of the same degree with neighborhoods $N(x) = \{x_1, ..., x_k\}$ and $N(y) = \{y_1, ..., y_k\}$. Let $r \geq 1$. Suppose that $T_x^{r-1} \cong S_y^{r-1}$ and $T_{x_i}^r \cong S_{y_i}^r$ for all $i \leq k$. Then $T_x^{r+1} \cong S_y^{r+1}$.*

A universal cover of a 2-colored bipartite graph is still 2 colored. When we lift nodes $v$ and hyperedge nodes $e$ to their universal cover, we keep their respective red and blue colors.

Define a rooted colored isomorphism $T_{\tilde{e}_1}^k \cong_c T_{\tilde{e}_2}^k$ as a colored tree isomorphism where blue/red node $\tilde{e}_1/\tilde{v}_1$ maps to blue/red node $\tilde{e}_2/\tilde{v}_2$ and vice versa.

In fact, Lemma B.6 holds for 2-colored isomorphisms, which we show below:

**Lemma B.7.** *Let $T$ and $S$ be 2-colored trees and $x \in V(T)$ and $y \in V(S)$ be their vertices of the same degree with neighborhoods $N(x) = \{x_1, ..., x_k\}$ and $N(y) = \{y_1, ..., y_k\}$. Let $r \geq 1$. Suppose that $T_x^{r-1} \cong_c S_y^{r-1}$ and $T_{x_i}^r \cong_c S_{y_i}^r$ for all $i \leq k$. Then $T_x^{r+1} \cong_c S_y^{r+1}$.*

*Proof.* Certainly 2-colored isomorphisms are rooted isomorphisms on 2-colored trees. The converse is true if the roots match in color since recursively all descendants of the root must match in color.

If $T_x^{r-1} \cong_c S_y^{r-1}$ and $T_{x_i}^r \cong_c S_{y_i}^r$ for all $i \leq k$ and $N(x) = \{x_1, ..., x_k\}, N(y) = \{y_1..y_k\}$, the roots $x$ and $y$ must match in color. The neighborhoods $N(x)$ and $N(y)$ then must both be of the opposing color. Since rooted colored isomorphisms are rooted isomorphisms, we must have $T_x^{r-1} \cong S_y^{r-1}$ and $T_{x_i}^r \cong S_{y_i}^r$ for all $i \leq k$. By Lemma B.6, we have $T_x^{r+1} \cong S_y^{r+1}$. Once the roots match in color, a rooted tree isomorphism is the same as a rooted 2-colored tree isomorphism. Thus, since $x$ and $y$ share the same color, $T_x^{r+1} \cong_c S_y^{r+1}$ $\square$

**Theorem B.8.** *Let $\mathcal{H}_1 = (\mathcal{V}_1, \mathcal{E}_1)$ and $\mathcal{H}_2 = (\mathcal{V}_2, \mathcal{E}_2)$ be two connected hypergraphs. Let $\mathcal{B}_{\mathcal{V}_1, \mathcal{E}_1}$ and $\mathcal{B}_{\mathcal{V}_2, \mathcal{E}_2}$ be two canonically colored bipartite graphs for $\mathcal{H}_1$ and $\mathcal{H}_2$ (vertices colored red and hyperedges colored blue)*

*For any $i \in \mathbb{Z}^+$, for any of the nodes $x_1 \in \mathcal{B}_{\mathcal{V}_1}, e_1 \in \mathcal{B}_{\mathcal{V}_1, \mathcal{E}_1}$ and $x_2 \in \mathcal{B}_{\mathcal{V}_1}, e_2 \in \mathcal{B}_{\mathcal{V}_2, \mathcal{E}_2}$:*

- *$(\tilde{\mathcal{B}}_{\mathcal{V}_1, \mathcal{E}_1}^{2i-1})_{\tilde{e}_1} \cong_c (\tilde{\mathcal{B}}_{\mathcal{V}_2, \mathcal{E}_2}^{2i-1})_{\tilde{e}_2}$ iff $f_{e_1}^i = f_{e_2}^i$*

- *$(\tilde{\mathcal{B}}_{\mathcal{V}_1, \mathcal{E}_1}^{2i})_{\tilde{x}_1} \cong_c (\tilde{\mathcal{B}}_{\mathcal{V}_2, \mathcal{E}_2}^{2i})_{\tilde{x}_2}$ iff $h_{x_1}^i = h_{x_2}^i$,*

*with $f_\bullet^i, h_\bullet^i$ the ith GWL-1 values for the hyperedges and nodes respectively where $e_1 = p_{\mathcal{B}_{\mathcal{V}_1, \mathcal{E}_1}}(\tilde{e}_1)$, $x_1 = p_{\mathcal{B}_{\mathcal{V}_1, \mathcal{E}_1}}(\tilde{x}_1)$, $e_2 = p_{\mathcal{B}_{\mathcal{V}_1, \mathcal{E}_1}}(\tilde{e}_2)$, $x_2 = p_{\mathcal{B}_{\mathcal{V}_1, \mathcal{E}_1}}(\tilde{x}_2)$. The maps $p_{\mathcal{B}_{\mathcal{V}_1, \mathcal{E}_1}} : \tilde{\mathcal{B}}_{\mathcal{V}_1, \mathcal{E}_1} \to \mathcal{B}_{\mathcal{V}_1, \mathcal{E}_1}, p_{\mathcal{B}_{\mathcal{V}_2, \mathcal{E}_2}} : \tilde{\mathcal{B}}_{\mathcal{V}_2, \mathcal{E}_2} \to \mathcal{B}_{\mathcal{V}_2, \mathcal{E}_2}$ are the universal covering maps of $\mathcal{B}_{\mathcal{V}_1, \mathcal{E}_1}$ and $\mathcal{B}_{\mathcal{V}_2, \mathcal{E}_2}$ respectively.*

*Proof.* We prove by induction:

Let $T_{\tilde{e}_1}^k := (\tilde{\mathcal{B}}_{\mathcal{V}_1, \mathcal{E}_1}^k)_{\tilde{e}_1}$ where $\tilde{e}_1$ is a pullback of a hyperedge, meaning $p_{\mathcal{B}_{\mathcal{V}_1, \mathcal{E}_2}}(\tilde{e}_1) = e_1$. Similarly, let $T_{\tilde{e}_2}^k := (\tilde{\mathcal{B}}_{\mathcal{V}_2, \mathcal{E}_2}^k)_{\tilde{e}_2}$, $T_{\tilde{x}_1}^k := (\tilde{\mathcal{B}}_{\mathcal{V}_1, \mathcal{E}_1}^k)_{\tilde{x}_1}$, $T_{\tilde{x}_2}^k := (\tilde{\mathcal{B}}_{\mathcal{V}_2, \mathcal{E}_2}^k)_{\tilde{x}_2}$, $\forall k \in \mathbb{N}$, where $\tilde{e}_1, \tilde{e}_2, \tilde{x}_1, \tilde{x}_2$ are the respective pullbacks of $e_1, e_2, x_1, x_2$.

Define an (2-colored) isomorphism of multisets of graphs to mean that there exists a bijection between the two multisets so that each graph in one multiset is (2-colored) isomorphic with exactly one other element in the other multiset.

By Observation 2 we can rewrite GWL-1 as:

$$f_e^0 \leftarrow \{\}, h_v^0 \leftarrow \{\} \tag{49}$$

$$f_e^{i+1} \leftarrow (f_e^i, \{\!\{h_v^i\}\!\}_{v \in e}) \forall e \in \mathcal{E}_\mathcal{H} \tag{50}$$

$$h_v^{i+1} \leftarrow (h_v^i, \{\!\{f_e^{i+1}\}\!\}_{v \in e}) \forall v \in \mathcal{V}_\mathcal{H} \tag{51}$$

Base Case $i = 1$:

$$T_{\tilde{e}_1}^1 \cong_c T_{\tilde{e}_2}^1 \text{ iff } (T_{\tilde{e}_1}^0 \cong_c T_{\tilde{e}_2}^0 \text{ and } \{\!\{T_{\tilde{x}_1}^0\}\!\}_{\tilde{x}_1 \in N(\tilde{e}_1)} \cong_c \{\!\{T_{\tilde{x}_2}^0\}\!\}_{\tilde{x}_2 \in N(\tilde{e}_2)}) \text{ (By Lemma B.7)} \tag{52a}$$

$$\text{iff } (f_{e_1}^0 = f_{e_2}^0 \text{ and } \{\!\{h_{x_1}^0\}\!\} = \{\!\{h_{x_2}^0\}\!\}) \text{ (By Equation 49)} \tag{52b}$$

$$\text{iff } f_{e_1}^1 = f_{e_2}^1 \text{ (By Equation 50)} \tag{52c}$$

$$T_{\tilde{x}_1}^2 \cong_c T_{\tilde{x}_2}^2 \text{ iff } (T_{\tilde{x}_1}^0 \cong_c T_{\tilde{x}_2}^0 \text{ and } \{\!\{T_{\tilde{e}_1}^1\}\!\}_{\tilde{e}_1 \in N(\tilde{x}_1)} \cong_c \{\!\{T_{\tilde{e}_2}^1\}\!\}_{\tilde{e}_2 \in N(\tilde{x}_2)}) \text{ (By Lemma B.7)} \tag{53a}$$

$$\text{iff } (h_{e_1}^0 = h_{e_2}^0 \text{ and } \{\!\{f_{x_1}^1\}\!\} = \{\!\{f_{x_2}^1\}\!\}) \text{ (By Equation 49)} \tag{53b}$$

$$\text{iff } f_{e_1}^1 = f_{e_2}^1 \text{ (By Equation 51)} \tag{53c}$$

Induction Hypothesis: For $i \geq 1$, $T_{\tilde{e}_1}^{2i-1} \cong_c T_{\tilde{e}_2}^{2i-1}$ iff $f_{e_1}^i = f_{e_2}^i$ and $T_{\tilde{x}_1}^{2i} \cong_c T_{\tilde{x}_2}^{2i}$ iff $h_{x_1}^i = h_{x_2}^i$

Induction Step:

$$T_{\tilde{e}_1}^{2i+1} \cong_c T_{\tilde{e}_2}^{2i+1} \text{ iff } (T_{\tilde{e}_1}^{2i-1} \cong_c T_{\tilde{e}_2}^{2i-1} \text{ and } \{\!\{T_{\tilde{x}_1}^{2i}\}\!\}_{\tilde{x}_1 \in N(\tilde{e}_1)} \cong_c \{\!\{T_{\tilde{x}_2}^{2i}\}\!\}_{\tilde{x}_2 \in N(\tilde{e}_2)}) \text{ (By Lemma B.7)} \tag{54a}$$

$$\text{iff } (f_{e_1}^i = f_{e_2}^i \text{ and } \{\!\{h_{x_1}^i\}\!\} = \{\!\{h_{x_2}^i\}\!\}) \text{ (By Induction Hypothesis)} \tag{54b}$$

$$\text{iff } f_{e_1}^{i+1} = f_{e_2}^{i+1} \text{ (By Equation 50)} \tag{54c}$$

$$T_{\tilde{x}_1}^{2i} \cong_c T_{\tilde{x}_2}^{2i} \text{ iff } (T_{\tilde{x}_1}^{2i-2} \cong_c T_{\tilde{x}_2}^{2i-2} \text{ and } \{\!\{T_{\tilde{e}_1}^{2i-1}\}\!\}_{\tilde{e}_1 \in N(\tilde{x}_1)} \cong_c \{\!\{T_{\tilde{e}_2}^{2i-1}\}\!\}_{\tilde{e}_2 \in N(\tilde{x}_2)}) \text{ (By Lemma B.7)} \tag{55a}$$

$$\text{iff } (h_{e_1}^i = h_{e_2}^i \text{ and } \{\!\{f_{x_1}^i\}\!\} = \{\!\{f_{x_2}^i\}\!\}) \text{ (By Equation 49)} \tag{55b}$$

$$\text{iff } h_{x_1}^i = h_{x_2}^i \text{ (By Equation 51)} \tag{55c}$$

$\square$

We write here the theorem characterizing the WL-1 algorithm on a graph by the graph's universal cover.

**Theorem B.9.** *[Krebs & Verbitsky (2015)] Let $G$ and $H$ be two connected graphs. Let $p_G : \tilde{G} \to G, p_H : \tilde{H} \to H$ be the universal covering maps of $G$ and $H$ respectively. For any $i \in \mathbb{N}$, for any two nodes $x \in G$ and $y \in H$: $\tilde{G}_{\tilde{x}}^i \cong \tilde{G}_{\tilde{y}}^i$ iff the WL-1 algorithm assigns the same value to nodes $x = p_G(\tilde{x})$ and $y = p_H(\tilde{y})$.*

It follows immediately from Theorem B.9 that the WL-1 algorithm on colored star expansion bipartite graphs of two hypergraphs corresponds to constructing their universal covers.

**Corollary 2.** *Let $\mathcal{H}_1 = (\mathcal{V}_1, \mathcal{E}_1)$ and $\mathcal{H}_2 = (\mathcal{V}_2, \mathcal{E}_2)$ be two connected hypergraphs. Let $\mathcal{B}_{\mathcal{V}_1, \mathcal{E}_1}$ and $\mathcal{B}_{\mathcal{V}_2, \mathcal{E}_2}$ be two canonically colored bipartite graphs for $\mathcal{H}_1$ and $\mathcal{H}_2$ (vertices colored red and hyperedges colored blue). Let $p_{\mathcal{B}_{\mathcal{V}_1, \mathcal{E}_1}} : \tilde{\mathcal{B}}_{\mathcal{V}_1, \mathcal{E}_1} \to \mathcal{B}_{\mathcal{V}_1, \mathcal{E}_1}, p_{\mathcal{B}_{\mathcal{V}_2, \mathcal{E}_2}} : \tilde{\mathcal{B}}_{\mathcal{V}_2, \mathcal{E}_2} \to \mathcal{B}_{\mathcal{V}_2, \mathcal{E}_2}$ be the universal coverings of $\mathcal{B}_{\mathcal{V}_1, \mathcal{E}_1}$ and $\mathcal{B}_{\mathcal{V}_2, \mathcal{E}_2}$ respectively. For any $i \in \mathbb{Z}^+$,*

- *$(\tilde{\mathcal{B}}_{\mathcal{V}_1, \mathcal{E}_1}^{2i-1})_{\tilde{v}_1} \cong_c (\tilde{\mathcal{B}}_{\mathcal{V}_2, \mathcal{E}_2}^{2i-1})_{\tilde{v}_2}$ iff $g_{v_1}^i = g_{v_2}^i$*

*with $g_{v_1}^i, g_{v_2}^i$ the $i$-th WL-1 values for $v_1, v_2 \in \mathcal{V}(\mathcal{B}_{\mathcal{V}_1, \mathcal{E}_1}), \mathcal{V}(\mathcal{B}_{\mathcal{V}_2, \mathcal{E}_2})$ respectively where $v_1 = p_{\mathcal{B}_{\mathcal{V}_1, \mathcal{E}_1}}(\tilde{v}_1)$, $v_2 = p_{\mathcal{B}_{\mathcal{V}_2, \mathcal{E}_2}}(\tilde{v}_2)$.*

*Furthermore, for $i \geq 2$,*

- *$g_{u_1}^{2+i} = g_{u_2}^{2+i}$ iff $h_{u_1}^{1+i} = h_{u_2}^{1+i}, \forall u_1 \in L(\mathcal{B}_{\mathcal{V}_1, \mathcal{E}_1}), \forall u_2 \in L(\mathcal{B}_{\mathcal{V}_2, \mathcal{E}_2})$ and*

- *$g_{e_1}^{2+i} = g_{e_2}^{2+i}$ iff $f_{e_1}^{2+i} = h_{e_2}^{2+i}, \forall e_1 \in R(\mathcal{B}_{\mathcal{V}_1, \mathcal{E}_1}), \forall e_2 \in R(\mathcal{B}_{\mathcal{V}_2, \mathcal{E}_2})$*

*Proof.* The first equivalence $(\tilde{\mathcal{B}}_{\mathcal{V}_1, \mathcal{E}_1}^{2i-1})_{\tilde{v}_1} \cong_c (\tilde{\mathcal{B}}_{\mathcal{V}_2, \mathcal{E}_2}^{2i-1})_{\tilde{v}_2}$ iff $g_{v_1}^i = g_{v_2}^i$ follows directly by Theorem 2.

The successive two equivalences follow by Theorem B.8 and the first equivalence. $\square$

**Observation 3.** *If the node values for nodes $x$ and $y$ from GWL-1 for $i$ iterations on two hypergraphs $\mathcal{H}_1$ and $\mathcal{H}_2$ are the same, then for all $j$ with $0 \leq j \leq i$, the node values for GWL-1 for $j$ iterations on $x$ and $y$ also agree. In particular $\deg(x) = \deg(y)$.*

*Proof.* There is a 2-color isomorphism on subtrees $(\tilde{\mathcal{B}}^j_{\mathcal{V}_1,\mathcal{E}_1})_{\tilde{x}}$ and $(\tilde{\mathcal{B}}^j_{\mathcal{V}_2,\mathcal{E}_2})_{\tilde{y}}$ of the $i$-hop subtrees of the universal covers rooted about nodes $x \in \mathcal{V}_1$ and $y \in \mathcal{V}_2$ for $0 \leq j \leq i$ since $(\tilde{\mathcal{B}}^i_{\mathcal{V}_1,\mathcal{E}_1})_{\tilde{x}} \cong_c (\tilde{\mathcal{B}}^i_{\mathcal{V}_2,\mathcal{E}_2})_{\tilde{y}}$. By Theorem B.8, we have that GWL-1 returns the same value for $x$ and $y$ for each $0 \leq j \leq i$. $\qquad\square$

**Theorem B.10.** *Let $h^L : [\mathcal{V}]^1 \times \mathbb{Z}_2^{n \times 2^n} \to \mathbb{R}^d$ be the $L$-GWL-1 representation of nodes for hypergraph $\mathcal{H}$ in Equation 7, then*

$$Aut(\mathcal{H}) \cong Stab(H) \subseteq Sym(h^L(H)) \cong Aut_c(\tilde{\mathcal{B}}^{2L}_{\mathcal{V},\mathcal{E}}), \forall L \geq 1 \tag{56}$$

*Proof.* 1. $Aut(\mathcal{H}) \cong Stab(H)$ follows by Proposition B.1.

2. $Stab(H) \subseteq Sym(h^L(H))$ follows by definition of the symmetry group of a representation map given in Definition B.6 and the equivariance of $L$-GWL-1 due to Proposition B.3. Let **Set** denote the collection of all finite sets. We know that

$$h^L(S,H) \triangleq AGG(\{h^L_v\}_{v \in S}) \tag{57}$$

for $AGG :$ **Set** $\to \mathbb{R}^d$ an injective set representation map and that $S$ has cardinality 1. For any $\pi \in Stab(H) \subseteq Sym(\mathcal{V})$, we must have $\pi \cdot H = H$ we check that $\pi$ satisfies:

$$\pi(u) = v \Rightarrow h(u,H) = h(v,H), \forall u,v \in \mathcal{V} \tag{58}$$

If $\pi(u) = v$ and $\pi \cdot H = H$, we can use the equivariance of $h^L$ to get right hand necessary condition: $h^L(u,H) = h^L(u,\pi(H)) = h^L(\pi(u),H) = h^L(v,H)$

3. The last group isomorphism follows by the equivalence between $L$-GWL-1 and the universal cover up to $2L$-hops given in Theorem B.8.

For any $\pi \in Sym(h^L(H))$, it must satisfy $\pi(u) = v \Rightarrow h(u,H) = h(v,H), \forall u,v \in \mathcal{V}$. We map each $\pi$ to a 2-colored isomorphism $\Phi : \pi \mapsto \phi_c$ which is the 2-colored isomorphism determined by the Theorem B.8:

$$(\tilde{\mathcal{B}}^{2L}_{\mathcal{V},\mathcal{E}})_{\tilde{u}} \cong_c (\tilde{\mathcal{B}}^{2L}_{\mathcal{V},\mathcal{E}})_{\widetilde{\pi(u))}} \text{ iff } h^L_u = h^L_{\pi(u)} = h^L(u,H) = h^L(\pi(u),H), \forall u \in \mathcal{V} \tag{59}$$

Certainly the map $\Phi : \pi \mapsto \phi_c$ is a homomorphism because:

1. $\Phi$ maps the identity to identity:

$$(\tilde{\mathcal{B}}^{2L}_{\mathcal{V},\mathcal{E}})_{\tilde{u}} \cong_c (\tilde{\mathcal{B}}^{2L}_{\mathcal{V},\mathcal{E}})_{\tilde{u})} \text{ iff } h^L_u = h^L(u,H), \forall u \in \mathcal{V} \tag{60}$$

can have only one 2-colored isomorphism determining $(\tilde{\mathcal{B}}^{2L}_{\mathcal{V},\mathcal{E}})_{\tilde{u}} \cong_c (\tilde{\mathcal{B}}^{2L}_{\mathcal{V},\mathcal{E}})_{\tilde{u}}$, which is the identity.

2. $\Phi$ perserves composition: $\pi_2 \circ \pi_1 \mapsto (\phi_2)_c \circ (\phi_1)_c$

By definition of $\pi_1$ and $\pi_2$:

$$\pi_1(u) = v \Rightarrow h(u,H) = h(v,H) \text{ iff } (\tilde{\mathcal{B}}^{2L}_{\mathcal{V},\mathcal{E}})_{\tilde{u}} \cong_c (\tilde{\mathcal{B}}^{2L}_{\mathcal{V},\mathcal{E}})_{\widetilde{\pi_1(u)}}, \forall u \in \mathcal{V} \tag{61a}$$

$$\pi_2(v) = w \Rightarrow h(v,H) = h(w,H) \text{ iff } (\tilde{\mathcal{B}}^{2L}_{\mathcal{V},\mathcal{E}})_{\tilde{v}} \cong_c (\tilde{\mathcal{B}}^{2L}_{\mathcal{V},\mathcal{E}})_{\widetilde{\pi_1(w)}}, \forall v \in \mathcal{V} \tag{61b}$$

Combining, we get:

$$(\tilde{\mathcal{B}}^{2L}_{\mathcal{V},\mathcal{E}})_{\tilde{u}} \cong_c (\tilde{\mathcal{B}}^{2L}_{\mathcal{V},\mathcal{E}})_{\widetilde{\pi_1(u)}} \cong_c (\tilde{\mathcal{B}}^{2L}_{\mathcal{V},\mathcal{E}})_{\widetilde{\pi_2(u)}} \cong (\tilde{\mathcal{B}}^{2L}_{\mathcal{V},\mathcal{E}})_{\widetilde{\pi_2 \circ \pi_1(u)}} \tag{62}$$

where the first isomorphism is $(\phi_1)_c$ on $(\tilde{\mathcal{B}}^{2L}_{\mathcal{V},\mathcal{E}})_{\tilde{u}}$, the second isomorphism is from $(\phi_2)_c$ on $(\tilde{\mathcal{B}}^{2L}_{\mathcal{V},\mathcal{E}})_{\tilde{u}}$ and the third isomorphism is $(\phi_1)_c \circ (\phi_1)_c$ on $(\tilde{\mathcal{B}}^{2L}_{\mathcal{V},\mathcal{E}})_{\widetilde{\pi_1(u)}}$ $\qquad\square$

**Proposition B.11.** *If GWL-1 cannot distinguish two connected hypergraphs $H_1$ and $H_2$ then HyperPageRank will not either.*

*Proof.* HyperPageRank is defined on a hypergraph with star expansion matrix $H$ as the following stationary distribution $\Pi$:

$$\lim_{n \to \infty}(D_v^{-1} \cdot H \cdot D_e^{-1} \cdot H^T)^n = \Pi \tag{63}$$

If $H$ is a connected bipartite graph, $\Pi$ must be the eigenvector of $(D_v^{-1} \cdot H \cdot D_e^{-1} \cdot H^T)$ for eigenvalue 1. In other words, $\Pi$ must satisfy

$$(D_v^{-1} \cdot H \cdot D_e^{-1} \cdot H^T) \cdot \Pi = \Pi \tag{64}$$

By Theorem 1 of Huang & Yang (2021), we know that the UniGCN defined by:

$$h_e^{i+1} \leftarrow \phi_2(h_e^i, h_v^i) = W_e \cdot H^T \cdot h_v^i \tag{65a}$$

$$h_v^{i+1} \leftarrow \phi_1(h_v^i, h_e^{i+1}) = W_v \cdot H \cdot h_e^{i+1} \tag{65b}$$

for constant $W_e$ and $W_v$ weight matrices, is equivalent to GWL-1 provided that $\phi_1$ and $\phi_2$ are both injective as functions. Without injectivity, we can only guarantee that if UniGCN distinguishes $H_1, H_2$ then GWL-1 distinguishes $H_1, H_2$. In fact, each matrix power of order $n$ in Equation 63 corresponds to $h_v^n$ so long as we satisfy the following constraints:

$$W_e \leftarrow D_e^{-1}, W_v \leftarrow D_v^{-1} \text{ and } h_v^0 \leftarrow I \tag{66}$$

We show that the matrix powers are UniGCN under the constraints of Equation 66 by induction:

Base Case: $n = 0$: $h_v^0 = I$

Induction Hypothesis: $n > 0$:

$$(D_v^{-1} \cdot H \cdot D_e^{-1} \cdot H^T)^n = h_v^n \tag{67}$$

Induction Step:

$$(D_v^{-1} \cdot H \cdot h_e^n) \tag{68a}$$

$$= (D_v^{-1} \cdot H \cdot ((D_e^{-1} \cdot H^T) \cdot h_v^n)) \tag{68b}$$

$$= (D_v^{-1} \cdot H \cdot D_e^{-1} \cdot H^T) \cdot (D_v^{-1} \cdot H \cdot D_e^{-1} \cdot H^T)^n \tag{68c}$$

$$= (D_v^{-1} \cdot H \cdot D_e^{-1} \cdot H^T)^{n+1} = h_v^{n+1} \tag{68d}$$

Since we cannot guarantee that the maps $\phi_1$ and $\phi_2$ are injective in Equation 68b, it must be that the output $h_v^n$, coming from UniGCN with the constraints of Equation 66, is at most as powerful as GWL-1.

In general, injectivity preserves more information. For example, if $\phi_1$ is injective and if $\phi_1'$ is an arbitrary map (not guaranteed to be injective) then:

$$\phi_1(h_1) = \phi_1(h_2) \Rightarrow h_1 = h_2 \Rightarrow \phi_1'(h_1) = \phi_1'(h_2) \tag{69}$$

HyperpageRank is exactly as powerful as UniGCN under the constraints of Equation 66. Thus HyperPageRank is at most as powerful as GWL-1 in distinguishing power. $\qquad\square$

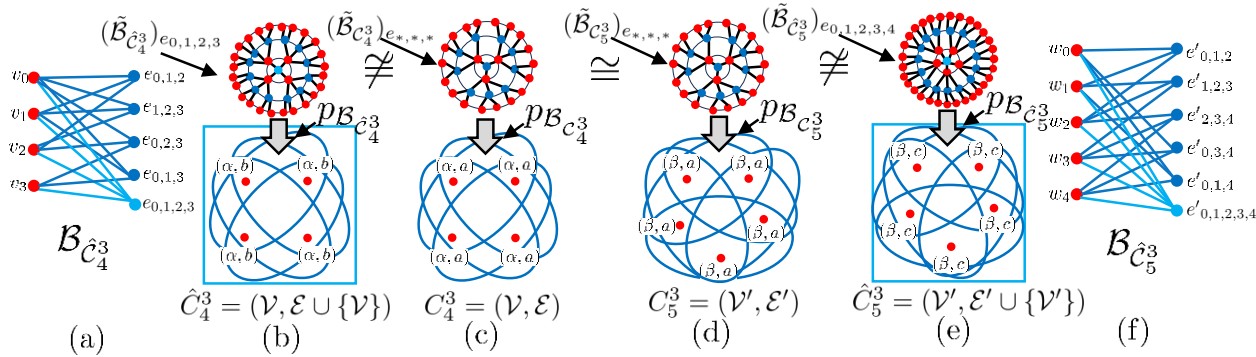

Figure 5: An illustration of hypergraph symmetry breaking. (c,d) 3-regular hypergraphs $C_4^3, C_5^3$ with 4 and 5 nodes respectively and their corresponding universal covers centered at any hyperedge $(\tilde{\mathcal{B}}_{C_4^3})_{e_{*,*,*}}, (\tilde{\mathcal{B}}_{C_5^3})_{e_{*,*,*}}$ with universal covering maps $p_{\mathcal{B}_{C_4^3}}, p_{\mathcal{B}_{C_5^3}}$. (b,e) the hypergraphs $\hat{C}_4^3, \hat{C}_5^3$, which are $C_4^3, C_5^3$ with $4, 5$-sized hyperedges attached to them and their corresponding universal covers and universal covering maps. (a,f) are the corresponding bipartite graphs of $\hat{C}_4^3, \hat{C}_5^3$. (c,d) are indistinguishable by GWL-1 and thus will give identical node values by Theorem B.8. On the other hand, (b,e) gives node values which are now sensitive to the the order of the hypergraphs $4, 5$, also by Theorem B.8.

## B.3 Method

We repeat here from the main text the symmetry finding algorithm:

---

**Algorithm 2:** A Symmetry Finding Algorithm

**Data:** Hypergraph $\mathcal{H} = (\mathcal{V}, \mathcal{E})$, represented by its star expansion matrix $H$. $L \in \mathbb{Z}^+$ is the number of iterations to run GWL-1.

**Result:** A pair of collections: $(\mathcal{R}_V = \{\mathcal{V}_{R_j}\}, \mathcal{R}_E = \cup_j \{\mathcal{E}_{R_j}\})$ where $R_j$ are disconnected subhypergraphs exhibiting symmetry in $\mathcal{H}$ that are indistinguishable by $L$-GWL-1.

1 $E_{deg} \leftarrow \{\{deg(v) : v \in e\} : \forall e \in \mathcal{E}\}$
2 $U_L \leftarrow h_v^L(H); \mathcal{G}_L \leftarrow \{U_L[v] : \forall v \in \mathcal{V}\}$ ;           /* $U_L[v]$ is the $L$-GWL-1 value of node $v \in \mathcal{V}$. */
3 $\mathcal{B}_{\mathcal{V}_\mathcal{H}, \mathcal{E}_\mathcal{H}} \leftarrow Bipartite(\mathcal{H})$ /* Construct the bipartite graph from $\mathcal{H}$. */
4 $\mathcal{R}_V \leftarrow \{\}; \mathcal{R}_E \leftarrow \{\}$
5 **for** $c_L \in \mathcal{G}_L$ **do**
6 $\quad \mathcal{V}_{c_L} \leftarrow \{v \in \mathcal{V} : U_L[v] = c_L\}, \mathcal{E}_{c_L} \leftarrow \{e \in \mathcal{E} : u \in \mathcal{V}_{c_L}, \forall u \in e\}$
7 $\quad \mathcal{C}_{c_L} \leftarrow \text{ConnectedComponents}(\mathcal{H}_{c_L} = (\mathcal{V}_{c_L}, \mathcal{E}_{c_L}))$
8 $\quad$ **for** $\mathcal{R}_{c_L,i} \in \mathcal{C}_{c_L}$ **do**
9 $\quad\quad \mathcal{R}_V \leftarrow \mathcal{R}_V \cup \{\mathcal{V}_{\mathcal{R}_{c_L,i}}\}; \mathcal{R}_E \leftarrow \mathcal{R}_E \cup \mathcal{E}_{\mathcal{R}_{c_L,i}}$
10 $\quad$ **end**
11 **end**
12 **return** $(\mathcal{R}_V, \mathcal{R}_E)$

---

We also repeat here for convenience some definitions used in the proofs. Given a hypergraph $\mathcal{H} = (\mathcal{V}, \mathcal{E})$, let

$$\mathcal{V}_{c_L} := \{v \in \mathcal{V} : c_L = h_v^L(H)\} \tag{70}$$

be the set of nodes of the same class $c_L$ as determined by $L$-GWL-1. Let $\mathcal{H}_{c_L}$ be an induced subgraph of $\mathcal{H}$ by $\mathcal{V}_{c_L}$.

**Definition B.10.** *A $L$-GWL-1 symmetric induced subhypergraph $\mathcal{R} \subset \mathcal{H}$ of $\mathcal{H}$ is a connected induced subhypergraph determined by $\mathcal{V}_\mathcal{R} \subseteq \mathcal{V}_\mathcal{H}$, some subset of nodes that are all indistinguishable amongst each other by $L$-GWL-1:*

$$h_u^L(H) = h_v^L(H), \forall u, v \in \mathcal{V}_\mathcal{R} \tag{71}$$

When $L = \infty$, we call such $\mathcal{R}$ a GWL-1 symmetric induced subhypergraph. Furthermore, if $\mathcal{R} = \mathcal{H}$, then we say $\mathcal{H}$ is GWL-1 symmetric.

**Definition B.11.** *A neighborhood-regular hypergraph is a hypergraph where all neighborhoods of each node are isomorphic to each other.*

**Observation 4.** *A hypergraph $\mathcal{H}$ is GWL-1 symmetric if and only if it is L-GWL-1 symmetric for all $L \geq 1$ if and only if $\mathcal{H}$ is neighborhood regular.*

*Proof.*
**1. First if and only if :**

By Theorem B.8, GWL-1 symmetric hypergraph $\mathcal{H} = (\mathcal{V}, \mathcal{E})$ means that for every pair of nodes $u, v \in \mathcal{V}$, $(\tilde{\mathcal{B}}_{\mathcal{V},\mathcal{E}})_{\tilde{u}} \cong_c (\tilde{\mathcal{B}}_{\mathcal{V},\mathcal{E}})_{\tilde{v}}$. This implies that for any $L \geq 1$, $(\tilde{\mathcal{B}}^{2L}_{\mathcal{V},\mathcal{E}})_{\tilde{u}} \cong_c (\tilde{\mathcal{B}}^{2L}_{\mathcal{V},\mathcal{E}})_{\tilde{v}}$ by restricting the rooted isomorphism to $2L$-hop rooted subtrees, which means that $h^L_u(H) = h^L_v(H)$. The converse is true since $L$ is arbitrary. If there are no cycles, we can just take the isomorphism for the largest. Otherwise, an isomorphism can be constructed for $L = \infty$ by infinite extension.

**2. Second if and only if :**

Let $p_{\mathcal{B}_{\mathcal{V},\mathcal{E}}}$ be the universal covering map for $\mathcal{B}_{\mathcal{V},\mathcal{E}}$. Denote $\tilde{v}, \tilde{u}$ by the lift of some nodes $v, u \in \mathcal{V}$ by $p_{\mathcal{B}_{\mathcal{V},\mathcal{E}}}$.

Let $(\tilde{N}(\tilde{u}))_{\tilde{u}}$ be the rooted bipartite lift of $(N(u))_u$. If $\mathcal{H}$ is $L$-GWL-1 symmetric for all $L \geq 1$ then with $L = 1$, $(\tilde{\mathcal{B}}^2_{\mathcal{V},\mathcal{E}})_{\tilde{u}} \cong_c (\tilde{N}(u))_{\tilde{u}} \cong_c (\tilde{N}(v))_{\tilde{v}} \cong_c (\tilde{\mathcal{B}}^2_{\mathcal{V},\mathcal{E}})_{\tilde{v}}$, iff $(N(u))_u \cong (N(v))_{\tilde{v}}, \forall u, v \in \mathcal{V}$ since $N(u)$ and $N(v)$ are cycle-less for any $u, v \in \mathcal{V}$. For the converse, assume all nodes $v \in \mathcal{V}$ have $(N(v))_v \cong (N^1)_x$ for some 1-hop rooted tree $(N^1)_x$ rooted at node $x$, independent of any $v \in \mathcal{V}$. We prove by induction that for all $L \geq 1$ and for all $v \in \mathcal{V}$, $(\tilde{\mathcal{B}}^{2L}_{\mathcal{V},\mathcal{E}})_{\tilde{v}} \cong_c (\tilde{N}^{2L})_x$ for a $2L$-hop tree $(\tilde{N}^{2L})_x$ rooted at node $x$.

Base case: $L = 1$ is by assumption.

Inductive step: If $(\tilde{\mathcal{B}}^{2L}_{\mathcal{V},\mathcal{E}})_{\tilde{v}} \cong_c (N^{2L})_x$, we can form $(\tilde{\mathcal{B}}^{2L+2}_{\mathcal{V},\mathcal{E}})_{\tilde{v}}$ by attaching $(\tilde{N}(\tilde{u}))_{\tilde{u}}$ to each node $\tilde{u}$ in the $2L$-th layer of $(\tilde{\mathcal{B}}^{2L}_{\mathcal{V},\mathcal{E}})_{\tilde{v}} \cong_c (\tilde{N}^{2L})_x$. Each $(\tilde{N}(u))_{\tilde{u}}$ is independent of the root $\tilde{v}$ since every $u \in \mathcal{V}$ has $(\tilde{N}(\tilde{u}))_{\tilde{u}} \cong_c (\tilde{N}^2)_x$ iff $(N(u))_{\tilde{u}} \cong (N^1)_x$ for an $x$ independent of $u \in \mathcal{V}$. This means $(\tilde{\mathcal{B}}^{2L+2}_{\mathcal{V},\mathcal{E}})_{\tilde{v}} \cong_c (\tilde{N}^{2L+2})_x$ for the same root node $x$ where $(\tilde{N}^{2L+2})_x$ is constructed in the same manner as $(\tilde{\mathcal{B}}^{2L+2}_{\mathcal{V},\mathcal{E}})_{\tilde{v}}, \forall v \in \mathcal{V}$.

$\square$

**Proposition B.12.** *Let $\mathcal{H} = (\mathcal{V}, \tilde{\mathcal{E}})$ be a multi-hypergraph.*

*A multi-hypergraph isomorphism, like for hypergraphs, is defined by a structure preserving map $(\rho_{\mathcal{V}} : \mathcal{V}_{\mathcal{H}} \to \mathcal{V}_{\mathcal{D}}, \rho_{\tilde{\mathcal{E}}} : \mathcal{E}_{\mathcal{H}} \to \mathcal{E}_{\mathcal{D}})$ but where $\rho_{\tilde{\mathcal{E}}}$ is a bijection between multisets.*

*The star expansion bipartite graph of the multi-hypergraph $\mathcal{H}$: $\mathcal{B}_{\mathcal{V},\tilde{\mathcal{E}}}$, is defined as before as the bipartite graph with vertices $\mathcal{V} \bigsqcup \tilde{\mathcal{E}}$ and edges $\{(v, e) \in \mathcal{V} \times \tilde{\mathcal{E}} \mid v \in e\}$.*

*With these definitions on multi-hypergraphs, Proposition B.5, Theorem B.8, and Corollary 2 also hold for multi-hypergraphs.*

*Proof.* In the proposition, theorem and corollary, replace the set $\mathcal{E}$ with the multiset $\tilde{\mathcal{E}}$ and the proofs become identical. $\square$

### B.3.1  Algorithm Guarantees

Continuing with the notation, as before, let $\mathcal{H} = (\mathcal{V}, \mathcal{E})$ be a hypergraph with star expansion matrix $H$ and let $(\mathcal{R}_{\mathcal{V}}, \mathcal{R}_{\mathcal{E}})$ be the output of Algorithm 1 on $H$ for $L \in \mathbb{Z}^+$. Denote $\mathcal{C}_{c_L}$ as the set of all connected components of $\mathcal{H}_{c_L}$:

$$\mathcal{C}_{c_L} \triangleq \{C_{c_L} : \text{conn. comp. } C_{c_L} \text{ of } \mathcal{H}_{c_L}\} \tag{72}$$

If $L = \infty$, then drop the $L$. Thus, the hypergraphs represented by $(\mathcal{R}_V, \mathcal{R}_E)$ come from $\mathcal{C}_{c_L}$ for each $c_L$. Let:

$$\hat{\mathcal{H}}_L \triangleq (\mathcal{V}, \mathcal{E} \cup \mathcal{R}_V) \tag{73}$$

be $\mathcal{H}$ after adding all the hyperedges from $\mathcal{R}_\mathcal{V}$ and let $\hat{H}_L$ be the star expansion matrix of the resulting multi-hypergraph $\hat{\mathcal{H}}_L$. Let:

$$\mathcal{G}_L \triangleq \{h_v^L(H) : v \in \mathcal{V}\} \tag{74}$$

be the set of all $L$-GWL-1 values on $H$. Let:

$$V_{c_L,s} \triangleq \{v \in \mathcal{V}_{c_L} : v \in R, R \in \mathcal{C}_{c_L}, |\mathcal{V}_R| = s\} \tag{75}$$

be the set of all nodes of $L$-GWL-1 class $c_L$ belonging to a connected component in $\mathcal{C}_{c_L}$ of $s \geq 1$ nodes in $\mathcal{H}_{c_L}$, the induced subhypergraph of $L$-GWL-1. Let:

$$\mathcal{S}_{c_L} \triangleq \{|\mathcal{V}_{\mathcal{R}_{c_L,i}}| : \mathcal{R}_{c_L,i} \in \mathcal{C}_{c_L}\} \tag{76}$$

be the set of node set sizes of the connected components in $\mathcal{H}_{c_L}$.

**Proposition B.13.** *If $L = \infty$, for any GWL-1 node value $c$ for $\mathcal{H}$, the connected component induced subhypergraphs $\mathcal{R}_{c,i}$, for $i = 1, ..., |\mathcal{C}_c|$ are GWL-1 symmetric and neighborhood-regular.*

*Proof.* Let $p_{\mathcal{B}_{\mathcal{V},\mathcal{E}}}$ be the universal covering map for $\mathcal{B}_{\mathcal{V},\mathcal{E}}$. Denote $\tilde{v}, \tilde{u}, \tilde{v}', \tilde{u}'$ by the lift of some nodes $v, u, v', u' \in \mathcal{V}$ by $p_{\mathcal{B}_{\mathcal{V},\mathcal{E}}}$.

Let $L = \infty$ and let $\mathcal{H}_c = (\mathcal{V}_c, \mathcal{E}_c)$. For any $i$, since $u, v \in \mathcal{V}_c$, $(\tilde{\mathcal{B}}_{\mathcal{V},\mathcal{E}})_u \cong_c (\tilde{\mathcal{B}}_{\mathcal{V},\mathcal{E}})_v$ for all $u, v \in \mathcal{V}_{\mathcal{R}_{c,i}}$. Since $\mathcal{R}_{c,i}$ is maximally connected we know that every neighborhood $N_{\mathcal{H}_c}(u)$ for $u \in \mathcal{V}_c$ induced by $\mathcal{H}_c$ has $N_{\mathcal{H}_c}(u) \cong N(u) \cap \mathcal{H}_c$. Since $L = \infty$ we have that $N_{\mathcal{H}_c}(u) \cong N_{\mathcal{H}_c}(v), \forall u, v \in \mathcal{V}_{R_{c,i}}$ since otherwise WLOG there are $u', v' \in \mathcal{V}_{R_{c,i}}$ with $N_{\mathcal{H}_c}(u') \ncong N_{\mathcal{H}_c}(v')$ then WLOG there is some hyperedge $e \in \mathcal{E}_{N_{\mathcal{H}_c}(u')}$ with some $w \in e$, $w \neq u'$ where $e$ cannot be in isomorphism with any $e' \in \mathcal{E}_{N_{\mathcal{H}_c}(v')}$. For two hyperedges to be in isomorphism means that their constituent nodes can be bijectively mapped to each other by a restriction of an isomorphism $\phi$ between $N_{\mathcal{H}_c}(u'), N_{\mathcal{H}_c}(v')$ to one of the hyperedges. This means that $(\tilde{\mathcal{B}}_{\mathcal{V}\setminus\{u'\},\mathcal{E}})_w$ is the rooted universal covering subtree centered about $w$ not passing through $u'$ that is connected to $u' \in (\tilde{\mathcal{B}}_{\mathcal{V},\mathcal{E}})_{u'}$ by $e$. However, $v'$ has no $e$ and thus cannot have a $T_x$ for $x \in \mathcal{V}_{(\tilde{N}(v'))_v}$ satisfying $T_x \cong_c (\tilde{\mathcal{B}}_{\mathcal{V}\setminus\{u'\},\mathcal{E}})_w$ with $x$ connected to $v'$ by a hyperedge $e'$ isomorphic to $e$ in its neighborhood in $(\tilde{\mathcal{B}}_{\mathcal{V},\mathcal{E}})_{v'}$. This contradicts that $(\tilde{\mathcal{B}}_{\mathcal{V},\mathcal{E}})_{u'} \cong_c (\tilde{\mathcal{B}}_{\mathcal{V},\mathcal{E}})_{v'}$.

We have thus shown that all nodes in $\mathcal{V}_c$ have isomorphic induced neighborhoods. By the Observation 4, this is equivalent to saying that $\mathcal{R}_{c,i}$ is GWL-1 symmetric and neighborhood regular. $\square$

We show the partitioning of $\mathcal{H}$ by Algorithm 12:

**Proposition B.14.** *If $L \geq 1$, the output $(\mathcal{R}_\mathcal{V}, \mathcal{R}_\mathcal{E})$ of Algorithm 1 partitions a subgraph of $\mathcal{H}$, meaning:*

$$\mathcal{V} = \sqcup_{V \in \mathcal{R}_\mathcal{V}} V \text{ and } \mathcal{E} \supset \sqcup_{E \in \mathcal{R}_\mathcal{E}} E \tag{77}$$

*Proof.* **1.** $\mathcal{V} = \sqcup_{V \in \mathcal{R}_\mathcal{V}} V$**:**

For a given subset of nodes $\mathcal{U} \subseteq \mathcal{V}$, let ConnectedComponents$_\mathcal{H}(\mathcal{U})$ be the collection of node subsets of $\mathcal{U}$ where each node subset forms a connected component in $\mathcal{H}$.

Let $h_v^L(H)$ denote the $L$-GWL-1 node value of $v \in \mathcal{V}$.

Let $\mathcal{R}_\mathcal{V}(L) \triangleq \bigcup_{v \in \mathcal{V}}$ ConnectedComponents$_\mathcal{H}(\{u \in \mathcal{V} : h_u^L(H) = h_v^L(H)\})$ denote the collection of node sets of common $L$-GWL-1 values for a given $L$.

By the definition of $\mathcal{R}_\mathcal{V}$, since every connected component is size atleast 1 and every node is considered, we must have $\bigcup_{v \in \mathcal{V}} \mathcal{R}_\mathcal{V} = \mathcal{V}$. Since each connected component of a given GWL-1 value is maximal, meaning there is no superset of nodes that is connected, no two connected components can intersect through either nodes or hyperedges. Furthermore, a single node can belong to only one GWL-1 value, thus the values form a partition of $\mathcal{V}$. This proves $\mathcal{V} = \sqcup_{V \in \mathcal{R}_\mathcal{V}} V$.

**2.** $\mathcal{E} \supset \sqcup_{E \in \mathcal{R}_\mathcal{E}} E$**:**

This follows since $\mathcal{R}_\mathcal{E}$ are the hyperedges of each connected component spanned by $\mathcal{R}_\mathcal{V}$. These connected components form disconnected subhypergraphs of $\mathcal{H}$. $\square$

**Prediction Guarantees:**

In order to guarantee that the GWL-1 symmetric components $\mathcal{R}_{c,i}$ found by Algorithm 12 carry additional information, there needs to be a separation between them to prevent an intersection between the rooted trees computed by GWL-1. We redefine from the main paper what it means for two node subsets to be sufficiently separated via the shortest hyperedge path distance between nodes in $\mathcal{V}$ as follows:

**Definition B.12.** *Two subsets of nodes* $\mathcal{U}_1, \mathcal{U}_2 \subseteq \mathcal{V}$ *are* **sufficiently $L$-separated** *if:*

$$\min_{v_1 \in \mathcal{U}_1, v_2 \in \mathcal{U}_2} d(v_1, v_2) > L \tag{78}$$

*where* $d(v_1, v_2) \triangleq \min_{e_1, \dots, e_k \in \mathcal{E}, v_1 \in e_1, v_2 \in e_k} k$ *is the shortest hyperedge path distance from* $v_1 \in \mathcal{V}$ *to* $v_2 \in \mathcal{V}$.

*A collection of node subsets* $\mathcal{C} \subseteq 2^{\mathcal{V}}$ *is* **sufficiently $L$-separated** *if all pairs of node subsets are* **sufficiently $L$-separated**.

Our definition of sufficiently $L$-separated is similar in nature to that of well separation between point sets Callahan & Kosaraju (1995) in Euclidean space.

We give another definition that will be useful for the proof of the following lemma:

**Definition B.13.** *A star graph* $N_x$ *is defined as a tree rooted at* $x$ *of depth* 1. *The root* $x$ *is the only node that can have degree more than* 1.

Assuming that the $\mathcal{C}_{c_L}$ are sufficiently $L$-separated from each other, intuitively meaning that no two nodes from two separate $\mathcal{V}_{\mathcal{R}_{c_L,i}} \in \mathcal{R}_V$ are within $L$ hyperedges away, then the cardinality of each component $|\mathcal{V}_{\mathcal{R}_{c_L,i}}|$ is recognizable.

**Lemma B.15.** *If* $L \in \mathbb{Z}^+$ *is small enough so that after running Algorithm 1 on* $L$, *for any* $L$-*GWL-1 node class* $c_L$ *on* $\mathcal{V}$ *the collection of* $\mathcal{C}_{c_L}$ *is* **sufficiently $L$-separated**,

*then after forming* $\hat{\mathcal{H}}_L$, *the new* $L$-*GWL-1 node classes of* $\mathcal{V}_{\mathcal{R}_{c_L,i}}$ *for* $i = 1, \dots, \mathcal{C}_{c_L}$ *in* $\hat{\mathcal{H}}_L$ *are all the same class* $c_L'$ *but are distinguishable from* $c_L$ *depending on* $|\mathcal{V}_{\mathcal{R}_{c_L,i}}|$.

*Proof.* After running Algorithm 1 on $\mathcal{H} = (\mathcal{V}, \mathcal{E})$, let $\hat{\mathcal{H}}_L = (\hat{\mathcal{V}}_L, \hat{\mathcal{E}}_L \triangleq \mathcal{E} \cup \bigsqcup_{c_L,i} \{\mathcal{V}_{\mathcal{R}_{c_L,i}}\})$ be the hypergraph formed by attaching a hyperedge to each $\mathcal{V}_{\mathcal{R}_{c_L,i}}$.

For any $c_L$, a $L$-GWL-1 node class, let $\mathcal{R}_{c_L,i}, i = 1, \dots, |\mathcal{C}_{c_L}|$ be a connected component subhypergraph of $\mathcal{H}_{c_L}$. Over all $(c_L, i)$ pairs, all the $\mathcal{R}_{c_L,i}$ are disconnected from each other and for each $c_L$ each $\mathcal{R}_{c_L,i}$ is maximally connected on $\mathcal{H}_{c_L}$.

Upon covering all the nodes $\mathcal{V}_{\mathcal{R}_{c_L,i}}$ of each induced connected component subhypergraph $\mathcal{R}_{c_L,i}$ with a single hyperedge $e = \mathcal{V}_{\mathcal{R}_{c_L,i}}$ of size $s = |\mathcal{V}_{\mathcal{R}_{c_L,i}}|$, we claim that every node of class $c_L$ becomes $c_{L,s}$, a $L$-GWL-1 node class depending on the original $L$-GWL-1 node class $c_L$ and the size of the hyperedge $s$.

Consider for each $v \in \mathcal{V}_{\mathcal{R}_{c_L,i}}$ the $2L$-hop rooted tree $(\tilde{\mathcal{B}}_{\mathcal{V},\mathcal{E}}^{2L})_{\tilde{v}}$ for $p_{\mathcal{B}_{\mathcal{V},\mathcal{E}}}(\tilde{v}) = v$. Also, for each $v \in \mathcal{V}_{\mathcal{R}_{c_L,i}}$, define the tree

$$T_e \triangleq (\tilde{\mathcal{B}}_{\hat{\mathcal{V}} \setminus \{v\}, \hat{\mathcal{E}}}^{2L-1})_{\tilde{e}} \tag{79}$$

We do not index the tree $T_e$ by $v$ since it does not depend on $v \in \mathcal{V}_{\mathcal{R}_{c_L,i}}$. We prove this in the following.

**proof for: $T_e$ does not depend on $v \in \mathcal{V}_{\mathcal{R}_{c_L,i}}$:**

Let node $\tilde{e}$ be the lift of $e$ to $(\tilde{\mathcal{B}}_{\hat{\mathcal{V}}, \hat{\mathcal{E}}}^{2L-1})_{\tilde{e}}$. Define the star graph $(N(\tilde{e}))_{\tilde{e}}$ as the 1-hop neighborhood of $\tilde{e}$ in $(\tilde{\mathcal{B}}_{\hat{\mathcal{V}}, \hat{\mathcal{E}}}^{2L-1})_{\tilde{e}}$. We must have:

$$(\tilde{\mathcal{B}}_{\hat{\mathcal{V}}, \hat{\mathcal{E}}}^{2L-1})_{\tilde{e}} \cong_c ((N(\tilde{e}))_{\tilde{e}} \sqcup \bigsqcup_{\tilde{u} \in \mathcal{V}_{N(\tilde{e})} \setminus \{\tilde{e}\}} (\tilde{\mathcal{B}}_{\mathcal{V},\mathcal{E}}^{2L-2})_{\tilde{u}})_{\tilde{e}} \tag{80}$$

Define for each node $v \in e$ with lift $\tilde{v}$:

$$(N(\tilde{e}, \tilde{v}))_{\tilde{e}} \triangleq (\mathcal{V}_{(N(\tilde{e}))_{\tilde{e}}} \setminus \{\tilde{v}\}, \mathcal{E}_{(N(\tilde{e}))_{\tilde{e}}} \setminus \{(\tilde{e}, \tilde{v})\})_{\tilde{e}} \tag{81}$$

The tree $(N(\tilde{e}, \tilde{v}))_{\tilde{e}}$ is a star graph with the node $\tilde{v}$ deleted from $(N(\tilde{e}))_{\tilde{e}}$. The star graphs $(N(\tilde{e}, \tilde{v}))_{\tilde{e}} \subseteq (N(\tilde{e}))_{\tilde{e}}$ do not depend on $\tilde{v}$ as long as $\tilde{v} \in \mathcal{V}_{(N(\tilde{e}))_{\tilde{e}}}$. In other words,

$$(N(\tilde{e}, \tilde{v}))_{\tilde{e}} \cong_c (N(\tilde{e}, \tilde{v}'))_{\tilde{e}}, \forall \tilde{v}, \tilde{v}' \in \mathcal{V}_{(N(\tilde{e}))_{\tilde{e}}} \setminus \{\tilde{e}\} \tag{82}$$

Since the rooted tree $(\tilde{\mathcal{B}}_{\hat{\mathcal{V}}, \hat{\mathcal{E}}}^{2L-1})_{\tilde{e}}$, where $\tilde{e}$ is the lift of $e$ by universal covering map $p_{\mathcal{B}_{\mathcal{V}, \mathcal{E}}}$, has all pairs of nodes $\tilde{u}, \tilde{u}' \in \tilde{e}$ in it with $(\tilde{\mathcal{B}}_{\mathcal{V}, \mathcal{E}}^{2L})_{\tilde{u}} \cong_c (\tilde{\mathcal{B}}_{\mathcal{V}, \mathcal{E}}^{2L})_{\tilde{u}'}$, which implies

$$(\tilde{\mathcal{B}}_{\mathcal{V}, \mathcal{E}}^{2L-2})_{\tilde{u}} \cong_c (\tilde{\mathcal{B}}_{\mathcal{V}, \mathcal{E}}^{2L-2})_{\tilde{u}'}, \forall \tilde{u}, \tilde{u}' \in \tilde{e} \tag{83}$$

By Equations 83, 82, we thus have:

$$(\tilde{\mathcal{B}}_{\hat{\mathcal{V}} \setminus \{v\}, \hat{\mathcal{E}}}^{2L-1})_{\tilde{e}} \cong_c ((N(\tilde{e}, \tilde{v}))_{\tilde{e}} \sqcup \bigsqcup_{\tilde{u} \in \mathcal{V}_{(N(\tilde{e}, \tilde{v}))_{\tilde{e}}} \setminus \{\tilde{e}\}} (\tilde{\mathcal{B}}_{\mathcal{V}, \mathcal{E}}^{2L-2})_{\tilde{u}})_{\tilde{e}} \tag{84}$$

This proves that $T_e$ does not need to be indexed by $v \in \mathcal{V}_{\mathcal{R}_{c_L, i}}$.

We continue with the proof that all nodes in $\mathcal{V}_{\mathcal{R}_{c_L, i}}$ become the $L$-GWL-1 node class $c_{L,s}$ for $s = |\mathcal{V}_{\mathcal{R}_{c_L, i}}|$.

Since every $v \in \mathcal{V}_{\mathcal{R}_{c_L, i}}$ becomes connected to a hyperedge $e = \mathcal{V}_{\mathcal{R}_{c_L, i}}$ in $\hat{\mathcal{H}}$, we must have:

$$(\tilde{\mathcal{B}}_{\hat{\mathcal{V}}, \hat{\mathcal{E}}}^{2L})_{\tilde{v}} \cong_c ((\tilde{\mathcal{B}}_{\mathcal{V}, \mathcal{E}}^{2L})_{\tilde{v}} \cup_{(\tilde{v}, \tilde{e})} T_e)_{\tilde{v}}, \forall v \in \mathcal{V}_{\mathcal{R}_{c_L, i}} \tag{85}$$

The notation $((\tilde{\mathcal{B}}_{\mathcal{V}, \mathcal{E}}^{2L})_{\tilde{v}} \cup_{(\tilde{v}, \tilde{e})} T_e)_{\tilde{v}}$ denotes a tree rooted at $\tilde{v}$ that is the attachment of the tree $T_e$ rooted at $\tilde{e}$ to the node $\tilde{v}$ by the edge $(\tilde{v}, \tilde{e})$. As is usual, we assume $\tilde{v}, \tilde{e}$ are the lifts of $v \in \mathcal{V}, e \in \mathcal{E}$ respectively. We only need to consider the single $e$ since $L$ was chosen small enough so that the $2L$-hop tree $(\tilde{\mathcal{B}}_{\hat{\mathcal{V}}, \hat{\mathcal{E}}}^{2L})_{\tilde{v}}$ does not contain a node $\tilde{u}$ satisfying $p_{\mathcal{B}_{\mathcal{V}, \mathcal{E}}}(\tilde{u}) = u$ with $u \in \mathcal{V}_{\mathcal{R}_{c_L, j}}$ for all $j = 1, ..., |\mathcal{C}_{c_L}|, j \neq i$.

Since $T_e$ does not depend on $v \in \mathcal{V}_{\mathcal{R}_{c_L, i}}$,

$$(\tilde{\mathcal{B}}_{\hat{\mathcal{V}}, \hat{\mathcal{E}}}^{2L})_{\tilde{u}} \cong_c (\tilde{\mathcal{B}}_{\hat{\mathcal{V}}, \hat{\mathcal{E}}}^{2L})_{\tilde{v}}, \forall u, v \in \mathcal{V}_{\mathcal{R}_{c_L, i}} \tag{86}$$

This shows that $h_u^L(\hat{H}) = h_v^L(\hat{H}), \forall u, v \in \mathcal{V}_{\mathcal{R}_{c_L, i}}$ by Theorem B.8. Furthermore, since each $v \in \mathcal{V}_{\mathcal{R}_{c_L, i}} \subseteq \hat{\mathcal{V}}$ in $\hat{\mathcal{H}}$ is now incident to a new hyperedge $e = \mathcal{V}_{\mathcal{R}_{c_L, i}}$, we must have that the $L$-GWL-1 class $c_L$ of $\mathcal{V}_{\mathcal{R}_{c_L, i}}$ on $\mathcal{H}$ is now distinguishable by $|\mathcal{V}_{\mathcal{R}_{c_L, i}}|$.

$\square$

We will need the following definition to prove the next lemma.

**Definition B.14.** *A partial universal cover of hypergraph $\mathcal{H} = (\mathcal{V}, \mathcal{E})$ with an unexpanded induced subhypergraph $\mathcal{R}$, denoted $U(\mathcal{H}, \mathcal{R})_{\mathcal{V}, \mathcal{E}}$ is a graph cover of $\mathcal{B}_{\mathcal{V}, \mathcal{E}}$ where we freeze $\mathcal{B}_{\mathcal{V}_{\mathcal{R}}, \mathcal{E}_{\mathcal{R}}} \subseteq \tilde{\mathcal{B}}_{\mathcal{V}, \mathcal{E}}$ as an induced subgraph.*

*A l-hop rooted partial universal cover of hypergraph $\mathcal{H} = (\mathcal{V}, \mathcal{E})$ with an unexpanded induced subhypergraph $\mathcal{R}$, denoted $(U^l(\mathcal{H}, \mathcal{R})_{\mathcal{V}, \mathcal{E}})_{\tilde{u}}$ for $u \in \mathcal{V}$ or $(U^l(\mathcal{H}, \mathcal{R})_{\mathcal{V}, \mathcal{E}})_{\tilde{e}}$ for $e \in \mathcal{E}$, where $\tilde{v}, \tilde{e}$ are lifts of $v, e$, is a rooted graph cover of $\mathcal{B}_{\mathcal{V}, \mathcal{E}}$ where we freeze $\mathcal{B}_{\mathcal{V}_{\mathcal{R}}, \mathcal{E}_{\mathcal{R}}} \subseteq \tilde{\mathcal{B}}_{\mathcal{V}, \mathcal{E}}$ as an induced subgraph.*

**Lemma B.16.** *Assuming the same conditions as Lemma B.15, where $\mathcal{H} = (\mathcal{V}, \mathcal{E})$ is a hypergraph and for all $L$-GWL-1 node classes $c_L$ with connected components $\mathcal{R}_{c_L, i}$, as discovered by Algorithm 1, so that $L \geq diam(\mathcal{R}_{c_L, i})$. Instead of only adding the hyperedges $\{\mathcal{V}_{\mathcal{R}_{c_L, i}}\}_{c_L, i}$ to $\mathcal{E}$ as stated in the main paper, let $\hat{\mathcal{H}}_{\dagger} \triangleq (\mathcal{V}, (\mathcal{E} \setminus \mathcal{R}_E) \sqcup \mathcal{R}_V)$, meaning $\mathcal{H}$ with each $\mathcal{R}_{c_L, i}$ for $i = 1, ..., |\mathcal{C}_{c_L}|$ having all of its hyperedges dropped and with a single hyperedge that covers $\mathcal{V}_{\mathcal{R}_{c_L, i}}$ and let $\hat{\mathcal{H}} = (\mathcal{V}, \mathcal{E} \sqcup \mathcal{R}_V)$ then:*

*The GWL-1 node classes of $\mathcal{V}_{\mathcal{R}_{c_L, i}}$ for $i = 1, ..., |\mathcal{C}_{c_L}|$ in $\hat{\mathcal{H}}$ are all the same class $c_L'$ but are distinguishable from $c_L$ depending on $|\mathcal{V}_{\mathcal{R}_{c_L, i}}|$.*

*Proof.* For any $c_L$, a $L$-GWL-1 node class, let $\mathcal{R}_{c_L,i}, i = 1, ..., |\mathcal{C}_{c_L}|$ be a connected component subhypergraph of $\mathcal{H}_{c_L}$. These connected components are discovered by the algorithm. Over all $(c_L, i)$ pairs, all the $\mathcal{R}_{c_L,i}$ are disconnected from each other. Upon arbitrarily deleting all hyperedges in each such induced connected component subhypergraph $\mathcal{R}_{c_L,i}$ and adding a single hyperedge of size $s = |\mathcal{V}_{\mathcal{R}_{c_L,i}}|$, we claim that every node of class $c_L$ becomes $c_{L,s}$, a $L$-GWL-1 node class depending on the original $L$-GWL-1 node class $c_L$ and the size of the hyperedge $s$.

Define the subhypergraph made up of the disconnected components $\mathcal{R}_{c_L,i}$ as:

$$\mathcal{R} := \bigcup_{c,i} \mathcal{R}_{c_L,i} \tag{87}$$

Since $L \geq diam(\mathcal{R}_{c_L,i})$, we can construct the $2L$-hop rooted partial universal cover with unexpanded induced subhypergraph $\mathcal{R}$, denoted by $(U^{2L}(\mathcal{H}, \mathcal{R})_{\mathcal{V},\mathcal{E}})_{\tilde{v}}, \forall v \in \mathcal{V}$ of $\mathcal{H}$ as given in Definition B.14.

Denote the hyperedge nodes, or right hand nodes of the bipartite graph by $\mathcal{B}(\mathcal{V}_\mathcal{R}, \mathcal{E}_\mathcal{R})$ by $R(\mathcal{B}(\mathcal{V}_\mathcal{R}, \mathcal{E}_\mathcal{R}))$. Their corresponding hyperedges are $\mathcal{E}_\mathcal{R} \subseteq \mathcal{E}(U(\mathcal{H}, \mathcal{R})) \subseteq \mathcal{E}$. Since each $\mathcal{R}_{c_L,i}$ is maximally connected, for any nodes $u, v \in \mathcal{V}_\mathcal{R}$ we have:

$$(U^{2L}(\mathcal{H}, \mathcal{R})_{\tilde{u}} \setminus R(\mathcal{B}(\mathcal{V}_\mathcal{R}, \mathcal{E}_\mathcal{R})))_{\tilde{u}} \cong_c (U^{2L}(\mathcal{H}, \mathcal{R})_{\tilde{v}} \setminus R(\mathcal{B}(\mathcal{V}_\mathcal{R}, \mathcal{E}_\mathcal{R})))_{\tilde{v}} \tag{88}$$

by Proposition B.13, where $U^{2L}(\mathcal{H}, \mathcal{R})_{\tilde{v}} \setminus R(\mathcal{B}(\mathcal{V}_\mathcal{R}, \mathcal{E}_\mathcal{R}))$ denotes removing the nodes $R(\mathcal{B}(\mathcal{V}_\mathcal{R}, \mathcal{E}_\mathcal{R}))$ from $U^{2L}(\mathcal{H}, \mathcal{R})_{\tilde{v}}$. This follows since removing $R(\mathcal{B}(\mathcal{V}_\mathcal{R}, \mathcal{E}_\mathcal{R}))$ removes an isomorphic neighborhood of hyperedges from each node in $\mathcal{V}_\mathcal{R}$. This requires assuming maximal connectedness of each $\mathcal{R}_{c_L,i}$. Upon adding the hyperedge

$$e_{c_L,i} \triangleq \mathcal{V}_{\mathcal{R}_{c_L,i}} \tag{89}$$

covering all of $\mathcal{V}_{\mathcal{R}_{c_L,i}}$ after the deletion of $\mathcal{E}_{\mathcal{R}_{c_L,i}}$ for every $(c_L, i)$ pair, we see that any node $u \in \mathcal{V}_{\mathcal{R}_{c_L,i}}$ is connected to any other node $v \in \mathcal{V}_{\mathcal{R}_{c_L,i}}$ through $e_{c_L,i}$ in the same way for all nodes $u, v \in \mathcal{V}_{\mathcal{R}_{c_L,i}}$. In fact, we claim that all the nodes in $\mathcal{V}_{\mathcal{R}_{c_L,i}}$ still have the same GWL-1 class.

We can write the multi-hypergraph $\hat{\mathcal{H}}_\dagger$ equivalently as $(\mathcal{V}, \bigsqcup_{c_L,i}(\mathcal{E} \setminus \mathcal{E}(\mathcal{R}_{c_L,i}) \sqcup \{\!\{e_{c_L,i}\}\!\}))$, which is the multi-hypergraph formed by the algorithm. The replacement operation on $\mathcal{H}$ can be viewed in the universal covering space $\tilde{\mathcal{B}}_{\mathcal{V},\mathcal{E}}$ as taking $U(\mathcal{H}, \mathcal{R})$ and replacing the frozen subgraph $\mathcal{B}_{\mathcal{V}_\mathcal{R}, \mathcal{E}_\mathcal{R}}$ with the star graphs $(N_{\hat{\mathcal{H}}_\dagger}(\tilde{e}_{c_L,i}))_{\tilde{e}_{c_L,i}}$ of root node $\tilde{e}_{c_L,i}$ determined by hyperedge $e_{c_L,i}$ for each connected component indexed by $(c_L, i)$. Since the star graphs $(N_{\hat{\mathcal{H}}_\dagger}(\tilde{e}_{c_L,i}))_{\tilde{e}_{c_L,i}}$ are cycle-less, we have that:

$$(U(\mathcal{H}, \mathcal{R}) \setminus R(\mathcal{B}(\mathcal{V}_\mathcal{R}, \mathcal{E}_\mathcal{R}))) \cup \bigcup_{c_L,i}(N_{\hat{\mathcal{H}}_\dagger}(\tilde{e}_{c_L,i}))_{\tilde{e}_{c_L,i}} \cong_c \tilde{\mathcal{B}}_{\mathcal{V}_{\hat{\mathcal{H}}_\dagger}, \mathcal{E}_{\hat{\mathcal{H}}_\dagger}} \tag{90}$$

Viewing Equation 90 locally, by our assumptions on $L$, for any $v \in \mathcal{V}_{\mathcal{R}_{c_L,i}}$, we must also have:

$$(U^{2L}(\mathcal{H}, \mathcal{R})_{\tilde{v}} \setminus R(\mathcal{B}(\mathcal{V}_\mathcal{R}, \mathcal{E}_\mathcal{R}))) \bigcup (N_{\hat{\mathcal{H}}_\dagger}(\tilde{e}_{c,i}))_{\tilde{e}_{c,i}} \cong_c \tilde{\mathcal{B}}_{\mathcal{V}_{\hat{\mathcal{H}}_\dagger}, \mathcal{E}_{\hat{\mathcal{H}}_\dagger}} \tag{91}$$

We thus have $(\tilde{\mathcal{B}}^{2L}_{\mathcal{V}_{\hat{\mathcal{H}}_\dagger}, \mathcal{E}_{\hat{\mathcal{H}}_\dagger}})_{\tilde{u}} \cong_c (\tilde{\mathcal{B}}^{2L}_{\mathcal{V}_{\hat{\mathcal{H}}_\dagger}, \mathcal{E}_{\hat{\mathcal{H}}_\dagger}})_{\tilde{v}}$ for every $u, v \in \mathcal{V}_{\mathcal{R}_{c_L,i}}$ with $\tilde{u}, \tilde{v}$ being the lifts of $u, v$ by $p_{\mathcal{B}_{\mathcal{V},\mathcal{E}}}$, since $(U^{2L}(\mathcal{H}, \mathcal{R})_{\tilde{u}} \setminus R(\mathcal{B}(\mathcal{V}_\mathcal{R}, \mathcal{E}_\mathcal{R})))_{\tilde{u}} \cong_c (U^{2L}(\mathcal{H}, \mathcal{R})_{\tilde{v}} \setminus R(\mathcal{B}(\mathcal{V}_\mathcal{R}, \mathcal{E}_\mathcal{R})))_{\tilde{v}}$ for every $u, v \in \mathcal{V}_{\mathcal{R}_{c_L,i}}$ as in Equation 88. These rooted universal covers now depend on a new hyperedge $e_{c_L,i}$ and thus depend on its size $s$.

This proves the claim that all the nodes in $\mathcal{V}_{\mathcal{R}_{c_L,i}}$ retain the same $L$-GWL-1 node class by changing $\mathcal{H}$ to $\hat{\mathcal{H}}_\dagger$ and that this new class is distinguishable by $s = |\mathcal{V}_{\mathcal{R}_{c_L,i}}|$. In otherwords, the new class can be determined by $c_s$. Furthermore, $c_{L,s}$ on the hyperedge $e_{c_L,i}$ cannot become the same class as an existing class due to the algorithm. $\square$

**Theorem B.17.** *Let $|\mathcal{V}| = n, L \in \mathbb{Z}^+$ and $vol(v) \triangleq \sum_{e \in \mathcal{E}:e \ni v} |e|$ and assuming that the collection of node subsets $\mathcal{C}_{c_L}$ is sufficiently $L$-separated.*

*If $vol(v) = O(\log^{\frac{1-\epsilon}{4L}} n), \forall v \in \mathcal{V}$ for any constant $\epsilon > 0$; $|\mathcal{S}_{c_L}| \leq S, \forall c_L \in \mathcal{C}_L$, $S$ constant, and $|V_{c_L,s}| = O(\frac{n^\epsilon}{\log^{\frac{1}{2k}}(n)}), \forall s \in \mathcal{C}_{c_L}$ , then for $k \in \mathbb{Z}^+$ and $k$-tuple $C = (c_{L,1}, ..., c_{L,k}), c_{L,i} \in \mathcal{G}_L, i = 1..k$ there exists*

$\omega(n^{2k\epsilon})$ *many pairs of $k$-node sets $S_1 \not\simeq S_2$ such that $(h_u^L(H))_{u \in S_1} = (h_{v \in S_2}^L(H)) = C$, as ordered $k$-tuples, while $h(S_1, \hat{H}_L) \neq h(S_2, \hat{H}_L)$ also by $L$ steps of GWL-1.*

*Proof.*

**1. Constructing forests from the rooted universal cover trees** :

The first part of the proof is similar to the first part of the proof of Theorem 2 of Zhang et al. (2021).

Consider an arbitrary node $v \in \mathcal{V}$ and denote the $2L$-hop tree rooted at $v$ from the universal cover as $(\tilde{\mathcal{B}}_{\mathcal{V},\mathcal{E}}^{2L})_v$ as in Theorem B.8. As each node $v \in \mathcal{V}$ has volume $vol(v) = \sum_{v \in e} |e| = O(\log^{\frac{1-\epsilon}{4L}} n)$, then every edge $e \in \mathcal{E}$ has $|e| = O(\log^{\frac{1-\epsilon}{4L}} n)$ and for all $v \in \mathcal{V}$ we have that $deg(v) = O(\log^{\frac{1-\epsilon}{4L}} n)$, we can say that every node in $(\tilde{\mathcal{B}}_{\mathcal{V},\mathcal{E}}^{2L})_{\tilde{v}}$ has degree $d = O(\log^{\frac{1-\epsilon}{4L}} n)$. Thus, the number of nodes in $(\tilde{\mathcal{B}}_{\mathcal{V},\mathcal{E}}^{2L})_{\tilde{v}}$, denoted by $|\mathcal{V}((\tilde{\mathcal{B}}_{\mathcal{V},\mathcal{E}}^{2L})_{\tilde{v}}|$, satisfies $|\mathcal{V}((\tilde{\mathcal{B}}_{\mathcal{V},\mathcal{E}}^{2L})_{\tilde{v}}| \leq \sum_{i=0}^{2L} d^i = O(d^{2L}) = O(\log^{\frac{1-\epsilon}{2}} n)$. We set $K \triangleq \max_{v \in V} |\mathcal{V}((\tilde{\mathcal{B}}_{\mathcal{V},\mathcal{E}}^{2L})_{\tilde{v}}|$ as the maximum number of nodes of $(\tilde{\mathcal{B}}_{\mathcal{V},\mathcal{E}}^{2L})_{\tilde{v}}$ and thus $K = O(\log^{\frac{1-\epsilon}{2}} n)$. For all $v \in \mathcal{V}$, expand trees $(\tilde{\mathcal{B}}_{\mathcal{V},\mathcal{E}}^{2L})_{\tilde{v}}$ to $\overline{(\tilde{\mathcal{B}}_{\mathcal{V},\mathcal{E}}^{2L})_{\tilde{v}}}$ by adding $K - |\mathcal{V}((\tilde{\mathcal{B}}_{\mathcal{V},\mathcal{E}}^{2L})_{\tilde{v}}|$ independent nodes. Then, all $\overline{(\tilde{\mathcal{B}}_{\mathcal{V},\mathcal{E}}^{L})_{\tilde{v}}}$ have the same number of nodes, which is $K$, becoming forests instead of trees.

**2. Counting $|\mathcal{G}_L|$:**

Next, we consider the number of non-isomorphic forests over $K$ nodes. Actually, the number of non-isomorphic graphs over K nodes is bounded by $2^{\binom{K}{2}} = exp(O(\log^{\frac{1-\epsilon}{2}} n)) = o(n^{1-\epsilon})$. Therefore, due to the pigeonhole principle, there exist $\frac{n}{o(n^{1-\epsilon})} = \omega(n^{\epsilon})$ many nodes $v$ whose $\overline{(\tilde{\mathcal{B}}_{\mathcal{V},\mathcal{E}}^{L})_{\tilde{v}}}$ are isomorphic to each other. Denote $\mathcal{G}_L$ as the set of all $L$-GWL-1 values. Denote the set of these nodes as $\mathcal{V}_{c_L}$, which consist of nodes whose $L$-GWL-1 values are all the same value $c_L \in \mathcal{G}_L$ after $L$ iterations of GWL-1 by Theorem B.8. For a fixed $L$, the sets $\mathcal{V}_{c_L}$ form a partition of $\mathcal{V}$, in other words, $\bigsqcup_{c_L \in \mathcal{G}_L} \mathcal{V}_{c_L} = \mathcal{V}$. Next, we focus on looking at $k$-sets of nodes that are not equivalent by GWL-1.

For any $c_L \in \mathcal{G}_L$, there is a partition $\mathcal{V}_{c_L} = \bigsqcup_s V_{c_L,s}$ where $V_{c_L,s}$ is the set of nodes all of which have $L$-GWL-1 class $c_L$ and that belong to a connected component of size $s$ in $\mathcal{H}_{c_L}$. Let $\mathcal{S}_{c_L} \triangleq \{|\mathcal{V}_{\mathcal{R}_{c_L,j}}| : \mathcal{R}_{c_L,j} \in \mathcal{C}_{c_L}\}$ denote the set of sizes $s \geq 1$ of connected component node sets of $\mathcal{H}_{c_L}$. We know that $|\mathcal{S}_{c_L}| \leq S$ where $S$ is independent of $n$.

**3. Computing the lower bound:**

Let $Y$ denote the number of pairs of $k$-node sets $S_1 \not\simeq S_2$ such that $(h_u^L(H))_{u \in S_1} = (h_v^L(H))_{v \in S_2} = C = (c_{(L,1)}, ..., c_{(L,k)})$, as ordered tuples, from $L$-steps of GWL-1. Since if any pair of nodes $u, v$ have the same $L$-GWL-1 values $c_L$, then they become distinguishable by the size of the connected component in $\mathcal{H}_{c_L}$ that they belong to. We can lower bound $Y$ by counting over all pairs of $k$ tuples of nodes $((u_1, ..., u_k), (v_1, ..., v_k)) \in (\prod_{i=1}^{k} \mathcal{V}_{c_{(L,i)}}) \times (\prod_{i=1}^{k} \mathcal{V}_{c_{(L,i)}})$ that both have $L$-GWL-1 values $(c_{(L,1)}, ..., c_{(L,k)})$ where there is atleast one $i \in \{1, .., k\}$ where $u_i$ and $v_i$ belong to different sized connected components $s_i, s_i' \in \mathcal{S}_{c_{(L,i)}}$ with $s_i \neq s_i'$. We have:

$$Y \geq \frac{1}{k!}[\sum_{\substack{((s_i)_{i=1}^{k}, (s_i')_{i=1}^{k}) \in [(\prod_{i=1}^{k} \mathcal{S}_{c_{(L,i)}}^{L})]^2 \\ :(s_i)_{i=1}^{k} \neq (s_i')_{i=1}^{k}}} \prod_{i=1}^{k} |V_{(c_{(L,i)}),s_i}^{L}||V_{(c_{(L,i)}),s_i'}^{L}|] \tag{92a}$$

$$= \frac{1}{k!}[\prod_{i=1}^{k}(\sum_{s_i \in \mathcal{S}_{c_i}^{L}} |V_{(c_{(L,i)}),s_i}^{L}|)^2 - \sum_{(s_i)_{i=1}^{k} \in \prod_{i=1}^{k} \mathcal{S}_{(c_{(L,i)})}^{L}} (\prod_{i=1}^{k} |V_{(c_{(L,i)}),s_i}^{L}|^2)] \tag{92b}$$

Using the fact that for each $i \in \{1, ..., k\}$, $|\mathcal{V}_{c_{(L,i)}}| = \sum_{s_i \in \mathcal{S}_{c_{(L,i)}}} |V_{(c_{(L,i)}),s_i}|$ and by assumption $|V_{(c_{(L,i)}),s_i}| = O(\frac{n^{\epsilon}}{\log^{\frac{1}{2k}} n})$ for any $s_i \in \mathcal{S}_{c_{(L,i)}}$, thus we have:

$$Y \geq \omega(n^{2k\epsilon}) - O(|S|^k \frac{n^{2k\epsilon}}{\log n})] = \omega(n^{2k\epsilon}) \tag{93}$$

$\square$

**Example:** A simple example of a hypergraph that statisfies the conditions of Theorem B.17 is a union of many disconnected hypergraphs $\mathcal{H} = \cup_i \mathcal{H}_i = (\mathcal{V}, \mathcal{E})$ with $|\mathcal{V}_{\mathcal{H}_i}| \leq S$ where $S < \infty$ is a small constant independent of $N = |\mathcal{V}| \geq S$. Such a hypergraph could be a social network where the nodes are user instances and the hyperedges are private groups. The disconnected hypergraphs represent disconnected communities where a user can only belong to a single community.

Even though Theorem B.17 does not depend on the cardinality of a set of disconnected hypergraphs $\mathcal{H}_i$ indistinguishable by GWL-1, due to the disconnected nature of $\mathcal{H}$ and the small size of its components, there is a large chance of obtaining a large number of such components. We give a very rough estimate of this in the following:

Assuming that $\mathcal{H} = \cup_i \mathcal{H}_i$ has each $\mathcal{H}_i$ i.i.d. sampled from a distribution of $s$-uniform $d$-regular hypergraphs of $n$ nodes, denoted $\mathcal{R}_{n,s,d}$: $P(\mathcal{H}_i = \mathcal{R}_{n,s,d})$. If the parameters $(n, s, d)$ for $\mathcal{R}_{n,s,d}$ satisfy $nd = |\mathcal{E}|s$ where $|\mathcal{E}| \in \mathbb{Z}^+$, then a well defined hypergraph is formed. This distribution can be factorized as follows:

$$P(\mathcal{H}_i = \mathcal{R}_{n,s,d}) = P(deg(v) = d | r = s, |\mathcal{V}| = n, nd \mod s \equiv 0) P(r = s | |\mathcal{V}| = n) P(|\mathcal{V}| = n) \tag{94}$$

where:

$$P(deg(v) = d | r = s, |\mathcal{V}(\mathcal{H}_i)| = n, nd \mod s \equiv 0) \geq \frac{1}{\binom{n-1}{s-1}} \geq \frac{1}{\binom{n}{s}} \geq \frac{1}{S^S}, \forall v \in \mathcal{V}_{\mathcal{H}_i} \tag{95a}$$

$$P(r = s | |\mathcal{V}(\mathcal{H}_i)| = S) = \frac{1}{n} \geq \frac{1}{S}, P(|\mathcal{V}(\mathcal{H}_i)| = n) = \frac{1}{S} \text{ for } n : n \leq S \leq N \tag{95b}$$

and we have that

$$P(\mathcal{H}_1 \text{ is neighborhood regular}) \geq P(\mathcal{H}_1 \text{ is a cycle graph}) \geq \frac{1}{S}^{S+2} \tag{96}$$

and that:

$$P(h(\mathcal{H}_i, \mathcal{H}) = h(\mathcal{H}_1, \mathcal{H}), \mathcal{H}_1 \text{ is neighborhood regular}) \tag{97a}$$

$$\geq P(h(\mathcal{H}_i, \mathcal{H}) = h(\mathcal{H}_1, \mathcal{H}), \mathcal{H}_1 \text{ is a cycle graph of length } S) \tag{97b}$$

$$\geq P(\mathcal{H}_i \text{ is a cycle graph of length } S) P(\mathcal{H}_1 \text{ is a cycle graph of length } S) \tag{97c}$$

$$\geq \frac{1}{S}^{2(S+2)}, \forall i > 1 \tag{97d}$$

where a cycle graph is a 2-uniform hypergraph where each node has degree 2.

Since we sample each $\mathcal{H}_i$ i.i.d., the indicator random variable is a Bernoulli random variable. By Hoeffding's inequality on the sum of Bernoulli random variables, we get:

$$Pr(\sum_{i=1}^{m} \mathbf{1}[\mathcal{H}_i \text{ is neighborhood regular and } h(\mathcal{H}_i, \mathcal{H}) = h(\mathcal{H}_1, \mathcal{H})] \geq (\frac{m}{S^{2S+4}} + t)) \leq e^{\frac{-2t^2}{m}} \tag{98}$$

where $\sum_{i=1}^{m} |\mathcal{V}_{\mathcal{H}_i}| = |\mathcal{V}|$. This means that with large number of samples $m$, or large $n$, it is possible for the number of regular hypergraphs $\mathcal{H}_i$ equivalent up to GWL-1 to be atleast of order $\Omega(\sqrt{m}) + \frac{m}{S^{2S+4}}$ with high probability. This is one of the simplest examples that demonstrates Theorem B.17.

For the following proof, we will denote $\cong_{\mathcal{H}}$ as a node or hypergraph automorphism with respect to a hypergraph $\mathcal{H}$.

**Theorem B.18** (Invariance and Expressivity)**.** *If $L = \infty$, GWL-1 enhanced by Algorithm 1 is still invariant to node isomorphism classes of $\mathcal{H}$ and can be strictly more expressive than GWL-1 to determine node isomorphism classes.*

*Proof.*

**1. Expressivity**:

Let $L \in \mathbb{Z}^+$ be arbitrary. We first prove that $L$-GWL-1 enhanced by Algorithm 1 is strictly more expressive for node distinguishing than $L$-GWL-1 on some hypergraph(s). Let $C_4^3$ and $C_5^3$ be two 3-regular hypergraphs from Figure 2. Let $\mathcal{H} = C_4^3 \bigsqcup C_5^3$ be the disjoint union of the two regular hypergraphs. $L$ iterations of GWL-1 will assign the same node class to all of $\mathcal{V}_\mathcal{H}$. These two subhypergraphs can be distinguished by $L$-GWL-1 for $L \geq 1$ after editing the hypergraph $\mathcal{H}$ from the output of Algorithm 1 and becoming $\hat{\mathcal{H}} = \hat{C}_4^3 \cup \hat{C}_5^3$. This is all shown in Figure 2. Since $L$ was arbitrary, this is true for $L = \infty$.

**2. Invariance**:

For any hypergraph $\mathcal{H}$, let $\hat{\mathcal{H}} = (\hat{\mathcal{V}}, \hat{\mathcal{E}})$ be $\mathcal{H}$ modified by the output of Algorithm 1 by adding hyperedges to $\mathcal{V}_{\mathcal{R}_{c,i}}$. GWL-1 remains invariant to node isomorphism classes of $\mathcal{H}$ on $\hat{\mathcal{H}}$.

**a. Case 1** (node $u \in \mathcal{V}$ has its class $c$ changed to class $c_s$):

Let $L \in \mathbb{Z}^+$ be arbitrary. For any node $u$ with $L$-GWL-1 class $c$ changed to $c_s$ in $\hat{\mathcal{H}}$, if $u \cong_\mathcal{H} v$ for any $v \in \mathcal{V}$, then the GWL-1 class of $v$ must also be $c_s$. In otherwords, both $u$ and $v$ belong to $s$-sized connected components in $\mathcal{H}_c$ We prove this by contradiction.

Say $u$ belong to a $L$-GWL-1 symmetric induced subhypergraph $S$ with $|\mathcal{V}_S| = s$.

**i.** Say $v$ is originally of $L$-GWL-1 class $c$ and changes to $L$-GWL-1 $c_{s'}$ for $s' < s$ on $\hat{\mathcal{H}}$, WLOG.

If this is the case then $v$ belongs to a $L$-GWL-1 symmetric induced subhypergraph $S'$ with $|\mathcal{V}_{S'}| = s'$. Since there is a $\pi \in Aut(\mathcal{H})$ with $\pi(u) = v$ and since $s' < s$, by the pigeonhole principle some node $w \in \mathcal{V}_S$ must have $\pi(w) \notin \mathcal{V}_{S'}$. Since $S$ and $S'$ are maximally connected, $\pi(w)$ cannot share the same $L$-GWL-1 class as $w$. Thus, it must be that $(\tilde{B}_{\mathcal{V},\mathcal{E}}^{2L})_{\widetilde{\pi(w)}} \not\cong_c (\tilde{B}_{\mathcal{V},\mathcal{E}}^{2L})_{\tilde{w}}$ where $\tilde{w}, \widetilde{\pi(w)}$ are the lifts of $w, \pi(w)$ by universal covering map $p_{\mathcal{B}_{\mathcal{V},\mathcal{E}}}$. However $w$ and $\pi(w)$ both belong to $L$-GWL-1 class $c$ in $\mathcal{H}$, meaning $(\tilde{B}_{\mathcal{V},\mathcal{E}}^{2L})_{\widetilde{\pi(w)}} \cong_c (\tilde{B}_{\mathcal{V},\mathcal{E}}^{2L})_{\tilde{w}}$, contradiction.

**ii.** Say node $v \in \mathcal{V}$ has its class $c$ unchanged.

The argument for when $v$ does not change its class $c$ after the algorithm, follows by noticing that since $c$ is the GWL-1 node class of $u$, $c_s$ is the GWL-1 node class of $v$ and $c \neq c_s$. Thus we must have $u \not\cong_\mathcal{H} v$ once again by the contrapositive of Theorem B.8. This also gives a contradiciton.

Since $L$ was arbitrary, the contradiction must be true for $L = \infty$.

**b. Case 2** (node $u \in \mathcal{V}$ has its class $c$ unchanged):

Now assume $L = \infty$. Let $p_{\mathcal{B}_{\mathcal{V},\mathcal{E}}}$ be the universal covering map of $\mathcal{B}_{\mathcal{V},\mathcal{E}}$. For all other nodes $u' \cong_\mathcal{H} v'$ for $u', v' \in \mathcal{V}$ unaffected by the replacement, meaning they do not belong to any $\mathcal{R}_{c,i}$ discovered by the algorithm, if the rooted universal covering tree rooted at node $\tilde{u}'$ connects to any node $\tilde{w}$ in $l$ hops in $(\tilde{\mathcal{B}}_{\mathcal{V},\mathcal{E}}^l)_{\tilde{u}'}$ where $p_{\mathcal{B}_{\mathcal{V},\mathcal{E}}}(\tilde{u}') = u', p_{\mathcal{B}_{\mathcal{V},\mathcal{E}}}(\tilde{w}) = w$ and where $w$ has any class $c$ in $\mathcal{H}$, then $\tilde{v}'$ must also connect to a node $\tilde{z}$ in $l$ hops in $(\tilde{\mathcal{B}}_{\mathcal{V},\mathcal{E}}^l)_{\tilde{u}'}$ where $p_{\mathcal{B}_{\mathcal{V},\mathcal{E}}}(\tilde{z}) = z$ and $w \cong_\mathcal{H} z$. Furthermore, if $w$ becomes class $c_s$ in $\mathcal{H}$ due to the algorithm, then $z$ also becomes class $c_s$ in $\hat{\mathcal{H}}$. This will follow by the previous result on isomorphic $w$ and $z$ both of class $c$ with $w$ becoming class $c_s$ in $\hat{\mathcal{H}}$.

Since $L = \infty$: For any $w \in \mathcal{V}$ connected by some path of hyperedges to $u' \in \mathcal{V}$, consider the smallest $l$ for which $(\tilde{B}_{\mathcal{V},\mathcal{E}}^l)_{\tilde{u}'}$, the $l$-hop universal covering tree of $\mathcal{H}$ rooted at $\tilde{u}'$, the lift of $u'$, contains the lifted $\tilde{w}$ of $w \in \mathcal{V}$ with GWL-1 node class $c$ at layer $l$. Since $u' \cong_\mathcal{H} v'$ by $\pi$. We can use $\pi$ to find some $z = \pi(w)$.

We claim that $\tilde{z}$ is $l$ hops away from $\tilde{v}'$. Since $u' \cong_\mathcal{H} v'$ due to some $\pi \in Aut(\mathcal{H})$ with $\pi(u') = v'$, using Proposition B.2 for singleton nodes and by Theorem B.8 we must have $(\tilde{\mathcal{B}}_{\mathcal{V},\mathcal{E}}^l)_{\tilde{u}'} \cong_c (\tilde{\mathcal{B}}_{\mathcal{V},\mathcal{E}}^l)_{\tilde{v}'}$ as isomorphic rooted universal covering trees due to an induced isomorphism $\tilde{\pi}$ of $\pi$ where we define an induced isomorphism $\tilde{\pi} : (\tilde{\mathcal{B}}_{\mathcal{V},\mathcal{E}})_{\tilde{u}'} \to (\tilde{\mathcal{B}}_{\mathcal{V},\mathcal{E}})_{\tilde{v}'}$ between rooted universal covers $(\tilde{\mathcal{B}}_{\mathcal{V},\mathcal{E}})_{\tilde{u}'}$ and $(\tilde{\mathcal{B}}_{\mathcal{V},\mathcal{E}})_{\tilde{v}'}$ for $\tilde{u}', \tilde{v}' \in \mathcal{V}(\tilde{\mathcal{B}}_{\mathcal{V},\mathcal{E}})$ as $\tilde{\pi}(\tilde{a}) = \tilde{b}$ if $\pi(a) = b$ $\forall a, b \in \mathcal{V}(\mathcal{B}_{\mathcal{V},\mathcal{E}})$ connected to $u'$ and $v'$ respectively and $p_{\mathcal{B}_{\mathcal{V},\mathcal{E}}}(\tilde{a}) = a, p_{\mathcal{B}_{\mathcal{V},\mathcal{E}}}(\tilde{b}) = b$. Since $l$ is the shortest path distance from $\tilde{u}'$ to $\tilde{w}$, there must exist some shortest (as defined by the path length in $\mathcal{B}_{\mathcal{V},\mathcal{E}}$)

path $P$ of hyperedges from $u'$ to $w$ with no cycles. Using $\pi$, we must map $P$ to another acyclic shortest path of the same length from $v'$ to $z$. This path correponds to a $l$ length shortest path from $\tilde{v}'$ to $\tilde{z}$ in $(\tilde{\mathcal{B}}_{\mathcal{V},\mathcal{E}})_{\tilde{v}'}$.

If $w$ has GWL-1 class $c$ in $\mathcal{H}$ that doesn't become affected by the algorithm, then $z$ also has GWL-1 class $c$ in $\mathcal{H}$ since $w \cong_{\mathcal{H}} z$.

If $w$ has class $c$ and becomes $c_s$ in $\hat{\mathcal{H}}$, by the previous result, since $w \cong_{\mathcal{H}} z$ we must have the GWL-1 classes $c'$ and $c''$ of $w$ and $z$ in $\hat{\mathcal{H}}$ be both equal to $c_s$.

The node $w$ connected to $u'$ was arbitrary and so both $\tilde{w}$ and the isomorphism induced $\tilde{z}$ are $l$ hops away from $\tilde{u}'$ and $\tilde{v}'$ respectively, with the same GWL-1 class $c'$ in $\hat{\mathcal{H}}$, thus $(\tilde{\mathcal{B}}_{\hat{\mathcal{V}},\hat{\mathcal{E}}})_{\tilde{u}'} \cong_c (\tilde{\mathcal{B}}_{\hat{\mathcal{V}},\hat{\mathcal{E}}})_{\tilde{v}'}$.

We have thus shown, if $u \cong_{\mathcal{H}} v$ for $u, v \in \mathcal{H}$, then in $\hat{\mathcal{H}}$ we have $h_u^L(\hat{H}) = h_v^L(\hat{H})$ using the duality between universal covers and GWL-1 from Theorem B.8 and Proposition B.12. $\qquad\square$

Here we redefine the symmetry group of the $L$-GWL-1 node representation map as in the main paper:

$$Sym(h^L(\hat{H}_L)) \triangleq \bigcap_{\hat{H}'_L \sim P(\hat{H}_L)} Sym(h^L(\hat{H}'_L)) \tag{99}$$

**Proposition B.19.** *The multi-hypergraph $\hat{\mathcal{H}}_L$ breaks the symmetry of the $L$-GWL-1 view of the hypergraph $\mathcal{H}$:*

$$Sym(h^L(\hat{H}_L)) \subseteq Aut_c(\tilde{\mathcal{B}}_{\mathcal{V},\mathcal{E}}^{2L}), \forall L \geq 1 \tag{100}$$

*Proof.* By Equation 99 we have that $Sym(h^L(\hat{H}_L)) = \bigcap_{\hat{H}'_L \sim P(\hat{H}_L)} Sym(h^L(\hat{H}'_L))$. Since the original incidence matrix $H$ belongs to $supp(P(\hat{H}_L))$, we must have $\bigcap_{\hat{H}'_L \sim P(\hat{H}_L)} Sym(h^L(\hat{H}'_L)) \subseteq Sym(h^L(H))$. Since $Sym(h^L(H)) \cong Aut_c(\tilde{\mathcal{B}}_{\mathcal{V},\mathcal{E}}^{2L})$, we thus have:

$$Sym(h^L(\hat{H}_L)) = \bigcap_{\hat{H}'_L \sim P(\hat{H}_L)} Sym(h^L(\hat{H}'_L)) \subseteq Sym(h^L(H)) \cong Aut_c(\tilde{\mathcal{B}}_{\mathcal{V},\mathcal{E}}^{2L}), \forall L \geq 1 \tag{101}$$

which proves the symmetry breaking statement. $\qquad\square$

**Proposition B.20** (Complexity). *Let $H$ be the star expansion matrix of $\mathcal{H}$. Algorithm 1 runs in time $O(L \cdot nnz(H) + (n+m))$, the size of the input hypergraph when viewing $L$ as constant, where $n$ is the number of nodes, $nnz(H) = vol(\mathcal{V}) \triangleq \sum_{v \in \mathcal{V}} deg(v)$ and $m$ is the number of hyperedges.*

*Proof.* Computing $E_{deg}$, which requires computing the degrees of all the nodes in each hyperedge takes time $O(nnz(H))$. The set $E_{deg}$ can be stored as a hashset datastructure. Constructing this takes $O(nnz(H))$. Computing GWL-1 takes $O(L \cdot nnz(H))$ time assuming a constant $L$ number of iterations. Constructing the bipartite graphs for $H$ takes time $O(nnz(H) + n + m)$ since it is an information preserving data structure change. Define for each $c \in C$, $n_c := |\mathcal{V}_c|, m_c := |\mathcal{E}_c|$. Since the classes partition $\mathcal{V}$, we must have:

$$n = \sum_{c \in C} n_c; m = \sum_{c \in C} m_c; nnz(H) = \sum_{c \in C} nnz(H_c) \tag{102}$$

where $H_c$ is the star expansion matrix of $\mathcal{H}_c$. Extracting the subgraphs can be implemented as a masking operation on the nodes taking time $O(n_c)$ to form $\mathcal{V}_c$ followed by searching over the neighbors of $\mathcal{V}_c$ in time $O(m_c)$ to construct $\mathcal{E}_c$. Computing the connected components for $\mathcal{H}_c$ for a predicted node class $c$ takes time $O(n_c + m_c + nnz(H_c))$. Iterating over each connected component for a given $c$ and extracting their nodes and hyperedges takes time $O(n_{c_i} + m_{c_i})$ where $n_c = \sum_i n_{c_i}, m_c = \sum_i m_{c_i}$. Checking that a connected component has size at least 3 takes $O(1)$ time. Computing the degree on $\mathcal{H}$ for all nodes in the connected component takes time $O(n_{c_i})$ since computing degree takes $O(1)$ time. Checking that the set of node degrees of the connected component doesn't belong to $E_{deg}$ can be implemented as a check that the hash of the set of degrees is not in the hashset datastructure for $E_{deg}$.

Adding up all the time complexities, we get the total complexity is:

$$O(nnz(H)) + O(nnz(H) + n + m) + \sum_{c \in C}(O(n_c + m_c + nnz(H_c)) + \sum_{\text{conn. comp. } i \text{ of } \mathcal{H}_c} O(n_{c_i} + m_{c_i})) \quad (103a)$$

$$= O(nnz(H) + n + m) + \sum_{c \in C}(O(n_c + m_c + nnz(H_c)) + O(n_c + m_c)) \quad (103b)$$

$$= O(nnz(H) + n + m) \quad (103c)$$

$$\square$$

**Proposition B.21.** *For a connected hypergraph $\mathcal{H}$, let $(\mathcal{R}_V, \mathcal{R}_E)$ be the output of Algorithm 1 on $\mathcal{H}$. Then there are Bernoulli probabilities $p, q_i$ for $i = 1, ..., |\mathcal{R}_V|$ for attaching a covering hyperedge so that $\hat{\pi}$ is an unbiased estimator of $\pi$.*

*Proof.* Let $\mathcal{C}_{c_L} = \{\mathcal{R}_{c_L,i}\}_i$ be the maximally connected components induced by the vertices with $L$-GWL-1 values $c_L$. The set of vertex sets $\{\mathcal{V}(\mathcal{R}_{c_L,i})\}$ and the set of all hyperedges $\cup_i\{\mathcal{E}(\mathcal{R}_{c_L,i})\}$ over all the connected components $\mathcal{R}_{c_L,i}$ for $i = 1, ..., \mathcal{C}_{c_L}$ form the pair $(\mathcal{R}_V, \mathcal{R}_E)$.

For a hypergraph random walk on connected $\mathcal{H} = (\mathcal{V}, \mathcal{E})$, its stationary distribution $\pi$ on $\mathcal{V}$ is given by the closed form:

$$\pi(v) = \frac{deg(v)}{\sum_{u \in \mathcal{V}} deg(u)} \quad (104)$$

for $v \in \mathcal{V}$.

Let $\hat{\mathcal{H}} = (\mathcal{V}, \hat{\mathcal{E}})$ be the random multi-hypergraph as determined by $p$ and $q_i$ for $i = 1, ..., |\mathcal{R}_V|$. These probabilities determine $\hat{\mathcal{H}}$ by the following operations on the hypergraph $\mathcal{H}$:

- Attaching a single hyperedge that covers $\mathcal{V}_{\mathcal{R}_{c_L,i}}$ with probability $q_i$ and not attaching with probability $1 - q_i$.

- All the hyperedges in $\mathcal{R}_{c_L,i}$ are dropped/kept with probability $p$ and $1 - p$ respectively.

**1. Setup:**

Let $deg(v) \triangleq |\{e : e \ni v\}|$ for $v \in \mathcal{V}(\mathcal{H})$ and $deg(\mathcal{S}) \triangleq \sum_{u \in \mathcal{V}(\mathcal{S})} deg(u)$ for $\mathcal{S} \subseteq \mathcal{H}$ a subhypergraph.

Let

$$Bernoulli(p) \triangleq \begin{cases} 1 & \text{prob. } p \\ 0 & \text{prob. } 1 - p \end{cases} \quad (105)$$

and

$$Binom(n, p) \triangleq \sum_{i=1}^{n} Bernoulli(p) \quad (106)$$

Define for each $v \in \mathcal{V}$, $C(v)$ to be the unique $\mathcal{R}_{c_L,i}$ where $v \in \mathcal{R}_{c_L,i}$ This means that we have the following independent random variables:

$$X_e \triangleq Bernoulli(1 - p), \forall e \in \mathcal{E} \text{ (i.i.d. across all } e \in \mathcal{E}) \quad (107a)$$

$$X_{C(v)} \triangleq Bernoulli(q_i) \quad (107b)$$

As well as the following constant, depending only on $C(v)$:

$$m_{C(v)} \triangleq \sum_{u \in \mathcal{V} \setminus C(v)} deg(u) \quad (108)$$

where $C(v) \subseteq \mathcal{V}, \forall v \in \mathcal{V}$

Let $\hat{\pi}$ be the stationary distribution of $\hat{H}$. Its expectation $\mathbb{E}[\hat{\pi}]$ can be written as:

$$\hat{\pi}(v) \triangleq \frac{\sum_{e \ni v: e \in \mathcal{E}} X_e + X_{C(v)}}{m_{C(v)} + \sum_{e \ni u: u \in C(v), e \in \mathcal{E}} X_e + X_{C(v)}} \tag{109}$$

Letting

$$N_v \triangleq \sum_{e \ni v: e \in \mathcal{E}} X_e = Binom(deg(v), 1-p) \tag{110a}$$

$$N \triangleq N_v + X_{C(v)} \tag{110b}$$

$$D \triangleq m_{C(v)} + \sum_{e \ni u: u \in C(v), e \in \mathcal{E}} X_e + X_{C(v)} \tag{110c}$$

$$C \triangleq D - (\sum_{e \ni v: e \in \mathcal{E}} |e|) N_v - m_{C(v)} = \sum_{e \ni u: v \notin e, u \in C(v), e \in \mathcal{E}} X_e - m_{C(v)} = Binom(F_v, 1-p) - m_{c(v)} \tag{110d}$$

$$\text{where } F_v \triangleq |\{e \ni u : v \notin e, u \in C(v), e \in \mathcal{E}\}|$$

and so we have: $\hat{\pi}(v) = \frac{N}{D}$

We have the following joint independence $N_v \perp X_{C(v)} \perp C$ due to the fact that each random variable describes disjoint hyperedge sets.

**2. Computing the Expectation:**

Writing out the expectation with conditioning on the joint distribution $P(D, N_v, X_{C(v)})$, we have:

$$\mathbb{E}[\hat{\pi}(v)] = \sum_{b=0}^{1} \sum_{j=0}^{deg(v)} \sum_{k=m_{C(v)}}^{deg(C(v))} \mathbb{E}[\hat{\pi}(v)|D=k, N=j]P(D=k, N_v=j, X_{C(v)}=b) \tag{111a}$$

$$= \sum_{b=0}^{1} \sum_{j=0}^{deg(v)} \sum_{k=m_{C(v)}}^{deg(C(v))} \frac{1}{k}\mathbb{E}[N|D=k, N_v=j, X_{C(v)}=b]P(D=k, N_v=j, X_{C(v)}=b) \tag{111b}$$

$$= \sum_{b=0}^{1} \sum_{j=0}^{deg(v)} \sum_{k=m_{C(v)}}^{deg(C(v))} \frac{j+b}{k}P(D=k, N_v=j, X_{C(v)}=b) \tag{111c}$$

$$= \sum_{b=0}^{1} \sum_{j=0}^{deg(v)} \sum_{k=m_{C(v)}}^{deg(C(v))} \frac{j+b}{k}P(D=k)P(N_v=j)P(X_{C(v)}=b) \tag{111d}$$

$$= \sum_{b=0}^{1} \sum_{j=0}^{deg(v)} \sum_{k=m_{C(v)}}^{deg(C(v))} \frac{j+b}{k}P(C=k-deg(v)j-m_{C(v)})P(N_v=j)P(X_{C(v)}=b) \tag{111e}$$

$$= \sum_{b=0}^{1} \sum_{j=0}^{deg(v)} \sum_{k=m_{C(v)}}^{deg(C(v))} \frac{j+b}{k}P(Binom(F_v, 1-p), 1-p) = k-deg(v)j-m_{C(v)})$$
$$\cdot P(Binom(deg(v), 1-p)=j) \cdot P(Bernoulli(q_i)=b) \tag{111f}$$

$$= \sum_{b=0}^{1} \sum_{j=0}^{deg(v)} \sum_{k=m_{C(v)}}^{deg(C(v))} \frac{j+b}{k}\binom{F_v}{k-deg(v)j-m_{C(v)}}(\frac{1}{2})^{F_v} \cdot \binom{deg(v)}{j}(\frac{1}{2})^{deg(v)} \cdot P(Bernoulli(q_i)=b) \tag{111g}$$

$$= \sum_{b=0}^{1} \sum_{j=0}^{deg(v)} \sum_{k=m_{C(v)}}^{deg(C(v))} \frac{j+b}{k}\binom{F_v}{k-deg(v)j-m_{C(v)}}(\frac{1}{2})^{F_v}$$
$$\cdot \binom{deg(v)}{j}(\frac{1}{2})^{deg(v)} \cdot P(Bernoulli(q_i)=b) \tag{111h}$$

$$= \sum_{j=0}^{deg(v)} \sum_{k=m_{C(v)}}^{deg(C(v))} \binom{F_v}{k - deg(v)j - m_{C(v)}} (\frac{1}{2})^{F_v} \cdot \binom{deg(v)}{j} (\frac{1}{2})^{deg(v)} [(1 - q_i)\frac{j}{k} + q_i \frac{j+1}{k}] \tag{111i}$$

$$= \sum_{j=0}^{deg(v)} \sum_{k=m_{C(v)}}^{deg(C(v))} \binom{F_v}{k - deg(v)j - m_{C(v)}} (1 - p)^{F_v - (k - deg(v))} p^{k - deg(v)} \cdot \binom{deg(v)}{j} (1 - p)^j p^{deg(v) - j} [\frac{j}{k} + q_i \frac{1}{k}] \tag{111j}$$

$$= C_1(p) + q_i C_2(p) \tag{111k}$$

**3. Pick $p$ and $q_i$:**

We want to find $p$ and $q_i$ so that $\mathbb{E}[\hat{\pi}(v)] = C_1(p) + q_i C_2(p) = \pi(v)$

We know that for a given $p \in [0, 1]$, we must have:

$$q_i = \frac{\pi(v) - C_1(p)}{C_2(p)} \tag{112}$$

In order for $q_i \in [0, 1]$, must have $\pi(v) \geq C_1(p)$ and $\pi(v) - C_1(p) \leq C_2(p)$.

**a. Pick $p$ sufficiently large:**

Notice that

$$0 \leq C_1(p) \leq O(\mathbb{E}[\frac{1}{Binom(F_v, 1 - p) + m_{C(v)}}] \cdot \mathbb{E}[Binom(deg(v), 1 - p)]) = O(\frac{1}{m_{C(v)}} deg(v)(1 - p)) \tag{113}$$

and that

$$0 \leq C_1(p) \leq O(C_2(p)) \tag{114}$$

for $p \in [0, 1]$ sufficiently large. This is because

$$C_1(p) \leq O(\frac{1}{m_{C(v)}} deg(v)(1 - p)) \tag{115}$$

and

$$\Omega(\frac{1}{m_{C(v)}} deg(v)(1 - p)) \leq C_2(p) \tag{116}$$

Piecing these two inequalities together gets the desired inequality 114.

We can then pick a $p \in [0, 1]$ even larger than the previous $p$ so that for the $C' > 0$ which gives $C_1(p) \leq \frac{C'}{m_{C(v)}} deg(v)(1 - p)$, we achieve

$$C_1(p) \leq \frac{C'}{m_{C(v)}} deg(v)(1 - p) < \pi(v) = \frac{deg(v)}{m_{C(v)} + \sum_{u \in C(v)} deg(u)} \tag{117}$$

We then have that there exists a $s > 1$ so that

$$sC_1(p) = \pi(v) \tag{118}$$

Using this relationship, we can then prove that for a sufficiently large $p \in [0, 1]$, we must have a $q_i \in [0, 1]$

**b. $p \in [0, 1]$ sufficiently large implies $q_i \geq 0$:**

We thus have $q_i \geq 0$ since its numerator is nonnegative:

$$\pi(v) - C_1(p) = (s - 1)C_1(p) \geq 0 \Rightarrow q_i \geq 0 \tag{119}$$

**c.** $p \in [0, 1]$ **sufficiently large implies** $q_i \leq 1$:

$$\pi(v) - C_1(p) = sC_1(p) - C_1(p) = (s-1)C_1(p) \leq C_2(p) \Rightarrow q_i \leq 1 \tag{120}$$

$\square$

## C  Additional Experiments

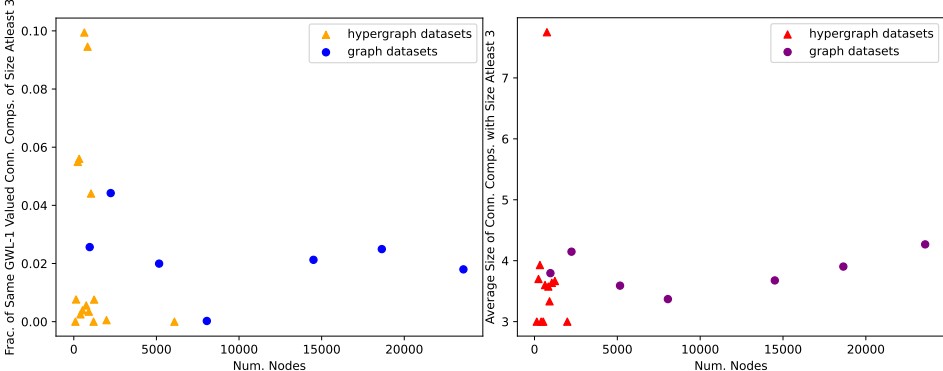

(a) Fraction of components of size atleast 3 selected by Algorithm 1.

(b) Average size of components of size atleast 3 from Algorithm 1.

Figure 6

As we are primarily concerned with symmetries in a hypergraph, we empirically measure the size and frequency of the components found by the Algorithm for real-world datasets. For the real-world datasets listed in Appendix D, in Figure 6a, we plot the fraction of connected components of the same $L$-GWL-1 value ($L = 2$) that are atleast 3 in cardinality from Algorithm 1 as a function of the number of nodes of the hypergraph. For these datasets, it is much more common for the connected components to be of sizes 1 and 2. On the right, in Figure 6b we show the distribution of the sizes of the connected components found by Algorithm 1. We see that, on average, the connected components are at least an order of magnitude smaller compared to the total number of nodes. Common to both plots, the graph datasets appear to have more nodes and a consistent fraction and size of components, while the hypergraph datasets have higher variance in the fraction of components, which is expected since there are more possibilities for the connections in a hypergraph.

We show below the critical difference diagrams Demšar (2006) of the average PR-AUC score ranks (percentiles) for two datasets from Table 1 across all the downstream hyperGNNs. Each plot contains the PR-AUC ranks of the three compared approaches: baseline, 50% hyperedge drop, and Our method. The bars in each plot represents statistical insignificance between the two approaches according to 4 runs of each experiment.

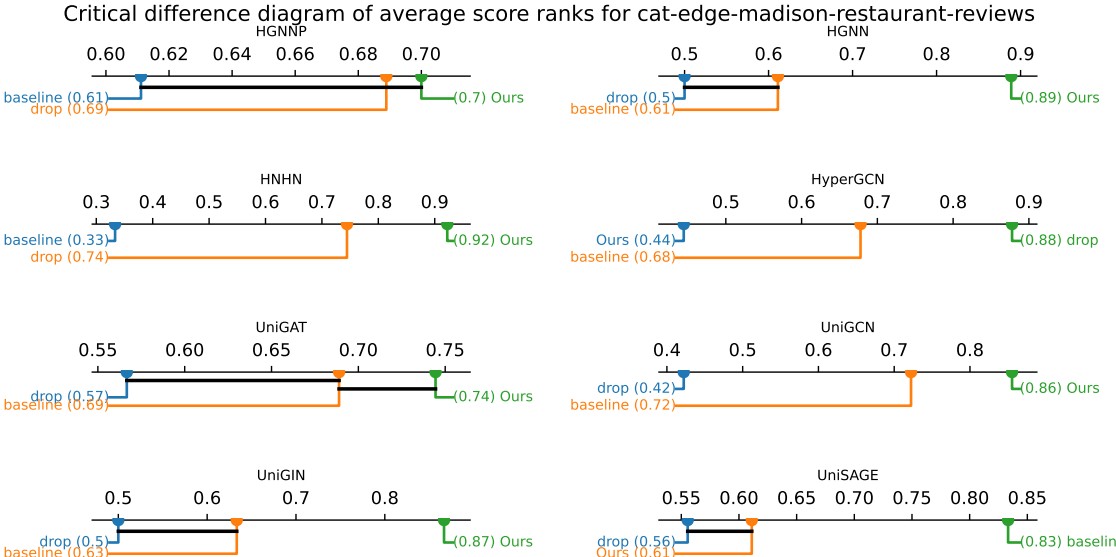

Figure 7: Critical difference diagrams of CAT-EDGE-MADISON-RESTAURANT-REVIEWS

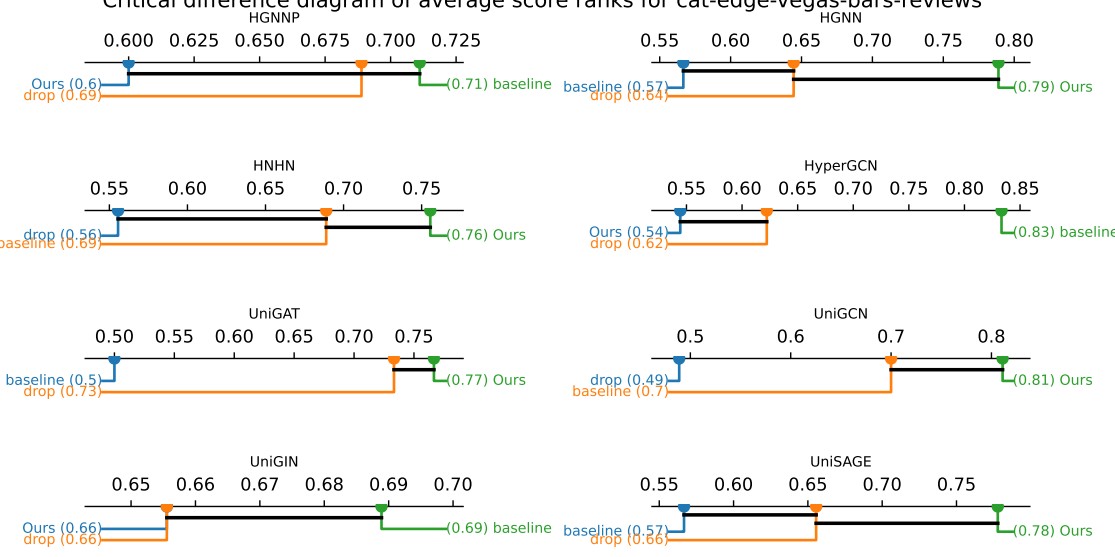

Figure 8: Critical difference diagrams of CAT-EDGE-VEGAS-BARS-REVIEWS

## C.1 Additional Experiments on Hypergraphs

In Table 3, we show the PR-AUC scores for four additional hypergraph datasets, CAT-EDGE-BRAIN, CAT-EDGE-VEGAS-BAR-REVIEWS, WIKIPEOPLE-0BI, and JF17K for predicting size 3 hyperedges.

Table 3: PR-AUC on four other hypergraph datasets. The top average scores for each hyperGNN method, or row, is colored. Red scores denote the top scores in a row. Orange scores denote a two way tie and brown scores denote a threeway tie.

| PR-AUC ↑ | Baseline | Ours | Baseln.+edrop |
|---|---|---|---|
| HGNN | $0.75 \pm 0.01$ | $0.79 \pm 0.11$ | $0.74 \pm 0.09$ |
| HGNNP | $0.75 \pm 0.05$ | $0.78 \pm 0.10$ | $0.74 \pm 0.12$ |
| HNHN | $0.74 \pm 0.04$ | $0.74 \pm 0.02$ | $0.74 \pm 0.05$ |
| HyperGCN | $0.74 \pm 0.09$ | $0.50 \pm 0.07$ | $0.50 \pm 0.12$ |
| UniGAT | $0.73 \pm 0.07$ | $0.81 \pm 0.10$ | $0.81 \pm 0.09$ |
| UniGCN | $0.78 \pm 0.04$ | $0.81 \pm 0.09$ | $0.71 \pm 0.08$ |
| UniGIN | $0.74 \pm 0.09$ | $0.74 \pm 0.03$ | $0.74 \pm 0.07$ |
| UniSAGE | $0.74 \pm 0.03$ | $0.74 \pm 0.12$ | $0.74 \pm 0.01$ |

(a) CAT-EDGE-BRAIN

| PR-AUC ↑ | Baseline | Ours | Baseln.+edrop |
|---|---|---|---|
| HGNN | $0.95 \pm 0.10$ | $0.99 \pm 0.04$ | $0.96 \pm 0.09$ |
| HGNNP | $0.95 \pm 0.06$ | $0.96 \pm 0.09$ | $0.96 \pm 0.08$ |
| HNHN | $1.00 \pm 0.08$ | $0.99 \pm 0.09$ | $0.95 \pm 0.10$ |
| HyperGCN | $0.76 \pm 0.03$ | $0.67 \pm 0.14$ | $0.68 \pm 0.09$ |
| UniGAT | $0.87 \pm 0.07$ | $1.00 \pm 0.09$ | $0.99 \pm 0.08$ |
| UniGCN | $0.99 \pm 0.07$ | $0.96 \pm 0.09$ | $0.92 \pm 0.05$ |
| UniGIN | $0.98 \pm 0.06$ | $0.96 \pm 0.08$ | $0.95 \pm 0.06$ |
| UniSAGE | $0.94 \pm 0.05$ | $0.98 \pm 0.07$ | $0.97 \pm 0.07$ |

(b) CAT-EDGE-VEGAS-BAR-REVIEWS

| PR-AUC ↑ | Baseline | Ours | Baseln.+edrop |
|---|---|---|---|
| HGNN | $0.52 \pm 0.01$ | $0.57 \pm 0.08$ | $0.54 \pm 0.10$ |
| HGNNP | $0.52 \pm 0.03$ | $0.54 \pm 0.07$ | $0.54 \pm 0.06$ |
| HNHN | $0.73 \pm 0.03$ | $0.73 \pm 0.07$ | $0.73 \pm 0.00$ |
| HyperGCN | $0.54 \pm 0.05$ | $0.55 \pm 0.02$ | $0.49 \pm 0.10$ |
| UniGAT | $0.49 \pm 0.09$ | $0.54 \pm 0.04$ | $0.53 \pm 0.04$ |
| UniGCN | $0.46 \pm 0.08$ | $0.68 \pm 0.08$ | $0.51 \pm 0.08$ |
| UniGIN | $0.73 \pm 0.09$ | $0.73 \pm 0.01$ | $0.73 \pm 0.02$ |
| UniSAGE | $0.73 \pm 0.06$ | $0.73 \pm 0.02$ | $0.73 \pm 0.08$ |

(c) WIKIPEOPLE-0BI

| PR-AUC ↑ | Baseline | Ours | Baseln.+edrop |
|---|---|---|---|
| HGNN | $0.59 \pm 0.04$ | $0.63 \pm 0.04$ | $0.45 \pm 0.09$ |
| HGNNP | $0.71 \pm 0.07$ | $0.63 \pm 0.07$ | $0.57 \pm 0.04$ |
| HNHN | $0.73 \pm 0.04$ | $0.73 \pm 0.03$ | $0.73 \pm 0.04$ |
| HyperGCN | $0.59 \pm 0.05$ | $0.58 \pm 0.09$ | $0.48 \pm 0.01$ |
| UniGAT | $0.61 \pm 0.07$ | $0.61 \pm 0.04$ | $0.51 \pm 0.08$ |
| UniGCN | $0.58 \pm 0.00$ | $0.60 \pm 0.03$ | $0.59 \pm 0.02$ |
| UniGIN | $0.80 \pm 0.04$ | $0.77 \pm 0.08$ | $0.75 \pm 0.05$ |
| UniSAGE | $0.79 \pm 0.02$ | $0.77 \pm 0.08$ | $0.74 \pm 0.01$ |

(d) JF17K

## C.2 Experiments on Graph Data

We show in Tables 4, 5, 6 the PR-AUC test scores for link prediction on some nonattributed graph datasets. The train-val-test splits are predefined for FB15K-237 and for the other graph datasets a single graph is deterministically split into 80/5/15 for train/val/test. We remove 10% of the edges in training and let them be positive examples $P_{tr}$ to predict. For validation and test, we remove 50% of the edges from both validation and test to set as the positive examples $P_{val}, P_{te}$ to predict. For train, validation, and test, we sample $1.2|P_{tr}|, 1.2|P_{val}|, 1.2|P_{te}|$ negative link samples from the links of train, validation and test. Along with hyperGNN architectures we use for the hypergraph experiments, we also compare with standard GNN architectures: APPNP Gasteiger et al. (2018), GAT Veličković et al. (2017), GCN2 Chen et al. (2020a), GCN Kipf & Welling (2016a), GIN Xu et al. (2018), and GraphSAGE Hamilton et al. (2017). For every hyperGNN/GNN architecture, we also apply drop-edge Rong et al. (2019) to the input graph and use this also as baseline. The number of layers of each GNN is set to 5 and the hidden dimension at 1024. For APPNP and GCN2, one MLP is used on the initial node positional encodings. Since graphs do not have any hyperedges beyond size 2, graph neural networks fit the inductive bias of the graph data more easily and thus may perform better than hypergraph neural network baselines more often than expected.

Table 4: PR-AUC on graph dataset JOHNSHOPKINS55. Each column is a comparison of the baseline PR-AUC scores against the PR-AUC score for our method (first row) applied to a standard hyperGNN architecture. Red color denotes the highest average score in the column. Orange color denotes a two-way tie in the column, and brown color denotes a three-or-more-way tie in the column.

| PR-AUC ↑ | HGNN | HGNNP | HNHN | HyperGCN | UniGAT | UniGCN | UniGIN | UniSAGE |
|---|---|---|---|---|---|---|---|---|
| Ours | 0.71 ± 0.04 | 0.71 ± 0.09 | 0.69 ± 0.09 | 0.75 ± 0.14 | 0.75 ± 0.09 | 0.74 ± 0.09 | 0.65 ± 0.08 | 0.65 ± 0.07 |
| hyperGNN Baseline | 0.68 ± 0.00 | 0.69 ± 0.06 | 0.67 ± 0.02 | 0.75 ± 0.04 | 0.74 ± 0.02 | 0.74 ± 0.00 | 0.65 ± 0.05 | 0.64 ± 0.08 |
| hyperGNN Baseln.+edrop | 0.67 ± 0.02 | 0.70 ± 0.07 | 0.66 ± 0.00 | 0.75 ± 0.03 | 0.73 ± 0.08 | 0.74 ± 0.05 | 0.63 ± 0.01 | 0.64 ± 0.03 |
| APPNP | 0.40 ± 0.03 | 0.40 ± 0.03 | 0.40 ± 0.03 | 0.40 ± 0.03 | 0.40 ± 0.03 | 0.40 ± 0.03 | 0.40 ± 0.03 | 0.40 ± 0.03 |
| APPNP+edrop | 0.40 ± 0.13 | 0.40 ± 0.13 | 0.40 ± 0.13 | 0.40 ± 0.13 | 0.40 ± 0.13 | 0.40 ± 0.13 | 0.40 ± 0.13 | 0.40 ± 0.13 |
| GAT | 0.49 ± 0.03 | 0.49 ± 0.03 | 0.49 ± 0.03 | 0.49 ± 0.03 | 0.49 ± 0.03 | 0.49 ± 0.03 | 0.49 ± 0.03 | 0.49 ± 0.03 |
| GAT+edrop | 0.51 ± 0.05 | 0.51 ± 0.05 | 0.51 ± 0.05 | 0.51 ± 0.05 | 0.51 ± 0.05 | 0.51 ± 0.05 | 0.51 ± 0.05 | 0.51 ± 0.05 |
| GCN2 | 0.50 ± 0.09 | 0.50 ± 0.09 | 0.50 ± 0.09 | 0.50 ± 0.09 | 0.50 ± 0.09 | 0.50 ± 0.09 | 0.50 ± 0.09 | 0.50 ± 0.09 |
| GCN2+edrop | 0.56 ± 0.07 | 0.56 ± 0.07 | 0.56 ± 0.07 | 0.56 ± 0.07 | 0.56 ± 0.07 | 0.56 ± 0.07 | 0.56 ± 0.07 | 0.56 ± 0.07 |
| GCN | 0.73 ± 0.02 | 0.73 ± 0.02 | 0.73 ± 0.02 | 0.73 ± 0.02 | 0.73 ± 0.02 | 0.73 ± 0.02 | 0.73 ± 0.02 | 0.73 ± 0.02 |
| GCN+edrop | 0.73 ± 0.01 | 0.73 ± 0.01 | 0.73 ± 0.01 | 0.73 ± 0.01 | 0.73 ± 0.01 | 0.73 ± 0.01 | 0.73 ± 0.01 | 0.73 ± 0.01 |
| GIN | 0.73 ± 0.06 | 0.73 ± 0.06 | 0.73 ± 0.06 | 0.73 ± 0.06 | 0.73 ± 0.06 | 0.73 ± 0.06 | 0.73 ± 0.06 | 0.73 ± 0.06 |
| GIN+edrop | 0.73 ± 0.01 | 0.73 ± 0.01 | 0.73 ± 0.01 | 0.73 ± 0.01 | 0.73 ± 0.01 | 0.73 ± 0.01 | 0.73 ± 0.01 | 0.73 ± 0.01 |
| GraphSAGE | 0.73 ± 0.08 | 0.73 ± 0.08 | 0.73 ± 0.08 | 0.73 ± 0.08 | 0.73 ± 0.08 | 0.73 ± 0.08 | 0.73 ± 0.08 | 0.73 ± 0.08 |
| GraphSAGE+edrop | 0.73 ± 0.09 | 0.73 ± 0.09 | 0.73 ± 0.09 | 0.73 ± 0.09 | 0.73 ± 0.09 | 0.73 ± 0.09 | 0.73 ± 0.09 | 0.73 ± 0.09 |

Table 5: PR-AUC on graph dataset FB15K-237. Each column is a comparison of the baseline PR-AUC scores against the PR-AUC score for our method (first row) applied to a standard hyperGNN architecture. Red color denotes the highest average score in the column. Orange color denotes a two-way tie in the column, and brown color denotes a three-or-more-way tie in the column.

| PR-AUC ↑ | HGNN | HGNNP | HNHN | HyperGCN | UniGAT | UniGCN | UniGIN | UniSAGE |
|---|---|---|---|---|---|---|---|---|
| Ours | 0.66 ± 0.06 | 0.78 ± 0.02 | 0.63 ± 0.07 | 0.82 ± 0.10 | 0.75 ± 0.05 | 0.74 ± 0.03 | 0.75 ± 0.03 | 0.75 ± 0.06 |
| hyperGNN Baseline | 0.65 ± 0.06 | 0.65 ± 0.06 | 0.65 ± 0.04 | 0.82 ± 0.09 | 0.74 ± 0.04 | 0.74 ± 0.05 | 0.75 ± 0.03 | 0.77 ± 0.01 |
| hyperGNN Baseln.+edrop | 0.65 ± 0.09 | 0.65 ± 0.00 | 0.64 ± 0.05 | 0.82 ± 0.00 | 0.72 ± 0.00 | 0.74 ± 0.07 | 0.73 ± 0.03 | 0.72 ± 0.07 |
| APPNP | 0.72 ± 0.10 | 0.72 ± 0.10 | 0.72 ± 0.10 | 0.72 ± 0.10 | 0.72 ± 0.10 | 0.72 ± 0.10 | 0.72 ± 0.10 | 0.72 ± 0.10 |
| APPNP+edrop | 0.71 ± 0.05 | 0.71 ± 0.05 | 0.71 ± 0.05 | 0.71 ± 0.05 | 0.71 ± 0.05 | 0.71 ± 0.05 | 0.71 ± 0.05 | 0.71 ± 0.05 |
| GAT | 0.64 ± 0.06 | 0.64 ± 0.06 | 0.64 ± 0.06 | 0.64 ± 0.06 | 0.64 ± 0.06 | 0.64 ± 0.06 | 0.64 ± 0.06 | 0.64 ± 0.06 |
| GAT+edrop | 0.61 ± 0.09 | 0.61 ± 0.09 | 0.61 ± 0.09 | 0.61 ± 0.09 | 0.61 ± 0.09 | 0.61 ± 0.09 | 0.61 ± 0.09 | 0.61 ± 0.09 |
| GCN2 | 0.66 ± 0.03 | 0.66 ± 0.03 | 0.66 ± 0.03 | 0.66 ± 0.03 | 0.66 ± 0.03 | 0.66 ± 0.03 | 0.66 ± 0.03 | 0.66 ± 0.03 |
| GCN2+edrop | 0.65 ± 0.10 | 0.65 ± 0.10 | 0.65 ± 0.10 | 0.65 ± 0.10 | 0.65 ± 0.10 | 0.65 ± 0.10 | 0.65 ± 0.10 | 0.65 ± 0.10 |
| GCN | 0.69 ± 0.03 | 0.69 ± 0.03 | 0.69 ± 0.03 | 0.69 ± 0.03 | 0.69 ± 0.03 | 0.69 ± 0.03 | 0.69 ± 0.03 | 0.69 ± 0.03 |
| GCN+edrop | 0.71 ± 0.06 | 0.71 ± 0.06 | 0.71 ± 0.06 | 0.71 ± 0.06 | 0.71 ± 0.06 | 0.71 ± 0.06 | 0.71 ± 0.06 | 0.71 ± 0.06 |
| GIN | 0.73 ± 0.03 | 0.73 ± 0.03 | 0.73 ± 0.03 | 0.73 ± 0.03 | 0.73 ± 0.03 | 0.73 ± 0.03 | 0.73 ± 0.03 | 0.73 ± 0.03 |
| GIN+edrop | 0.56 ± 0.07 | 0.56 ± 0.07 | 0.56 ± 0.07 | 0.56 ± 0.07 | 0.56 ± 0.07 | 0.56 ± 0.07 | 0.56 ± 0.07 | 0.56 ± 0.07 |
| GraphSAGE | 0.46 ± 0.15 | 0.46 ± 0.15 | 0.46 ± 0.15 | 0.46 ± 0.15 | 0.46 ± 0.15 | 0.46 ± 0.15 | 0.46 ± 0.15 | 0.46 ± 0.15 |
| GraphSAGE+edrop | 0.47 ± 0.01 | 0.47 ± 0.01 | 0.47 ± 0.01 | 0.47 ± 0.01 | 0.47 ± 0.01 | 0.47 ± 0.01 | 0.47 ± 0.01 | 0.47 ± 0.01 |

Table 6: PR-AUC on graph dataset AIFB. Each column is a comparison of the baseline PR-AUC scores against the PR-AUC score for our method (first row) applied to a standard hyperGNN architecture. Red color denotes the highest average score in the column. Orange color denotes a two-way tie in the column, and brown color denotes a three-or-more-way tie in the column.

| PR-AUC ↑ | HGNN | HGNNP | HNHN | HyperGCN | UniGAT | UniGCN | UniGIN | UniSAGE |
|---|---|---|---|---|---|---|---|---|
| Ours | 0.79 ± 0.11 | 0.73 ± 0.10 | 0.73 ± 0.02 | 0.85 ± 0.07 | 0.75 ± 0.10 | 0.84 ± 0.09 | 0.72 ± 0.03 | 0.72 ± 0.12 |
| hyperGNN Baseline | 0.72 ± 0.07 | 0.72 ± 0.07 | 0.72 ± 0.06 | 0.85 ± 0.05 | 0.75 ± 0.09 | 0.84 ± 0.05 | 0.72 ± 0.07 | 0.72 ± 0.06 |
| hyperGNN Baseln.+edrop | 0.72 ± 0.05 | 0.72 ± 0.08 | 0.72 ± 0.06 | 0.85 ± 0.07 | 0.73 ± 0.09 | 0.84 ± 0.06 | 0.72 ± 0.03 | 0.72 ± 0.07 |
| APPNP | 0.81 ± 0.12 | 0.81 ± 0.12 | 0.81 ± 0.12 | 0.81 ± 0.12 | 0.81 ± 0.12 | 0.81 ± 0.12 | 0.81 ± 0.12 | 0.81 ± 0.12 |
| APPNP+edrop | 0.80 ± 0.05 | 0.80 ± 0.05 | 0.80 ± 0.05 | 0.80 ± 0.05 | 0.80 ± 0.05 | 0.80 ± 0.05 | 0.80 ± 0.05 | 0.80 ± 0.05 |
| GAT | 0.50 ± 0.02 | 0.50 ± 0.02 | 0.50 ± 0.02 | 0.50 ± 0.02 | 0.50 ± 0.02 | 0.50 ± 0.02 | 0.50 ± 0.02 | 0.50 ± 0.02 |
| GAT+edrop | 0.33 ± 0.02 | 0.33 ± 0.02 | 0.33 ± 0.02 | 0.33 ± 0.02 | 0.33 ± 0.02 | 0.33 ± 0.02 | 0.33 ± 0.02 | 0.33 ± 0.02 |
| GCN2 | 0.83 ± 0.05 | 0.83 ± 0.05 | 0.83 ± 0.05 | 0.83 ± 0.05 | 0.83 ± 0.05 | 0.83 ± 0.05 | 0.83 ± 0.05 | 0.83 ± 0.05 |
| GCN2+edrop | 0.78 ± 0.04 | 0.78 ± 0.04 | 0.78 ± 0.04 | 0.78 ± 0.04 | 0.78 ± 0.04 | 0.78 ± 0.04 | 0.78 ± 0.04 | 0.78 ± 0.04 |
| GCN | 0.73 ± 0.14 | 0.73 ± 0.14 | 0.73 ± 0.14 | 0.73 ± 0.14 | 0.73 ± 0.14 | 0.73 ± 0.14 | 0.73 ± 0.14 | 0.73 ± 0.14 |
| GCN+edrop | 0.75 ± 0.08 | 0.75 ± 0.08 | 0.75 ± 0.08 | 0.75 ± 0.08 | 0.75 ± 0.08 | 0.75 ± 0.08 | 0.75 ± 0.08 | 0.75 ± 0.08 |
| GIN | 0.73 ± 0.00 | 0.73 ± 0.00 | 0.73 ± 0.00 | 0.73 ± 0.00 | 0.73 ± 0.00 | 0.73 ± 0.00 | 0.73 ± 0.00 | 0.73 ± 0.00 |
| GIN+edrop | 0.73 ± 0.10 | 0.73 ± 0.10 | 0.73 ± 0.10 | 0.73 ± 0.10 | 0.73 ± 0.10 | 0.73 ± 0.10 | 0.73 ± 0.10 | 0.73 ± 0.10 |
| GraphSAGE | 0.46 ± 0.15 | 0.46 ± 0.15 | 0.46 ± 0.15 | 0.46 ± 0.15 | 0.46 ± 0.15 | 0.46 ± 0.15 | 0.46 ± 0.15 | 0.46 ± 0.15 |
| GraphSAGE+edrop | 0.47 ± 0.01 | 0.47 ± 0.01 | 0.47 ± 0.01 | 0.47 ± 0.01 | 0.47 ± 0.01 | 0.47 ± 0.01 | 0.47 ± 0.01 | 0.47 ± 0.01 |

# D  Dataset and Hyperparameters

Table 7 lists the datasets and hyperparameters used in our experiments. All datasets are originally from Benson et al. (2018b) or are general hypergraph datasets provided in Sinha et al. (2015); Amburg et al. (2020a). We list the total number of hyperedges $|\mathcal{E}|$, the total number of vertices $|\mathcal{V}|$, the positive to negative label ratios for train/val/test, and the percentage of the connected components searched over by our algorithm that are size atleast 3. A node isomorphism class is determined by our isomorphism testing algorithm. By Proposition B.2 we can guarantee that if two nodes are in separate isomorphism classes by our isomorphism tester, then they are actually nonisomorphic.

We use 1024 dimensions for all hyperGNN/GNN layer latent spaces, 5 layers for all hypergraph/graph neural networks, and a common learning rate of 0.01. Exactly 2000 epochs are used for training.

The hyperGNN architecture baselines are described in the follwoing:

- HGNN Feng et al. (2019) A neural network that generalizes the graph convolution to hypergraphs where there are hyperedge weights. Its architecture can be described by the following update step for the $l + 1$-layer from the $l$th layer:

$$X^{(l+1)} = \sigma(D_v^{-\frac{1}{2}} HW D_e^{-1} H^T D_v^{-\frac{1}{2}} X^{(l)} W^{(l)})  \tag{121}$$

  where $D_v \in \mathbb{R}^{n \times n}$ is the diagonal node degree matrix, $D_e \in \mathbb{R}^{m \times m}$ is the diagonal hyperedge degree matrix, $H \in \mathbb{R}^{n \times m}$ is the star incidence matrix, $W$ is the diagonal hyperedge weight matrix, $X^{(l)} \in \mathbb{R}^{n \times d}$ is a node signal matrix, $W^{(l)} \in \mathbb{R}^{d \times d}$ is a weight matrix, and $\sigma$ is a nonlinear activation. Following the matrix products, as a message passing neural network, HGNN is GWL-1 based since the nodes pass to the hyperedges and back.

- HGNNP Feng et al. (2023) is an improved version of HGNN where asymmetry is introduced into the message passing weightings to distinguish the vertices from the hyperedges. This is also a GWL-1 based message passing neural network. It is described by the following node signal update equation:

$$X^{(l+1)} = \sigma(D_v^{-1} HW D_e^{-1} H^T X^{(l)} W^{(l)})  \tag{122}$$

  where the matrices are exactly the same as from HGNN.

- HyperGCN Yadati et al. (2019) computes GCN on a clique expansion of a hypergraph. This has an updateable adjacency matrix defined as follows:

$$A_{i,j}^{(l)} = \begin{cases} 1 & (i,j) \in E^{(l)} \\ 0 & (i,j) \notin E^{(l)} \end{cases}  \tag{123}$$

  where

$$E^{(l)} = \{(i_e, j_e) = argmax_{i,j \in e} |X_i^{(l)} - X_j^{(l)}| : e \in \mathcal{E}\}  \tag{124}$$

$$X_v^{(l+1)} = \sigma\left(\sum_{u \in N(v)} ([A^{(l)}]_{v,u} X_u^{(l)} W^{(l)})\right)  \tag{125}$$

  The $X^{(l)} \in \mathbb{R}^{n \times d}$ is the node signal matrix at layer $l$, the $W^{(l)} \in \mathbb{R}^{d \times d}$ is the weight matrix at layer $l$, and $\sigma$ is some nonlinear activation. This architecture has less expressive power than GWL-1.

- HNHN Dong et al. (2020) This is like HGNN but where the message passing is explicitly broken up into two hyperedge to node and node to hyperedge layers.

$$X_E^{(l)} = \sigma(H^T X_V^{(l)} W_E^{(l)} + b_E^{(l)})  \tag{126a}$$

  and

$$X_V^{(l+1)} = \sigma(H X_E^{(l)} W_V^{(l)} + b_V^{(l)})  \tag{126b}$$

  where $H \in \mathbb{R}^{n \times m}$ is the star expansion incidence matrix, $W_E^{(l)}, W_V^{(l)} \in \mathbb{R}^{d \times d}, b_E^{(l)} \in \mathbb{R}^m, b_V^{(l)} \in \mathbb{R}^n$ are weights and biases, $X_E^{(l)}, X_V^{(l)}$ are the hyperedge and node signal matrices at layer $l$, and $\sigma$ is a nonlinear activation function. The bias vectors prevent HNHN from being permutation equivariant.

- UniGNN Huang & Yang (2021) The idea is directly related to generalizing WL-1 GNNs to Hypergraphs. Define the following hyperedge representation for hyperedge $e \in \mathcal{E}$:

$$h_e^{(l)} = \frac{1}{|e|} \sum_{u \in e} X_u^{(l)} \tag{127}$$

  – UniGCN: a generalization of GCN to hypergraphs

$$X_v^{(l)} = \frac{1}{\sqrt{d_v}} \sum_{e \ni v} \frac{1}{\sqrt{d_e}} W^{(l)} h_e^{(l)} \tag{128}$$

  – UniGAT: a generalization of GAT to hypergraphs

$$\alpha_{ue} = \sigma(a^T [X_i^{(l)} W^{(l)}; X_j^{(l)} W^{(l)}]) \tag{129a}$$

$$\tilde{\alpha}_{ue} = \frac{e^{\alpha_{ue}}}{\sum_{v \in e} e^{\alpha_{ve}}} \tag{129b}$$

$$X_v^{(l+1)} = \sum_{e \ni v} \alpha_{ve} h_e W^{(l)} \tag{129c}$$

  – UniGIN: a generalization of GIN to hypergraphs

$$X_v^{(l+1)} = ((1 + \epsilon) X_v^{(l)} + \sum_{e \ni v} h_e) \tag{130}$$

  – UniSAGE: a generalization of GraphSAGE to hypergraphs

$$X_v^{(l+1)} = (X_v^{(l)} + \sum_{e \ni v} (h_e)) \tag{131}$$

All positional encodings are computed from the training hyperedges before data augmentation. The loss we use for higher order link prediction is the Binary Cross Entropy Loss for all the positive and negatives samples. Hypergraph neural network implementations were mostly taken from `https://github.com/iMoonLab/DeepHypergraph`, which uses the Apache License 2.0.

Table 7: Dataset statistics and training hyperparameters used for all datasets in scoring all experiments.

| Dataset Information | | | | | | |
|---|---|---|---|---|---|---|
| Dataset | $|\mathcal{E}|$ | $|\mathcal{V}|$ | $\frac{\Delta_{+,tr}}{\Delta_{-,tr}}$ | $\frac{\Delta_{+,val}}{\Delta_{-,val}}$ | $\frac{\Delta_{+,te}}{\Delta_{-,te}}$ | % of Conn. Comps. Selected |
| CAT-EDGE-DAWN | 87,104 | 2,109 | 8,802/10,547 | 1,915/2,296 | 1,867/2,237 | 0.05% |
| EMAIL-EU | 234,760 | 998 | 1,803/2,159 | 570/681 | 626/749 | 0.6% |
| CONTACT-PRIMARY-SCHOOL | 106,879 | 242 | 1,620/1,921 | 461/545 | 350/415 | 9.3% |
| CAT-EDGE-MUSIC-BLUES-REVIEWS | 694 | 1,106 | 16/19 | 7/6 | 3/3 | 0.14% |
| CAT-EDGE-VEGAS-BARS-REVIEWS | 1,194 | 1,234 | 72/86 | 12/14 | 11/13 | 0.7% |
| CONTACT-HIGH-SCHOOL | 7,818 | 327 | 2,646/3,143 | 176/208 | 175/205 | 5.6% |
| CAT-EDGE-BRAIN | 21,180 | 638 | 13,037/13,817 | 2,793/3,135 | 2,794/3,020 | 9.9% |
| JOHNSHOPKINS55 | 298,537 | 5,163 | 29,853/35,634 | 9,329/11,120 | 27,988/29,853 | 2.0% |
| AIFB | 46,468 | 8,083 | 4,646/5,575 | 1,452/1,739 | 4,356/5,222 | 0.02% |
| AMHERST41 | 145,526 | 2,234 | 14,552/17,211 | 4,547/5,379 | 16,125/13,643 | 4.4% |
| FB15K-237 | 272,115 | 14,505 | 27,211/32,630 | 8,767/10,509 | 10,233/12,271 | 2.1% |
| WIKIPEOPLE-0BI | 18,828 | 43,388 | 27,211/32,630 | 10,254/12,301 | 1,164/1,396 | 0.05% |
| JF17K | 76,379 | 28,645 | 11,907/14,287 | 1,341/1,608 | 1,341/1,608 | 0.6% |

We describe here some more information about each dataset we use in our experiments as provided by Benson et al. (2018b): Here is some information about the hypergraph datasets:

- Amburg et al. (2020a) CAT-EDGE-DAWN: Here nodes are drugs, hyperedges are combinations of drugs taken by a patient prior to an emergency room visit and edge categories indicate the patient disposition (e.g., "sent home", "surgery", "released to detox").

- Benson et al. (2018a); Yin et al. (2017); Leskovec et al. (2007)EMAIL-EU: This is a temporal higher-order network dataset, which here means a sequence of timestamped simplices, or hyperedges with all its node subsets existing as hyperedges, where each simplex is a set of nodes. In email communication, messages can be sent to multiple recipients. In this dataset, nodes are email addresses at a European research institution. The original data source only contains (sender, receiver, timestamp) tuples, where timestamps are recorded at 1-second resolution. Simplices consist of a sender and all receivers such that the email between the two has the same timestamp. We restricted to simplices that consist of at most 25 nodes.

- Stehlé et al. (2011) CONTACT-PRIMARY-SCHOOL: This is a temporal higher-order network dataset, which here means a sequence of timestamped simplices where each simplex is a set of nodes. The dataset is constructed from interactions recorded by wearable sensors by people at a primary school. The sensors record interactions at a resolution of 20 seconds (recording all interactions from the previous 20 seconds). Nodes are the people and simplices are maximal cliques of interacting individuals from an interval.

- Amburg et al. (2020b)CAT-EDGE-VEGAS-BARS-REVIEWS: Hypergraph where nodes are Yelp users and hyperedges are users who reviewed an establishment of a particular category (different types of bars in Las Vegas, NV) within a month timeframe.

- Benson et al. (2018a); Mastrandrea et al. (2015) CONTACT-HIGH-SCHOOL: This is a temporal higher-order network dataset, which here means a sequence of timestamped simplices where each simplex is a set of nodes. The dataset is constructed from interactions recorded by wearable sensors by people at a high school. The sensors record interactions at a resolution of 20 seconds (recording all interactions from the previous 20 seconds). Nodes are the people and simplices are maximal cliques of interacting individuals from an interval.

- Crossley et al. (2013) CAT-EDGE-BRAIN: This is a graph whose edges have categorical edge labels. Nodes represent brain regions from an MRI scan. There are two edge categories: one for connecting regions with high fMRI correlation and one for connecting regions with similar activation patterns.

- Lim et al. (2021)JOHNSHOPKINS55: Non-homophilous graph datasets from the facebook100 dataset.

- Ristoski & Paulheim (2016)AIFB: The AIFB dataset describes the AIFB research institute in terms of its staff, research groups, and publications. The dataset was first used to predict the affiliation (i.e., research group) for people in the dataset. The dataset contains 178 members of five research groups, however, the smallest group contains only four people, which is removed from the dataset, leaving four classes.

- Lim et al. (2021)AMHERST41: Non-homophilous graph datasets from the facebook100 dataset.

- Bordes et al. (2013)FB15K-237: A subset of entities that are also present in the Wikilinks database Singh et al. (2012) and that also have at least 100 mentions in Freebase (for both entities and relationships). Relationships like '!/people/person/nationality' which just reverses the head and tail compared to the relationship '/people/person/nationality' are removed. This resulted in 592,213 triplets with 14,951 entities and 1,345 relationships which were randomly split.

- Guan et al. (2019)WIKIPEOPLE-0BI: The Wikidata dump was downloaded and the facts concerning entities of type human were extracted. These facts are denoised. Subsequently, the subsets of elements which have at least 30 mentions were selected. And the facts related to these elements were kept. Further, each fact was parsed into a set of its role-value pairs. The remaining facts were randomly split into training set, validation set and test set by a percentage of 80%:10%:10%. All binary relations are removed for simplicity. This modifies WikiPeople to WikiPeople-0bi.

- Wen et al. (2016)JF17K: The full Freebase data in RDF format was downloaded. Entities involved in very few triples and the triples involving String, Enumeration Type and Numbers were removed. A fact representation was recovered from the remaining triples. Facts from meta-relations having only a single role were removed. From each meta-relation containing more than 10,000 facts, 10,000 facts were randomly selected.

### D.1 Timings

We perform experiments on a cluster of machines equipped with AMD MI100s GPUs and 112 shared AMD EPYC 7453 28-Core Processors with 2.6 PB shared RAM. We show here the times for computing each method. The timings may vary heavily for different machines as the memory we used is shared and during peak usage there is a lot of paging. We notice that although our data preprocessing algorithm involves seemingly costly steps such as GWL-1, connected connected components etc. The complexity of the entire preprocessing algorithm is linear in the size of the input as shown in Proposition B.20. Thus these operations are actually very efficient in practice as shown by Tables 9 and 10 for the hypergraph and graph datasets respectively. The preprocessing algorithm is run on CPU while the training is run on GPU for 2000 epochs.

Table 8: Timings for our method broken up into the preprocessing phase and training phases (2000 epochs) for the hypergraph datasets.

| Timings (hh:mm) ± (s) | | | Timings (hh:mm) ± (s) | | |
|---|---|---|---|---|---|
| Method | Preprocessing Time | Training Time | Method | Preprocessing Time | Training Time |
| HGNN | 2m:45s±108s | 35m:9s±13s | HGNN | 1.72s±5s | 2m:11s±11s |
| HGNNP | 1m:52s±0s | 35m:16s±0s | HGNNP | 1.42s±0s | 2m:10s±0s |
| HNHN | 1m:55s±0s | 35m:0s±1s | HNHN | 1.99s±0s | 3m:43s±2s |
| HyperGCN | 1m:50s±0s | 58m:17s±79s | HyperGCN | 1.47s±2s | 4m:12s±3s |
| UniGAT | 1m:54s±0s | 1h:19m:34s±0s | UniGAT | 1.85s±0s | 3m:54s±287s |
| UniGCN | 1m:50s±2s | 35m:19s±2s | UniGCN | 2.93s±0s | 3m:15s±19s |
| UniGIN | 1m:50s±1s | 35m:12s±1288s | UniGIN | 2.24s±0s | 3m:17s±18s |
| UniSAGE | 1m:51s±0s | 35m:16s±0s | UniSAGE | 2.04s±0s | 3m:13s±3s |

(a) CAT-EDGE-DAWN     (b) CAT-EDGE-MUSIC-BLUES-REVIEWS

| Timings (hh:mm) ± (s) | | | Timings (hh:mm) ± (s) | | |
|---|---|---|---|---|---|
| Method | Preprocessing Time | Training Time | Method | Preprocessing Time | Training Time |
| HGNN | 4.17s±0s | 2m:34s±1954s | HGNN | 5.84s±1s | 6m:49s±8s |
| HGNNP | 4.54s±0s | 2m:41s±53s | HGNNP | 5.82s±0s | 9m:8s±19s |
| HNHN | 3.06s±0s | 2m:27s±15s | HNHN | 5.95s±0s | 8m:21s±19s |
| HyperGCN | 1.81s±1s | 2m:27s±0s | HyperGCN | 5.74s±0s | 10m:16s±1s |
| UniGAT | 1.91s±0s | 2m:27s±306s | UniGAT | 8.80s±0s | 2m:31s±282s |
| UniGCN | 2.84s±0s | 2m:30s±72s | UniGCN | 6.35s±0s | 6m:9s±957s |
| UniGIN | 3.20s±0s | 2m:27s±1189s | UniGIN | 5.99s±0s | 10m:41s±43s |
| UniSAGE | 1.65s±0s | 2m:27s±0s | UniSAGE | 5.97s±0s | 9m:50s±0s |

(c) CAT-EDGE-VEGAS-BARS-REVIEWS     (d) CONTACT-PRIMARY-SCHOOL

| Timings (hh:mm) ± (s) | | | Timings (hh:mm) ± (s) | | |
|---|---|---|---|---|---|
| Method | Preprocessing Time | Training Time | Method | Preprocessing Time | Training Time |
| HGNN | 23.25s±1s | 25m:41s±17s | HGNN | 4.89s±6s | 1m:27s±8s |
| HGNNP | 23.25s±0s | 19m:52s±49s | HGNNP | 2.12s±0s | 2m:42s±30s |
| HNHN | 24.27s±1s | 5m:12s±63s | HNHN | 2.12s±0s | 2m:39s±42s |
| HyperGCN | 24.00s±0s | 21m:16s±0s | HyperGCN | 2.11s±0s | 40.11s±3s |
| UniGAT | 14.27s±0s | 5m:13s±243s | UniGAT | 2.13s±0s | 3m:18s±8s |
| UniGCN | 25.44s±0s | 5m:51s±1019s | UniGCN | 2.11s±0s | 3m:21s±2s |
| UniGIN | 13.71s±1s | 19m:10s±3972s | UniGIN | 2.11s±0s | 2m:24s±70s |
| UniSAGE | 14.08s±2s | 36m:29s±5s | UniSAGE | 2.11s±0s | 2m:8s±49s |

(e) EMAIL-EU     (f) CAT-EDGE-MADISON-RESTAURANT-REVIEWS

Table 9: Timings for our method broken up into the preprocessing phase and training phases (2000 epochs) for the hypergraph datasets.

| Timings (hh:mm) ± (s) | | | Timings (hh:mm) ± (s) | | |
|---|---|---|---|---|---|
| Method | Preprocessing Time | Training Time | Method | Preprocessing Time | Training Time |
| HGNN | 15.11s±4s | 4m:59s±1s | HGNN | 11.34s±10s | 4m:24s±6s |
| HGNNP | 12.72s±0s | 2m:29s±0s | HGNNP | 6.02s±0s | 4m:13s±2s |
| HNHN | 12.17s±0s | 3m:6s±0s | HNHN | 6.01s±0s | 5m:31s±1s |
| HyperGCN | 12.47s±0s | 49.25s±0s | HyperGCN | 6.32s±0s | 1m:33s±0s |
| UniGAT | 12.74s±0s | 2m:1s±1s | UniGAT | 6.04s±0s | 4m:11s±0s |
| UniGCN | 12.50s±0s | 2m:29s±3s | UniGCN | 5.79s±0s | 4m:12s±0s |
| UniGIN | 12.57s±0s | 2m:16s±3s | UniGIN | 6.64s±1s | 3m:4s±1s |
| UniSAGE | 12.67s±0s | 1m:50s±29s | UniSAGE | 5.79s±0s | 3m:2s±0s |

(a) CONTACT-HIGH-SCHOOL

(b) CAT-EDGE-BRAIN

| Timings (hh:mm) ± (s) | | | Timings (hh:mm) ± (s) | | |
|---|---|---|---|---|---|
| Method | Preprocessing Time | Training Time | Method | Preprocessing Time | Training Time |
| HGNN | 3m:30s±5s | 1h:29m:33s±6s | HGNN | 8m:11s±52s | 37m:18s±9s |
| HGNNP | 3m:34s±1s | 1h:48m:57s±1s | HGNNP | 7m:34s±1s | 47m:56s±1s |
| HNHN | 3m:41s±1s | 2h:9m:34s±1s | HNHN | 6m:21s±1s | 49m:33s±1s |
| HyperGCN | 3m:24s±1s | 58m:27s±1s | HyperGCN | 8m:20s±1s | 28m:25s±1s |
| UniGAT | 3m:50s±1s | 4h:21m:24s±1s | UniGAT | 10m:40s±1s | 1h:54m:36s±1s |
| UniGCN | 3m:38s±1s | 29m:14s±1s | UniGCN | 7m:25s±1s | 2h:40m:20s±1s |
| UniGIN | 3m:50s±1s | 27m:50s±1s | UniGIN | 10m:37s±1s | 2h:48m:35s±1s |
| UniSAGE | 3m:41s±1s | 27m:22s±1s | UniSAGE | 6m:58s±1s | 2h:35m:4s±1s |

(c) WIKIPEOPLE-0BI

(d) JF17K

Table 10: Timings for our method broken up into the preprocessing phase and training phases (2000 epochs) for the graph datasets.

| Timings (hh:mm) ± (s) | | |
|---|---|---|
| Method | Preprocessing Time | Training Time |
| HGNN | 11m:14s±75s | 53m:21s±2845s |
| HGNNP | 11m:10s±21s | 1h:34m:25s±35s |
| HNHN | 5m:15s±395s | 1h:35m:15s±419s |
| HyperGCN | 33.98s±0s | 5m:8s±0s |
| UniGAT | 1m:59s±120s | 2h:2m:47s±25s |
| UniGCN | 34.37s±0s | 1h:17m:38s±2s |
| UniGIN | 34.05s±0s | 1h:16m:38s±7s |
| UniSAGE | 34.36s±0s | 1h:16m:34s±3s |

(a) JOHNSHOPKINS55

| Timings (hh:mm) ± (s) | | |
|---|---|---|
| Method | Preprocessing Time | Training Time |
| HGNN | 17m:9s±164s | 12m:38s±549s |
| HGNNP | 15m:34s±61s | 20m:26s±124s |
| HNHN | 15m:31s±83s | 18m:11s±30s |
| HyperGCN | 15m:46s±32s | 4m:17s±80s |
| UniGAT | 1m:27s±6s | 16m:30s±0s |
| UniGCN | 15m:57s±24s | 18m:42s±170s |
| UniGIN | 16m:14s±73s | 16m:22s±39s |
| UniSAGE | 8m:42s±610s | 8m:49s±324s |

(b) AIFB

| Timings (hh:mm) ± (s) | | |
|---|---|---|
| Method | Preprocessing Time | Training Time |
| HGNN | 4m:1s±11s | 22m:30s±1177s |
| HGNNP | 3m:53s±4s | 39m:30s±3s |
| HNHN | 3m:16s±22s | 44m:7s±71s |
| HyperGCN | 3m:35s±23s | 5m:22s±25s |
| UniGAT | 11.92s±0s | 1h:51m:53s±123s |
| UniGCN | 3m:20s±6s | 39m:18s±51s |
| UniGIN | 3m:21s±8s | 38m:3s±0s |
| UniSAGE | 11.27s±0s | 58m:48s±956s |

(c) AMHERST41

| Timings (hh:mm) ± (s) | | |
|---|---|---|
| Method | Preprocessing Time | Training Time |
| HGNN | 3m:32s±9s | 1h:19m:5s±4684s |
| HGNNP | 3m:26s±10s | 2h:19m:44s±3586s |
| HNHN | 3m:27s±0s | 1h:55m:48s±22s |
| HyperGCN | 3m:28s±0s | 10m:31s±18s |
| UniGAT | 3m:24s±5s | 3h:50m:24s±91s |
| UniGCN | 3m:19s±4s | 1h:39m:46s±13s |
| UniGIN | 3m:17s±0s | 1h:36m:47s±35s |
| UniSAGE | 3m:25s±13s | 1h:37m:16s±102s |

(d) FB15K-237

