# OpenReview forum: "Expressive Higher-Order Link Prediction through Hypergraph Symmetry Breaking"
_TMLR — Accepted by TMLR_

### Review · Reviewer_FDVh · 2024-07-26

**Summary Of Contributions:**

The authors tackle the problem of hyper edge prediction in temporal hyper graphs. The motivation is that many approaches use  the GWL-1 algorithm that fails to distinguish some non-isomorphic cases.
They propose an efficient pre-processing algorithm that randomly drops hyper edges when maximally connected nodes with the same GWL-1 value are detected. They prove that this approach provably increases expressivity (i.e. the pair of k-node sets that became distinguishable) under certain conditions. They empirically evaluate the effectiveness of the approach on several hyper graph neural networks architectures on several dataset and compare it to a baseline of randomly dropping a hyper edge with a 50% uniform chance.

**Audience:**

Yes

**Claims And Evidence:**

No

**Requested Changes:**

The authors make the following claims on the issues regarding existing alternative approaches:
1. some methods depend on a fixed subgraph radius size which prevents capturing symmetries that span
long ranges across the graph
2. some methods add metric information on each node relative to all other nodes which would be very computationally expensive on hyper-graphs
3. some methods specifically tackle cycles (which are responsible for many symmetries) with approaches that are not permutation invariant or that attach cells to cycles yielding much denser hyper edge distributions
4. some approaches use cell attention networks
The underlying notion is that the proposed approach does not have the same issues. It is however unclear if these issues have an effect in practice. It would be desirable to empirically show that this is the case:
1. are there datasets where the hyper edge existence depends on long range dependencies? can it be shown that the proposed approach can deal with these cases? does the distance of the long range dependency have any effect on the performance of the method? (the downstream approach is still based on message passing, so it would be surprising if this was the case)
2. the proposed approach is based on running the GWL-1 algorithm for L steps, so there is a cost associated, how does this cost compare in practice with the approaches that add metric information to nodes?
3. does the proposed approach suffer from the presence of cycles? what is the dependency w.r.t. the density of cycles in input?
4. what is the drawback of cell attention networks? does the approach deal gracefully with the cases that are difficult for cell attention networks?

The authors identify the conditions under which the approach guarantees that an increasing number of node sets become distinguishable: these conditions include the notion of L-separation and controlled growth of node set size. The paper would benefit from showing in practice that these conditions are met in real world cases.

Simple baselines to gauge the difficulty of the tasks are not available. What is the performance of a hyper edge predictor based on a frequency threshold (e.g. if a hyper edge existed in the past above a certain frequency than it will be predicted to exist in the future).

The tabular results for the  empirical evaluation  are not easy to interpret, i.e. is the proposed approach significantly better than the baseline or then the random dropping approach? Please report a critical difference diagram (Demˇsar J (2006) Statistical comparisons of classifiers over multiple data sets. Machine Learning Research 7:1–30) to show the significance of the results (e.g. using https://scikit-posthocs.readthedocs.io/en/latest/generated/scikit_posthocs.critical_difference_diagram.html).

**Strengths And Weaknesses:**

The approach has a low computational complexity and since it is a data pre-processing step, it can be applied to any downstream hyper graph neural networks architecture.

---

> ### Author Response · Authors · 2024-08-13
> **Response to Reviewer FDVh**
>
> We thank the reviewer for insightful comments. To set up the context for discussion, we reiterate the point of the paper first:
>
> Our approach addresses the symmetry problem for hyperlink prediction on hypergraphs. This refers to the automorphism group of the hypergraph. Common neural encoders for the hypergraph, called hyperGNNs, follow the message passing order of the GWL-1 algorithm. This views the training hypergraph as a collection of rooted trees, introducing more symmetries to the hypergraph than the automorphism group on the training hypergraph. For the task of transductive hyperlink prediction, the training and testing hyperedges are on the same set of nodes and separated by time. In such a case, the ground truth automorphism group is actually on the union of all training, validation and testing hyperedges. This means that the ground truth automorphism group is not being recognized by the hyperGNN. We address this issue by breaking the symmetry of the hyperGNN, as explained by Equation 17.
> We address the concerns one by one in the following discussion: (Q: question, R: response)
>
> Q: “some approaches use cell attention ”
> Q: “what is the drawback of cell attention ?”
>
> R: Cell attention networks message pass amongst upper and lower neighborhoods of the edges of chordless cycles on a graph. In terms of the topology of the message passing, this would mean that every chordless cycle is filled in with all possible edge-to-edge chords across all edges in the cycle. This approach is limited to graphs with chordless cycles. So, if the input graph is a forest, then the cell attention network would act like a WL-1 based message passing GNN. We think forest-like graph data is quite abundant. For example, real world decision tree data form rooted forests. Our approach finds a cover of the training hypergraph by a disjoint collection of maximally connected subhypergraphs whose nodes are of the same GWL-1 value. Then, in a very adaptable manner, we perturb the training hypergraph during training to break the symmetry across the entire hypergraph. This is a much more general method and targets the symmetry of the hyperGNN encoder directly. Cell attention networks are also quite expensive when there are many chordless cycles. For graphs, this means a single message passing step is on the order of $O(mn)$ where $m$ is the number of edges.  Actually finding these chordless cycles is even more expensive. According to the implementation in the cell attention network code: https://github.com/lrnzgiusti/can/blob/185050a20707406d24d377fe21b6e447573667fb/utils/utils.py#L362 , the whole search is of the order $O(n!)$ due to the call to a subgraph isomorphism algorithm.
>
> Q: “Are there datasets ...”:
>
> R: Certainly long range dependencies could be correlated with denser hyperedge distributions. Our approach is not intended to directly address this potential correlation since we are more concerned with the automorphism group of the hypergraph and not with the hypergraph’s “path space.” Symmetry on a hypergraph is not directly caused by the length of paths on the hypergraph.
>
> Q: “The proposed ... nodes?”
>
> R: Of course, it is always possible to introduce a metric on a hypergraph. There are many such as the shortest path metric. According to the APSP hypothesis, the all pairs shortest path metric on the hypergraph’s corresponding bipartite graph would take $\Omega((n+m)^3)$ time to compute. It is also possible to introduce potentially physically not-meaningful metrics on the hypergraph. GWL-1 is much cheaper and is also guaranteed to provide information about the isomorphism classes of the hypergraph nodes, which is what we are concerned with.
>
> Q: “Does the ... input?”
>
> R: It is possible that there are many cycles and yet the automorphism group of the graph is the identity group. Thus, the density or number of cycles isn’t the direct cause of a symmetry bias.
>
> Q: “The authors ...  cases.”
>
> R: In Figure 4, we show the boxplots for the equal colored GWL-1 connected components found by our algorithm. On the left (a), we have a hypergraph formed by the hy-MMSBM sampling algorithm for three independent communities. Here the distribution is thin-tailed, so most of the time we would have bounded node set sizes for the equal colored GWL-1 connected components. Since the communities are independent, they are automatically sufficiently $L$-separated. On the right (b) we have three communities that leak connections. The tail is slightly heavier. However, the behavior of the size distribution is similar to (a). With a sufficiently large number of nodes, there should be plenty of opportunity to separate equal colored GWL-1 connected components from different communities, forming sufficiently $L$-separated clusters.
>
> Q: “Simple baselines ...”
>
> R: In Table 7 in the Appendix, we cover all the frequency counts of the positive and negative samples of train/val/test of every dataset.
>
> Q: "The tabular ..."
>
> R: See new Figures 7 and 8

---

### Review · Reviewer_woGA · 2024-08-21

**Summary Of Contributions:**

This paper addresses the problem of higher-order link prediction in hypergraphs, focusing on improving the expressivity of hypergraph neural networks (hyperGNNs) based on the Generalized Weisfeiler-Lehman-1 (GWL-1) algorithm. The authors propose a preprocessing algorithm to identify symmetric substructures in hypergraphs and a data augmentation scheme to break these symmetries during training. The goal is to enhance the ability of GWL-1-based models to capture fine-grained structural information without significantly increasing computational complexity.

**Audience:**

Yes

**Broader Impact Concerns:**

NA.

**Claims And Evidence:**

Yes

**Requested Changes:**

Please refer to the above weaknesses.

**Strengths And Weaknesses:**

Pros:

The paper introduces a novel approach to improving the expressivity of hyperGNNs by targeting topological symmetries. This is an interesting direction that has yet to be widely explored in the context of hypergraph learning.

The authors provide a formal analysis of GWL-1 on hypergraphs from an algebraic topology perspective, which offers new insights into the algorithm's behavior on complex hypergraph structures. This theoretical foundation strengthens the paper's contributions.

The proposed preprocessing algorithm to identify symmetric sub-hypergraphs is efficient, with linear time complexity relative to the input size. This is important for practical applications on large-scale hypergraphs.

The paper reports extensive experiments on both hypergraph and graph datasets, demonstrating consistent improvements across different hyperGNN architectures. This suggests the method's potential broad applicability.

Cons:

It would be helpful to see more detailed comparisons with other state-of-the-art methods for higher-order link prediction. While the authors mention improvements, more context on how their method compares to other recent approaches would strengthen the empirical results.

There's limited discussion. There is limited discussion, particularly in the data augmentation scheme. More analysis in this area could help readers understand the robustness of the approach. Despite these minor points for potential improvement, the paper presents a novel and theoretically grounded approach to an important problem in hypergraph learning. The combination of theoretical analysis, algorithmic innovation, and empirical validation makes a compelling case for the method's effectiveness. The work makes a meaningful contribution to the field.

Despite these minor points for potential improvement, the paper presents a novel and theoretically grounded approach to an important problem in hypergraph learning. The combination of theoretical analysis, algorithmic innovation, and empirical validation makes a compelling case for the method's effectiveness. The work appears to make a meaningful contribution to the field.

---

> ### Author Response · Authors · 2024-09-14
> **Response to Reviewer woGA**
>
> We would like to thank you for your insightful comments. With regard to our work’s relationship to data augmentation, our method falls into the category of “rule-based data augmentation.” These are defined as (stochastic) data augmentations determined by a-priori knowledge about the data. In this case, we know that any representation of an unattributed hypergraph must respect its automorphism group. Our approach is also related to robustness since the automorphism group at test time is not the same as that at training. We have added a paragraph in the related work section (Section 3 page 8) to discuss data augmentation and graph data augmentation in particular. In the Method (Section 5 page 8) and Discussion section (Section 7 page 18 and 19), we also discuss that besides data augmentation symmetry breaking can also be used for feature averaging and ensemble methods.

---

### Review · Reviewer_8Vjw · 2024-08-30

**Summary Of Contributions:**

The paper proposes a method to use symmetry to improve higher order link prediction strategies.  The paper offers a lot of theory, a method using some of these concepts, and then experimental results.

**Audience:**

Yes

**Claims And Evidence:**

No

**Requested Changes:**

It is critical to improve the technical writing of the paper, so individual sentences and paragraphs are formally correct, and every concept is defined before it is used.  To achieve this, a rigorous reading and editing of the complete text is necessary, my detailed comments only are a sample of the issues in the text.

It is also very important to provide sufficient discussion and context to the paper, e.g.,
- for which applications do you expect your method to yield more substantial or less substantial gains?
- what in the computational cost in practice, and how does this trade off to predictive performance gains?
- what are limitations of the method?
- what alternative choices could you have made and what would have been the effect?
- and maybe: how do methods working on symmetries in the data (such as the proposed one) compare to methods working on symmetries in the model (in the spirit of convolutional neural networks or GNNs which constrain the dependencies of nodes on their inputs to be symmetric) ?

**Strengths And Weaknesses:**

Symmetry is an important problem when predicting.  In several domains one has found that specific wiring in a neural network (e.g., a convolutional neural network) can exploit symmetry (e.g., invariance under translation in a picture) simplifies the learning problem.  Looking at a graph-theoretic representation is clearly a good idea to start with.

However, I have several concerns:
* The paper doesn't discuss clearly the expected applicability and limitations of the proposed approach.
* The paper has a lot of minor technical issues which make accumulate and make it very hard to follow after a number of sections (details see below).
* While the paper says (correctly) hypergraph computation is costly in terms of computation, there is no systematic experimental evaluation of the benefits against the added computational cost
* The discussion is mostly restricted to the proposed preprocessing algorithm, there is little discussion of the interaction with other ingredients of the learning

Sec 1.
* "For this task, when the hypergraph is unattributed, it is important to respect the hypergraph’s symmetries, or automorphism group." : this sentence is rather vague.  Until we succeed at predicting all missing links correctly, we can't determine the automorphism group of a graph.  Maybe you want to make a more statistical statement?  In such case one could first try to learn the statistic pattern (maybe expressing some symmetry) after which one could try to respect that pattern during link prediction.  It would help a lot if you could provide an example here of what you mean.
* "GWL-1 views a hypergraph as a collection of trees rooted at the nodes. These rooted trees are formed by viewing each node-hyperedge incidence as an edge and recursively expanding about the nodes and hyperedges alternately." : it is quite common to represent a hyper-edge as a tree, but the part " ... and recursively expanding about the nodes and hyperedges alternately" is unclear to me.
* "We can recover hyperGNNs" : what is "recover" here?  Do you mean "represent"?
* "by expressing the computation of GWL-1 as a matrix equation.": While we can express WL and its variants using matrix equations in a few ways, it is unclear how this brings us closer to a hyperGNN.
* "This means just the neighborhood size of the nodes in hypergraphs can grow exponentially with the number of nodes of the hypergraph" : The neighborhood of a node v is commonly defined as the set of all nodes adjacent to v.  This set is bounded by the total number of nodes in the (hyper)graph, and hence not exponential.  Maybe you mean that the number of edges in a hypergraph can be exponential in the number of nodes of the hypergraph, as for $n$ nodes we get $2^n-1$ possible hyperedges ?
* "In order to address the issue of the expressivity of hyperGNNS for the task of hyperlink prediction" : It is still unclear what "expressivity" exactly is meant here.  this paragraphs forms the motivation of the work and hence deserves being clear.
* "This limits the space for augmentation, which can prevent extreme perturbations of the data." : it is unclear what is "augmentation" here.  The term has meanings in various context, e.g., "data augmentation" refers to adding more data without changing the original data (and hence without perturbing it).


Sec 2.

* $\tilde{B}\subset \tilde{A}$ requires as the first of two conditions that $\tilde{B}\subset \tilde{A}$ which looks like a recursive definition.  maybe the condition must be $B\subset A$ ?  What if $A=B$ but for some elements $m'(e)<m(e)$, the normal interpretation of multisubset is that in that case too $\tilde{B}\subset \tilde{A}$, but not according to the definition provided.
* Multiset union:  Some other authors define $m_{A\cup B} = \max(m_A,m_B)$, which is consistent with set union, while the provided definition is not.  Indeed if $A=\\{1,2\\}$ and $B=\\{2,3\\}$ then $A\cup B=\\{1,2,3\\}=(\\{1,2,3\\},\\{(1,1),(2,1),(3,1)\\})$, but if $A$ and $B$ are seen as multisets we get $A\sqcup B = (\\{1,2,3\\},\\{(1,1),(2,2),(3,1)\\})$
* "an induced subhypergraph is a subhypergraph" : given that two hypergraphs were around in the previous sentence, it is less ambiguous to say "an induced subhypergraph of $H$ is ..."
* "A hypergraph isomorphism is a structure preserving map $\rho = (\rho_V , \rho_E )$ such that both $\rho_V$ and $\rho_E$ are bijective" : this definition deviates from the normal one in literature.  Consider hypergraphs $H=(V_H,E_H)$ and $D=(V_D,E_D)$ with $E_D=E_H$ and $V_D\subset V_H$ (i.e., $H$ is $D$ plus a few isolated vertices), and consider $\phi_V=\\{(v,v)|v\in V_D\\}$ and $\rho_E=\\{(e,e)|e\in E_D\\}$, then both mappings are bijections but $H$ and $D$ are commonly not called isomorphic.
* Definition of neighborhood: The definition says the neighborhood of v is a hypergraph with v and all edges incident with v.  The sentence says "is the subhypergraph of H induced by the set of all hyperedges incident to v".  This is not consistent.  If $H=(\\{1,2,3\\},\\{\\{1,2\\},\\{2,3\\},\\{1,3\\}\\})$, then $(\\{1,2,3\\},\\{\\{1,2\\},\\{1,3\\}\\})$ is not an induced subgraph according to the earlier definition of induced subgraph.
* Def 2.8, second condition.  In formal logic, there is a problem.  An implication $p\Rightarrow q$ is true is either $p$ is false or $q$ is true.  The condition hence means that there does not exist a permutation for which $\pi(S)\neq S'$.  Except for degenerate cases, finding a permutation such that $\pi(S)\neq S'$ is rather easy, implying that except for degenerate cases (the empty graph, the complete graph), there is no "most expressive" g.
* Even if the logic of Def 2.8 would make sense, i would still dislike the word "most" in the term "most expressive", as "most" implies a comparison, and no order relation is defined.


In later sections too, a lot of minor issues obscure the explanation.
As a result, I'm not fully understanding the algorithm presented in Section 5.


The proposed algorithm is a preprocessing algorithm.  Section 6 doesn't seem to provide an in-depth explanation of the link prediction training algorithm that is combined with this preprocessing algorithm, and doesn't discuss potential interactions between the choice of the preprocessing algorithm and the choice of the training algorithm.

---

> ### Author Response · Authors · 2024-09-14
> **Response to Reviewer 8Vjw**
>
> Thank you for your careful reading with long and detailed comments and questions that have been helpful to improve the readability and clarity.
>
> # Sec. 1:
> * Any hypergraph is equivalent to itself under automorphisms. This is already known regardless of whether it is computed or not.
> * The recursive expansion is referring to expanding the partially expanded tree by the hyperedge star-graph tree about each of its leaves as you mentioned.
> * We do not mean “represent” because hyperGNNs do not have to be defined by GWL-1. In HGNN, for example, they are viewing the neural network as a first order Chebychev polynomial approximation of a hypergraph spectral convolution as in the GCN paper. However, from GWL-1 we can *recover* hyperGNNs such as HNHN, see Equation 7, page 7.
> * See previous statement.
> * Any node in the hypergraph can have up to $n-1$ other nodes which it can group together with as a set. Thus, a single node can have as many as $2^{n-1}$ hyperedges in its neighborhood.
> * See Definition 2.8. a representation is “expressive” if no stabilizer $\pi$ on the matrix representation has two node sets equivalent to $\pi$ where the representations equate on the two node sets. This is standard notation in the graph representation literature. The idea is that invariance is already guaranteed by the graph encoder but not necessarily expressive.
> * An augmentation is any transformation on the data. We are augmenting the hyperedge data. This is standard terminology.
> # Sec. 2:
> * Yes, it should say $A \subset B$
> * Yes, it should say multiset sum instead of multiset union.
> * Yes, we have changed this.
> * Your counter example is not an isomorphism since the vertex map is not bijective.
> * We have formally defined what it means to induce a hypergraph by a set of hyperedges after Definition 2.2
> * There is nothing wrong with Definition 2.8. $k$-node expressive means that there is no stabilizer on $H$ with $\pi(S)= S’ \Rightarrow g(S,H)\neq g(S’,H)$. For example, $g$ could assign different values to every possible $S \in {\mathcal{V} \choose k}$
> * Most expressive is standard terminology in representation learning literature, particularly link prediction. It is “most” expressive because it is expressive and fully respects the symmetry of the data.
>
> Algorithm 1 finds the connected components of the equi-colored GWL-1 nodes. The section denoted “Downstream training” describes how the collected hyperedges from the connected components are perturbed.
>
> Regarding your last comments, we have written a discussion section before the conclusion in the paper (Section 7 page 18-19). This section discusses the reason for symmetry breaking with regards to the second law of thermodynamics. If the training and testing automorphism groups are separated by a temporal shift, then we can justify symmetry breaking through an hypergraph topological entropy measure. Here are our responses directly addressing your requests.
> * The method should work best on graphs that have a discrepancy between its GWL-1 node symmetries and its true node symmetries up to automorphism.
> * There is very little computational cost. GWL-1 symmetry might not be the only predictor of future links, however.
> * Of course, if the GWL-1 symmetry group is trivial then there is no need for symmetry breaking.
> * You can always compute distances on the hypergraph. This is expensive, however. (exponential time)
> * See our related work. We mention many such models. Many of these models learn a specific symmetry from training such as universal equivariant hyperGNNs. None of these methods, however, can address symmetry changes from training to testing.

---

> > ### Comment · Reviewer_8Vjw · 2024-09-29
> >
> > Thanks for your answers.
> > I'm not sure which answers correspond to which questions. First I assumed that the unnumbered list of answers was following the same order as my unnumbered list of questions, but several answers don't seem to be a reply to the corresponding question then?
> >
> > E.g., In sec 1:
> > * my first bullet asks how one can compute the automorphism group of an unknown graph.  Your first bullet says that a graph is trivially automorphic under the identity mapping,  This is true but doesn't answer my question.
> > * My 2nd bullet says  "... and recursively expanding about the nodes and hyperedges alternately" is unclear to me.  Your 2nd bullet talks about recursive expansion but isn't really clearer (and doesn't say the text will be improved)
> > * My 3rd bullet asks what "recover" means or how it would work, your 3rd bullet seems to reply, it points to Eq (7) but Eq (7) does not say how one can recover hyperGNNs.  "recover" normally means "return to a normal state" or "reconstruct what has been lost".
> > * My 4th bullet expresses confusion about expressing as a matrix equation without further clarification.  Your 4th bullet seems to point to the answer the my 3rd bullet I didn't understand.
> >
> > It is possible the answers are correct, but unfortunately, (a) they are not so clear I can easily understand them, (b) they don't seem to promise sufficient text improvements to avoid readers get similar questions.
> >
> >
> > > There is nothing wrong with Definition 2.8
> >
> > Maybe your intentions are correct, but formally it one can easily
> > * select different sets $S$ and $S'$ of the same size $k$
> > * select as $\pi$ the identity mapping (which belongs to $stab(H)$)
> > * observe that $\pi(S) = S'$ is False
> > * observe that therefore $\pi(S') = S \Rightarrow ...$ is True (whatever is "...")
> > * observe that there does exist a $\pi$ such that $\pi(S') = S \Rightarrow ...$ is True.
> > * observe that the second condition therefore is False, as it claims such $\pi$ does not exist.
> >
> > The conclusion is that no $k$-node most expressive $g$ exist as soon as $|\mathcal{V}|>k$.
> >
> > > * There is very little computational cost. GWL-1 symmetry might not be the only predictor of future links, however.
> > > * Of course, if the GWL-1 symmetry group is trivial then there is no need for symmetry breaking.
> >
> > I'm not really understanding this: usually any computation with non-trivial graph symmetries are very expensive.  The author answers refer several times to trivial automorphisms and trivial symmetry groups, but it is unclear how these could bring information so I guess that these are not the cases which would bring added value.
> >
> > Saying the computational cost is small (even if true) does not necessarily say how computational cost trades of with prediction performance, nor does it give any concrete idea of the cost (e.g., in terms of the size or complexity of the input).
> >
> > Section 7 adds some useful but quite vague discussion.
> > * It uses the term "temporal shift" without introducing it, maybe the idea is that the test set is the same graph as the training graph but now with new (most recently added) edges?
> > * It is highly unclear whether the second law of thermodynamics says anything usefum on real-world information systems which are in general not closed systems (e.g., they typically get inputs and outputs) and because one typically is not interested in the stationary state in the limit (we know that in $10^{50}$ years most protons will have decayed and the universe will achieve a thermodynamical equilibrium, but usually we are interested in predicting what will happen the next few seconds or days).

---

> > > ### Comment · Action_Editor_pwb4 · 2024-10-09
> > >
> > > Authors:  The reviewer 8Vjw provides some suggestions in how the paper could be made more clear.  Most of these are simply asking for a bit more exposition in the paper to help explain the technical machinery.  Can you also attempt to address these?
> > >
> > > The reviewer 8Vjw also poses a question about your Definition 2.8.  Can you please respond to this to see if it can be made more clear?
> > >
> > > I think adding a response here or adding a few more points of clarity in the paper would improve the final resulting paper.

---

> > > ### Author Response · Authors · 2024-10-10
> > > **Response to Reviewer 8Vjw**
> > >
> > > Thank you for your response.
> > >
> > > In response to your questions:
> > >
> > > 1. Sorry if the response caused confusion. The group of automorphisms consist of transformations on the data that convert the data into data that is *considered equivalent*. What we meant was that the automorphism group does not need to be computed. It only needs to be recognized by the encoder.
> > >
> > > 2. The nodes and hyperedges can be viewed as *incident* to each other. As said in the text: "These rooted trees are formed by viewing each node-hyperedge incidence as an edge and recursively expanding about the nodes and hyperedges alternately." This means that we can view the nodes and hyperedges as "vertices" connected by edges and do a recursive expansion. We have provided a simple example in that paragraph for two layers of GWL-1. For a complete explanation there would have to be many formal definitions so the example is kept to the definition of a rooted tree. Figure 2 gives an illustration of the expansion after having gone through the formal definitions.
> > >
> > > 3. Yes, we are recovering hyperGNNs because they can be defined on hypergraphs independently of GWL-1. This is considered a "recovery" because hyperGNNs learn the task on the hypergraph while GWL-1 is just an algorithm to approximate isomorphism on hypergraphs.
> > >
> > > 4. I hope my explanation in (3) answers your question.
> > >
> > > * To counter Definition 2.8 part 2, you would have to show that for any $g$ and any $S,S' \subset V$ with $\lvert S \rvert = \lvert S' \rvert = k$ there is a $\pi$ so that $\pi(S)=S' \text{ and } g(S,H)=g(S',H)$. In your counter, bullet 3 states "observe that it is False that $\pi(S)=S'$. This would always make $\pi(S)=S' \text{ and } g(S,H)=g(S',H)$ false and so cannot counter Definition 2.8 part 2.
> > >
> > > 1. Yes, more computation does not always correlate to more predictive performance. However, any hypergraph computation would be in time and space $\Omega(2^n)$ for $n$ nodes. Thus efficiency is very important. The trivial automorphism group is common in the ground truth hypergraph as we have found experimentally and as discussed in the discussion section. This is why we mention the trivial symmetries.
> > >
> > > 2. Yes, temporal shift in this paper’s context means that the testing hyperedge distribution is temporally later than the training hyperedge distribution. In general, a temporal shift means that the training distribution has a “happened before” relationship with the testing distribution, see [1].
> > >
> > > 3. The second law of thermodynamics states that the change in entropy is always positive for a closed system. We have rewritten the discussion section to more formally explain how the second law of thermodynamics affects the node isomorphism classes. We do not assume a long time into the future to form the test hyperedges. In fact, in the limit of infinite time, the hypergraph would form a complete hypergraph, which has maximal symmetry with $\Delta S=0$. We only assume a change in entropy, namely $\Delta S>0$.
> > >
> > > [1] Yao, Huaxiu, et al. "Wild-time: A benchmark of in-the-wild distribution shift over time." Advances in Neural Information Processing Systems 35 (2022): 10309-10324.

---

### Decision · Action_Editor_pwb4 · 2024-10-12

**Recommendation:** Accept as is

**Comment:**

The paper has gone through revisions, and I believe is now in good state.  There may be more suggestions, but I think the paper now hits the right notes.

**Audience:**

Link prediction in graphs is a classic ML problem.  Extending this to hyper graphs is an interesting direction that extends the context, and this paper provides useful progress towards that goal.  The reviewers agree it is relevant for the audience.

**Claims And Evidence:**

This is a technically dense paper that address a question in link prediction among hyper graphs, and how symmetry breaking can help in this task.  The main "Claims and Evidence" concern that arose in the reviewing and revisions of this paper was if mathematical methods and claims were not just correct, but accessible to the TMLR audience.  I believe the expert reviewers are a good representation of the potential audience, and are (for the most part) satisfied with presentation.  The authors have shown willingness to evolve the presentation to make the paper more accessible, and I think the current state is good.